# Energetics of Surface Melt in West Antarctica

Madison L. Ghiz[1], Ryan C. Scott[2], Andrew M. Vogelmann[3], Jan T. M. Lenaerts[4], Matthew Lazzara[5], Dan Lubin[1]

[1]Scripps Institution of Oceanography, University of California San Diego, La Jolla CA 92093-0206 USA
[2]Science Systems and Applications, Inc., Hampton VA 23666 USA
[3]Environmental and Climate Sciences Department, Brookhaven National Laboratory, Upton NY 11973-5000 USA
[4]Department of Atmospheric and Oceanic Sciences, University of Colorado, Boulder CO 80309-0311 USA
[5]Antarctic Meteorological Research Center, SSEC, University of Wisconsin, Madison WI 53706 USA

*Correspondence to*: Dan Lubin (dlubin@ucsd.edu)

**Abstract.** We use reanalysis data and satellite remote sensing of cloud properties to examine how meteorological conditions alter the surface energy balance to cause surface melt that is detectable in satellite passive microwave imagery over West Antarctica. This analysis can detect each of the three primary mechanisms for inducing surface melt at a specific location: thermal blanketing involving sensible heat flux and/or longwave heating by optically thick cloud cover, all-wave radiative enhancement by optically thin cloud cover, and föhn winds. We examine case studies over Pine Island and Thwaites Glaciers, which are of interest for ice shelf and ice sheet stability, and over Siple Dome, which is more readily accessible for field work. During January 2015 over Siple Dome we identified a melt event whose origin is an all-wave radiative enhancement by optically thin clouds. During December 2011 over Pine Island and Thwaites Glaciers, we identified a melt event caused mainly by thermal blanketing from optically thick clouds. Over Siple Dome, those same 2011 synoptic conditions yielded a thermal blanketing-driven melt event that was initiated by an impulse of sensible heat flux then prolonged by cloud longwave heating. The December 2011 synoptic conditions also generated föhn winds at a location on the Ross Ice Shelf adjacent to the Transantarctic mountains, and we analyse this case with additional support from automatic weather station data. In contrast, a late-summer thermal blanketing period over Pine Island and Thwaites Glaciers during February 2013 showed surface melt initiated by cloud longwave heating then prolonged by enhanced sensible heat flux. One limitation thus far with this type of analysis involves uncertainties in the cloud optical properties. Nevertheless, with improvements this type of analysis can enable quantitative prediction of atmospheric stress on the vulnerable Antarctic ice shelves in a steadily warming climate.

## 1 Introduction

The contribution of West Antarctic mass loss to sea level rise, presently the second largest cryospheric contribution to sea level rise after the Greenland Ice Sheet (Mouginot et al., 2019; Rignot et al., 2019), is driven by a complex mechanical and thermodynamic system involving grounded ice sheets, their floating ice shelf extensions, and the surrounding ocean and atmosphere. While a warming ocean causes a retreat of West Antarctic ice sheet grounding lines on numerous reverse slopes,

by gradually accelerating the ice sheet outflow via the well-known marine ice sheet instability (Weertman, 1974; Oppenheimer, 1998; Joughin et al., 2014; Alley et al., 2015), the ice shelves mitigate this outflow through the buttressing they provide by being in contact with adjacent land masses (Fürst et al., 2016). But the ice shelves are themselves thinning via basal melting from the warming ocean (Pritchard et al., 2012; Paolo et al., 2015), which compromises their buttressing strength and also enhances the overall meltwater loss of Antarctic glacial ice (Adusumilli et al., 2020). Structural integrity of an ice shelf can be further compromised when surface meltwater filters through crevasses into its interior mass, rendering the extremities more vulnerable to wave action (DeConto and Pollard, 2016; Bell et al., 2018). Extensive summer melt ponds occurring in a warming atmosphere were the major factor in the loss of the Larsen B Ice Shelf in 2002 (Scambos et al., 2003; van den Broeke, 2005; Glasser and Scambos, 2008). The loss of this ice shelf immediately facilitated faster ice calving of the upstream glaciers (Scambos et al., 2004). In 2008 similar ice shelf failures occurred on the Wilkins Ice Shelf, at the base of the Antarctic Peninsula near West Antarctica proper (Scambos et al., 2009). Surface and lower tropospheric warming are now understood to prevail throughout West Antarctica and across the Ross Ice Shelf (RIS) as far as Ross Island (Steig et al., 2009; Bromwich et al., 2013). Lhermitte et al. (2020) report satellite observational evidence of a corresponding ice shelf structural weakening in the Pine Island and Thwaites Glaciers region of West Antarctica over the past decade.

Remote sensing studies now document frequent warm-season surface melting over West Antarctica and the Ross Ice Shelf (e.g., Kingslake et al., 2017). The energetics of a major melt event over West Antarctica during January 2016 were measured with modern atmospheric science equipment during the joint US Antarctic Program and Department of Energy Atmospheric Radiation Measurement (ARM) user facility's West Antarctic Radiation Experiment (AWARE; Nicolas et al., 2017; Lubin et al., 2020). These measurements provided insight into the role of atmospheric thermodynamics and cloud radiative properties in generating local surface melt. But in contrast to the Antarctic Peninsula and Greenland Ice Sheet, West Antarctic melt events tend to be shorter in duration and exhibit greater spatial, interannual and intra-seasonal variability. Remote sensing assessment of their total meltwater equivalent (e.g., Kuipers Munneke et al., 2012a), which is much smaller than that of basal melting, can give the impression that surface melt might not be an important consideration. But the potential for West Antarctic surface melt to aggravate ice mass loss involves structural degradation of ice shelves through ponding and hydrofracturing, as has already happened throughout much of the Antarctic Peninsula region. Recent studies of Antarctic ice mass balance now account for spatial and temporal variability on multiple scales (Lenaerts et al., 2018; Donat-Magnin et al., 2020; Adusumilli et al., 2020). When evaluating the potential impact of surface melt in West Antarctica, one should focus on assessing the frequency and duration of melt events directly on the vulnerable ice shelves, and also on determining the specific physical mechanisms causing each melt event.

The objective of this work is to determine if readily available satellite remote sensing and meteorological reanalysis data can be used to identify the mechanisms that drive specific Antarctic surface melt events: thermal blanketing involving sensible heat flux and/or longwave heating by optically thick cloud cover, all-wave radiative enhancement by optically thin cloud

cover, and föhn winds. Scott et al. (2019) identify the large-scale meteorological drivers of West Antarctic surface melt, and the approach presented here considers their application to specific locations using available satellite and surface data. If successful, then this approach can be used to assess future risk to the vulnerable West Antarctic ice shelves. For example, if melt events occur frequently under common polar meteorological phenomena such as optically thin clouds that produce the

all-wave radiative enhancement, then the stress on the ice shelves might be perennially constant. Conversely, if melt events occur mainly under optically thick clouds only associated with strong atmospheric rivers (e.g., Wille et al., 2019), then one might expect more of a long-term risk in a warming atmosphere. Ultimately multi-year assessment of melt event mechanisms would need to be understood in terms of the large-scale meteorological drivers (Scott et al., 2019) to make such a risk assessment. Here we demonstrate with case studies that each of the above three melt-inducing mechanisms can be

identified in satellite and reanalysis data.

## 2 Data and Methods

Over the cryosphere the surface energy balance (SEB) can be expressed in terms of the melt energy $ME$ (W m$^{-2}$):

$$ME = F_{SW}^{\downarrow} - F_{SW}^{\uparrow} + F_{LW}^{\downarrow} - F_{LW}^{\uparrow} + F_{SH} + F_{LH} - G \qquad (1)$$

where the individual energy components are the downwelling and upwelling shortwave (SW) and longwave (LW) radiation, the sensible heat flux (SH), the latent heat flux (LH) and the ground conduction $G$. The sum of the four SW and LW fluxes is the net radiation. The sum of SH and LH fluxes is the net turbulent flux, and here we use the European Centre for Medium-Range Weather Forecasts (ECMWF) convention where a positive sign signifies energy going into the surface. Advection of air warmer than $0^{\circ}$C appears in the $ME$ as positive SH flux, whose magnitude depends on both the air temperature gradient

and the wind speed. Strictly speaking equation (1) is valid when the snow surface temperature $T_s$ is at or above the melt point. If $T_s$ is below the melt point and the SEB doesn't close (i.e., the net radiation is not balanced by the sum of the other energy components), it is likely due to ground conduction. Local radiative heating of a snowpack can induce melt at temperatures as low as -2$^{\circ}$C by internal scattering and absorption (e.g., Nicolas et al., 2017). If $T_s$ is at or above freezing a positive $ME$ maintains surface melting while a negative $ME$ represents a surface cooling that if sustained will reduce the

surface temperature below freezing. A negative $ME$ also represents a phase change (i.e., refreezing of the surface, if the $T_s$ is at the melt point. The actual cooling happens through LW radiation and ground conduction. On daily timescales, $G$ over Antarctic firn is usually an order of magnitude smaller than the individual radiative and turbulent flux components (e.g., van As et al., 2005; Fisher et al. 2015), though it can become somewhat important on sub-daily timescales (i.e., warming of the snowpack in the morning, and cooling it at night, after potential refreezing).

If the *ME* remains positive across at least two diurnal cycles, then this condition combined with skin or 2-m air temperatures at or just below freezing is often associated with a surface melt that is detectable in satellite passive microwave (PMW) data (Nicolas et al., 2017). This does not mean that surface melt is occurring throughout those diurnal cycles. Melt occurs only when $T_s$ is between -2$^{\circ}$C and 0$^{\circ}$C, depending on surface microphysics. At colder $T_s$, the positive *ME* goes into warming the snowpack but does not cause detectable melt. The PMW data are instantaneous observations made twice daily (morning and evening overpasses). If the PMW-measured brightness temperature ($T_b$, section 2.1 below) is consistent with a significant increase in surface emissivity as compared with the previous observation, this signifies a moistening of surface firn layer and/or accumulation of meltwater in response to a positive *ME* at $T_s \geq$ -2$^{\circ}$C. Identification of a time interval in the *ME* time series that remains positive across two or more diurnal cycles should therefore be regarded as a strong indicator of satellite-detectable melt at some point during the interval.

The largest individual terms in (1) are the upwelling and downwelling radiative fluxes, and they are strongly modulated by cloud cover, which is extensive over West Antarctica (Scott et al., 2017). Therefore the net (downwelling minus upwelling) radiative fluxes are just as capable of driving *ME* > 0 for extended time periods as a strong impulse of positive SH flux. The result is that three distinct mechanisms for inducing surface melt can be at play over West Antarctic ice sheets, either individually or in conjunction reinforcing each other.

One mechanism is thermal blanketing. If an airmass contains overcast cloud cover within a few hundred meters of the surface having liquid water path (*LWP*) > 50 gm$^{-2}$, this cloud cover will radiate in the LW as a blackbody at very close to surface temperature, while also attenuating the net SW flux. The result is a surface net LW flux close to zero, and sometimes even positive, along with a constantly positive net SW flux that has a diurnal cycle of relatively small amplitude. If the net turbulent flux is also positive such that the *ME* remains positive over two more diurnal cycles, this will usually induce surface melt, if the starting skin temperature is warm enough (e.g., Trusel et al., 2013). This situation prevailed during the large-scale January 2016 melt event over West Antarctica (Nicolas et al., 2017). Wille et al. (2019) have correlated most Antarctic surface melt events with the presence of atmospheric rivers (ARs). If ARs impinging on the Antarctic continent tend to bring mainly large cloud *LWP*, then thermal blanketing would be a widespread source of stress on the ice shelves.

A second mechanism involves an all-wave (SW plus LW) radiative enhancement by optically thin clouds. Bennartz et al. (2013) discovered this cloud radiative effect and showed that is extensive over the Greenland Ice Sheet (GIS) during warm summers that drive surface melt. When overcast or broken cloud cover has *LWP* between 10-40 g m$^{-2}$, generally very common in the Antarctic atmosphere (e.g., Bromwich et al., 2013; Scott & Lubin 2014; 2016), this cloud cover will radiate substantially toward the surface in the LW while still allowing large SW fluxes to reach the surface. In combination with a mostly positive net turbulent flux, these clouds can often prolong a positive *ME* over multiple diurnal cycles, causing surface

melt. Van Tricht et al. (2016) found an additional role for optically thin low cloud cover, in slowing down the refreezing of

meltwater, and this effect may also appear in one of our case studies.

A third mechanism very common throughout Antarctica is a föhn wind (Elvidge and Renfrew, 2016). The föhn effect occurs when an airmass crosses high terrain such as a mountain range. As the airmass is forced upslope it expands and cools, and the moisture condenses and may form clouds or precipitation, releasing latent heat. Adiabatic descent on the lee side of the

high terrain warms the air even more substantially than the latent heat release and, combined with turbulent mixing upon reaching the lower terrain, brings a large positive turbulent flux input to the surface, potentially great enough to initiate surface melt. Föhn winds are especially prevalent on the lee side of the Antarctic Peninsula, causing stress to the Larsen C Ice Shelf (e.g., Elvidge et al., 2015; King et al., 2017; Datta et al., 2019). However, due to widely varying high terrain over Antarctica, in particular the Transantarctic Mountains, föhn winds can occur and impact an ice shelf depending if the

prevailing synoptic conditions yield airflow perpendicular to mountainous terrain (e.g., Zhou et al., 2018).

## 2.1 Melt Detection

We identify the Antarctic surface melt events with a standard PMW technique using the Defense Meteorological Satellite Program Special Sensor Microwave Imager/Sounder (SSMIS), but with a new NASA-supported Making Earth System Data

Records for Use in Research Environments (MEaSUREs) data product archived at the National Snow and Ice Data Center (NSIDC). We use the Equal-Area Scalable-Earth Grid version 2 (EASE-Grid 2.0) Level-2 PMW brightness temperature ($T_b$) at 19.35 GHz with horizontal polarization (19 GHz-H; K-band) from the evening overpass at 25-km grid spacing (Brodzik et al., 2016, updated 2020). We base our melt detection technique on an algorithm originally proposed by Zwally & Feigles (1994) and subsequently refined and validated by Torinesi et al. (2003) and Tedesco (2009). For a given grid cell, surface

melt is detected when the PMW $T_b$ measurement exceeds the prior cold season average by 30 K. The cold season average is constructed by averaging daily $T_b$ measurements from 1 April of the prior year through 31 March of the given year. This average is then repeated twice, each time after removing daily values >30 K above the previous average.

This technique is generally used to detect and map surface melt over large areas and on seasonal timescales. Here we

examine monthly $T_b$ time series in the three regions depicted in Figure 1. The Pine Island and Thwaites Glacier region presents the greatest concern for West Antarctic Ice Sheet (WAIS) loss (e.g., Alley et al., 2015). Siple Dome is a site at an intermediate elevation on the WAIS (607 m above sea level) that has a multi-decadal automatic weather station (AWS; Lazzara et al., 2012) record and a US Antarctic Program (USAP) summer field camp that has been used for some field work on the physics of snowmelt (Das and Alley 2005; 2008). Siple Dome is considered here because it is accessible by the US

Antarctic Program for future field work. In addition to the AWS, the University of Wisconsin Antarctic Meteorological

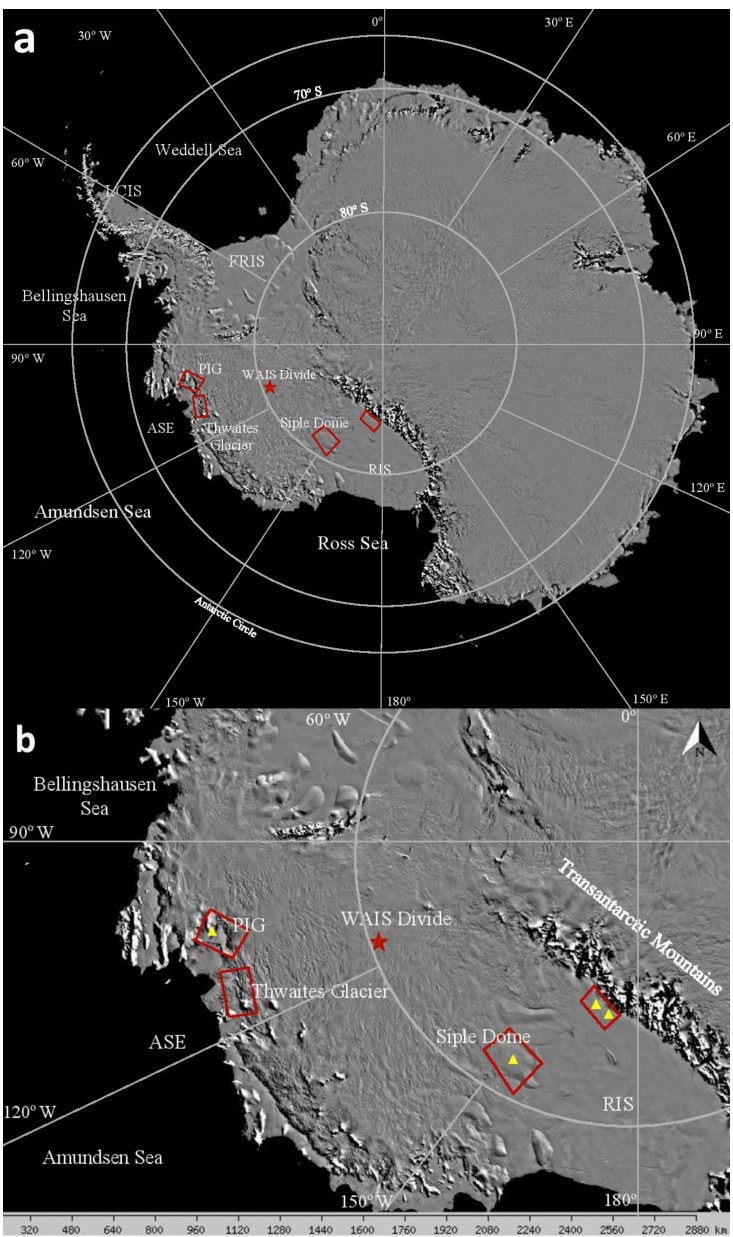

**Figure 1. Maps of (a) Antarctica and (b) West Antarctica showing the case study locations. The red boxes depict the regions from which the satellite and reanalysis data are extracted for analysis. The AWARE data were collected at the WAIS Divide Ice Camp, shown by the red star. The locations of automatic weather stations are shown as yellow triangles. Abbreviations are Ross Ice Shelf (RIS), Amundsen Sea Embayment (ASE), Pine Island Glacier (PIG), Filchner-Ronne Ice Shelf (FRIS) and Larsen-C Ice Shelf (LCIS). Figure constructed from the Mosaic of Antarctica (Scambos et al. 2007; Haran et al., 2014).**

Research Center archives manual surface weather observations from numerous field camps and expeditions, and some of these are available for our case studies, over Pine Island Glacier and Siple Dome.

We choose a third location on the Ross Ice Shelf (RIS) near the Transantarctic mountains that contains two AWS, Tom (84.430°S, 171.455°W) and Sabrina 84.248°S, 170.044°W), whose data have suggested the presence of strong föhn winds. In the AWS measurements, a föhn condition can be inferred from an increase in wind speed along with a south to southeasterly wind direction from the Transantarctic mountains. Each of the regions depicted in Figure 1 contains between 800-1300 25-km EASE-Grid cells. This gives us an opportunity to examine local-scale spatial variability resulting perhaps from varying topography or differential melting and refreezing frequency across the local domain, in addition to time variation. In the monthly $T_b$ time series, we identify melt events of short duration (<5 days) by comparing the daily mean, median, and range with the prior cold season average and the 30-K melt detection threshold. Short duration melt events provide relatively straightforward case studies in which we can readily identify the changing meteorological conditions and shifts in individual *ME* components that lead to melt onset and subsequent recovery. Such case studies allow us to observe the basic physics and develop an understanding of what is driving these surface melt events at a local spatial scale.

**2.2 Surface Energy Budget Analysis**

For our SEB analysis, we use the fifth-generation ECMWF meteorological reanalysis data (ERA5; Hersbach et al., 2020). Previous studies have shown better agreement between ECMWF data and Antarctic in situ data than other reanalysis models (e.g., Lenaerts et al., 2017). The ERA5 model physics includes prognostic determination of cloud water and ice, cloud fraction, rain and snow (Hersbach et al., 2020), more modern atmospheric radiative transfer schemes than its predecessor ERA-Interim (Dee et al., 2011), and a sophisticated snow component in the land surface model (Dutra et al. 2010). We compute *ME* using the surface radiative and turbulent fluxes on a 0.5°x0.5° latitude-longitude grid with hourly time resolution. Other ERA5 fields we analyse include the near-surface (2 m) air temperature, skin temperature, and 850 hPa wind components.

Because of known errors in polar cloud microphysics simulated by ERA5 and other reanalysis and regional models (e.g., Silber et al., 2019), we found it necessary to supplement the ERA5 *ME* calculations with satellite-retrieved cloud properties. We therefore use satellite data products from the NASA Cloud and Earth's Radiant Energy System (CERES) program; specifically, the synoptic 1-degree (SYN1deg) data product. Here CERES top-of-atmosphere (TOA) fluxes, surface fluxes, cloud masking and cloud properties are interpolated to hourly time resolution using geostationary satellite data and gridded to 1° in both latitude and longitude. The SYN1deg product contains NASA A-Train retrievals of cloud *LWP* and *IWP* based primarily on the Moderate-Resolution Imagine Spectroradiometer (MODIS) data from the Aqua spacecraft (Rutan et al.,

2015). Radiometric calibration uncertainty with the MODIS sensor itself is generally taken to be 5% in all bands, for the purpose of evaluating retrieval uncertainties (Platnick et al., 2017). In the MODIS radiative transfer-based retrieval algorithms that use an independent homogeneous pixel approximation, uncertainty in cloud optical depth is of order 10% in the range 3-20 (Platnick et al., 2004; 2017), and increases for both smaller and larger cloud optical depths. Over polar regions, Khanal and Wang (2018) have identified additional uncertainties and biases resulting from mixed-phase cloud effects, large solar zenith angles, and cloud spatial inhomogeneity. For the purpose of this study, MODIS-based cloud property retrievals have shown consistency with ground-based remote sensing data from West Antarctica (Wilson et al., 2018), sufficient to discriminate between optically thin and optically thick clouds associated with the distinct mechanisms that induce surface melt.

We analyse our case studies by calculating the *ME* with ERA5 radiative and turbulent fluxes, and then examine the CERES SYN1deg cloud *LWP* and *IWP* as a separate check on the realism of cloud properties simulated by ERA5. Justification for this approach is given in Appendix A.

## 3 Results and Discussion

We organize this work into four case studies, the first three of which involve synoptic conditions that drive surface melt events lasting several days at one location. The final case involves synoptic conditions that drive surface melt over the entire Amundsen Sea Embayment (ASE), with contrasting mechanisms at each of the locations considered herein.

### 3.1 Siple Dome January 2015

Our first case study reveals evidence of an all-wave radiative enhancement by optically thin clouds, which led to satellite-detected surface melt on Siple Dome between 5-7 January 2015. As seen in Figure 2, during these days a low-pressure system over the Ross-Amundsen Sea experienced blocking by a weak ridge of high pressure. This synoptic set-up drove a warm, moist marine air intrusion over Marie Byrd Land, which subsequently descended over Siple Coast, causing adiabatic warming and drying of the airmass. This descent may have reduced the optical thickness of any previously thick clouds into the Bennartz et al. (2013) thin cloud range ($LWP$ = 10-40 g m$^{-2}$).

Figure 3a shows the daily $T_b$ statistics throughout the Siple Dome region depicted in Figure 1. The surface melt detected by the satellite, using the 30 K threshold, begins in some of the region on 5 January and extends through most of the region over the next two days. This is seen in the relative number of $T_b$ data points above and below the 30 K threshold as depicted by the daily box plots. Figure 3b shows estimates of the surface emissivity sampled from five grid cells with $T_b$ values ranging from the 5th to 99th percentiles on 6 January. These grid cells were chosen from within the Siple Dome region with the

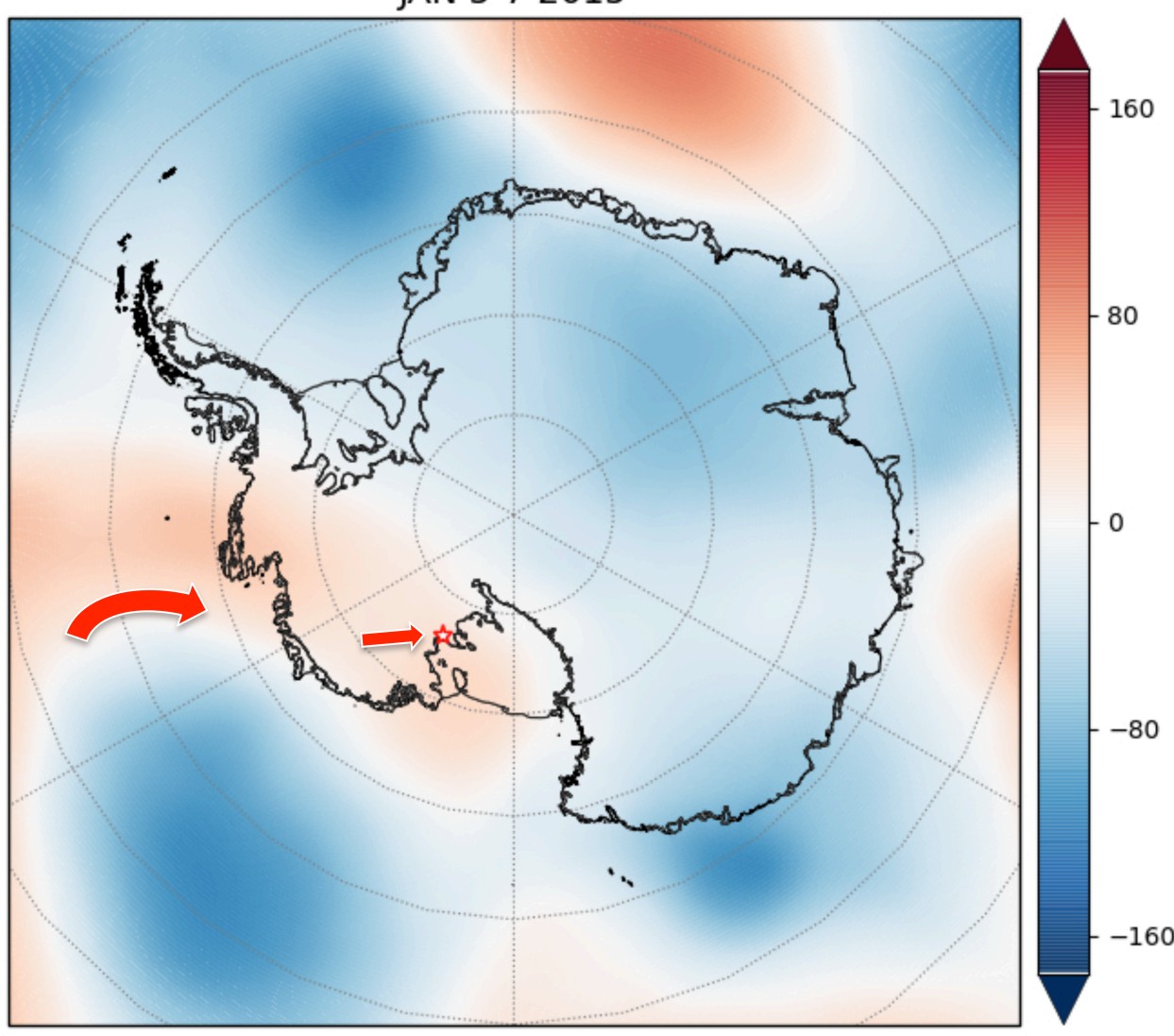

JAN 5-7 2015

$Z_{700}$ Composite Anomaly - meters

**Figure 2. Composite anomaly of ERA5 700 hPa geopotential height for 5-7 January 2015. The red star depicts the location of Siple Dome. The red arrows depict the direction of 700 hPa winds relevant to the case study.**


criteria that they have a fully overlapping ERA5 grid cell, and span a range of 5 to 99th percentile referenced to the max $T_b$ observed in the region. Here surface emissivity is approximated as the ratio of the satellite-measured $T_b$ to the ERA5 skin temperature. Before this short melt event, and also beginning four days after it ends (after the 12th), the surface emissivity appears to be spatially uniform. During the melt period the surface shows large variability in emissivity throughout the

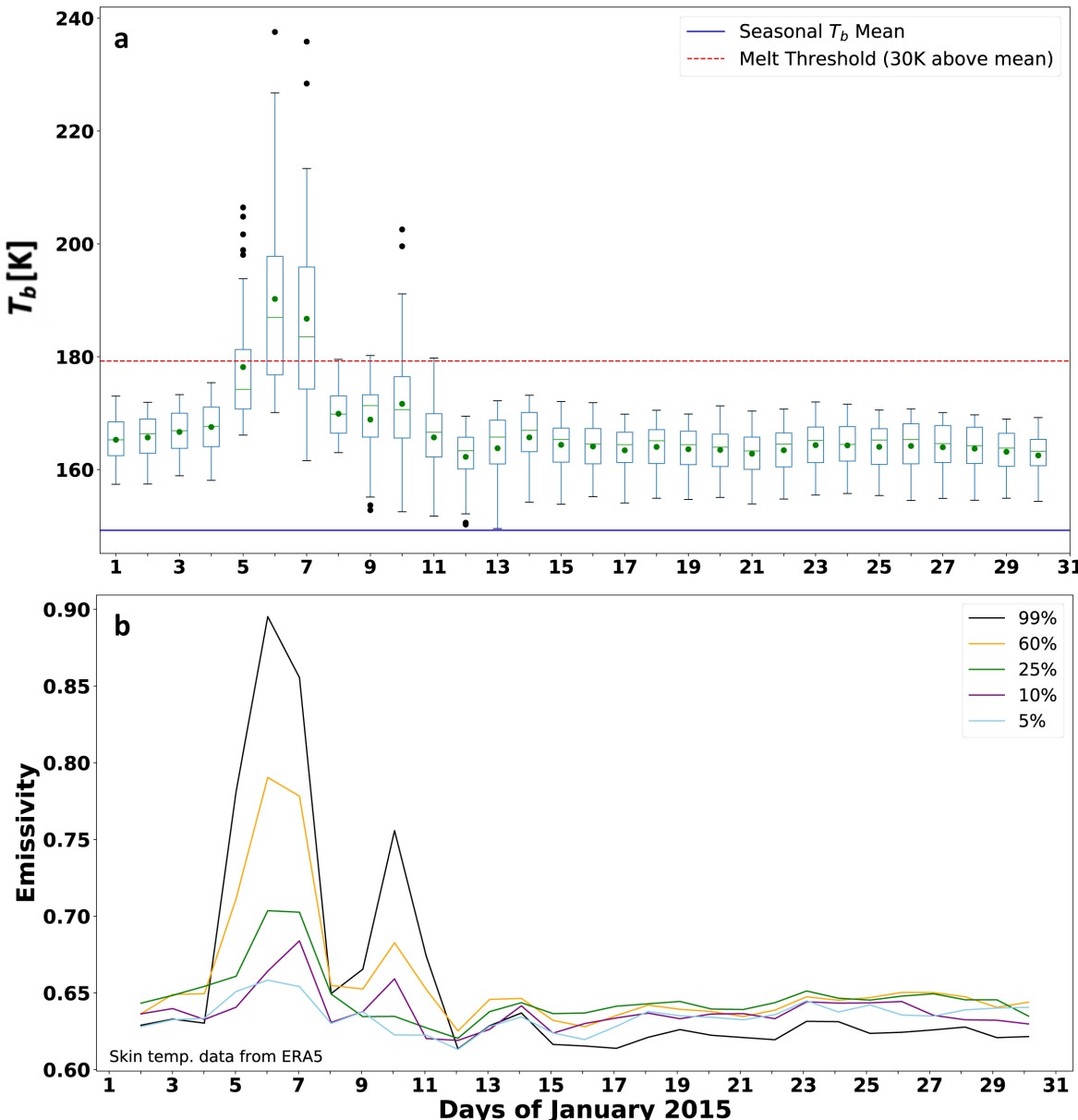


**Figure 3. (a) Time series of daily evening overpass SSMIS brightness temperatures $T_b$ over the Siple Dome region during January 2015 as daily statistics, with the box denoting the first to third interquartile range ($Q_1$ to $Q_3$), the horizontal line in the box denoting the median, the green dot denoting the mean, the whiskers denoting the distance 1.5 ($Q_3$ - $Q_1$), and individual black points beyond them denoting outliers beyond the range 1.5 ($Q_3$ - $Q_1$). The blue**
**horizontal line is the prior cold season mean and the red horizontal line is the standard melt detection threshold lying 30 K above the prior cold season mean (Tedesco 2009). (b) Five estimates of surface emissivity sampled throughout the region with percentiles referenced to the maximum $T_b$ value in the region on 6 January.**

region, possibly reflecting differential surface properties resulting from non-uniform snow accumulation or refreezing from

prior melt periods. Examples of spatial variability in the satellite-measured $T_b$, on the day when the surface melt is most

pronounced, are shown in Appendix B.

Figure 4 presents time series of the individual $ME$ components. The shaded period of interest contains the melt onset, peak

and decrease to when most pixels show no satellite-detected melt. Cloud cover reduces the net SW flux to a monthly

minimum on 6 January, while at the same time the net LW flux rises from < -50 W m$^{-2}$ to ~ -25 W m$^{-2}$ (Figure 4a). The total

net radiation is at a monthly maximum on 6 January (Figure 4b). SH flux is small but mostly positive between 5-9 January

(Figure 4c), resulting from warmer air just above the surface but this is largely cancelled by mostly negative LH flux so that

the net turbulent flux (Figure 4d) does not remain positive over more than one entire diurnal cycle between 5-9 January. The

$ME$ remains positive over two full diurnal cycles 6-7 January (Figure 4e), but at no other time in January. This corresponds

with a monthly maximum in 2 m air temperature and skin temperature (Figure 4f).

Cloud $LWP$ and $IWP$ (Figure 5) show discrepancies between ERA5 and CERES SYN1deg, but overall suggest the presence

of optically thin cloud cover. ERA5 predicts very low $LWP$ but an impulse of high $IWP$ on 6 January. This may be

unrealistic, as Silber et al. (2019) show that ERA5 often produces too much cloud ice water and too little cloud liquid water

over West Antarctica. In contrast, CERES data indicate low $IWP$ but an impulse of elevated $LWP$ that briefly reaches a

maximum of 49 g m$^{-2}$ on 6 January. Throughout 5-9 January, the CERES average $LWP$ is 21.2±13.7 g m$^{-2}$. We note that the

ERA5 radiative transfer algorithm uses the high $IWP$ values when computing the SW and LW fluxes in Figure 4a-b.

Examining the vertical profiles in cloud water content over 5-9 January, we find that maximum liquid water content occurs

mainly in the pressure range 850-950 hPa, while maximum ice water content occurs in the more vertically extensive pressure

range 700-850 hPa (figure not shown). Although the ERA5 $IWP$ values exceed 80 g m$^{-2}$ on 6 January, they are still likely to

manifest as an optically thin cloud in the radiative transfer calculation if the effective cloud particle size is in the range 40-50

µm observed for Antarctic clouds (e.g., Scott and Lubin, 2016). In this case, the cloud optical depth would most likely be

less than 5, as opposed to a liquid water cloud that, with effective droplet radius of order 7-10 µm, would have an optical

depth of order 10-15 and would therefore radiate in the LW as a blackbody at a temperature characteristic of the pressure

range 850-950 hPa. The higher and more vertically extensive range of the ERA5 cloud ice water content on 6 January also

signifies colder radiating temperature and therefore smaller LW flux emitted to the surface. This case study underscores the

need for improvement in mixed-phase cloud microphysics used in reanalysis models. Gilbert et al. (2020) have demonstrated

how surface SW and LW fluxes governing surface melt on the Larsen C Ice Shelf are sensitive to cloud vertical profile as

well as thermodynamic phase, and the same considerations apply to West Antarctica. We also note that CERES data show a

second impulse in cloud $LWP$ on 9 January. Being absent in the ERA5 cloud simulation, its effect does not appear in the

radiative fluxes in Figure 6a-b. However, it may help to explain the satellite $T_b$ signals of partial surface melt in the region

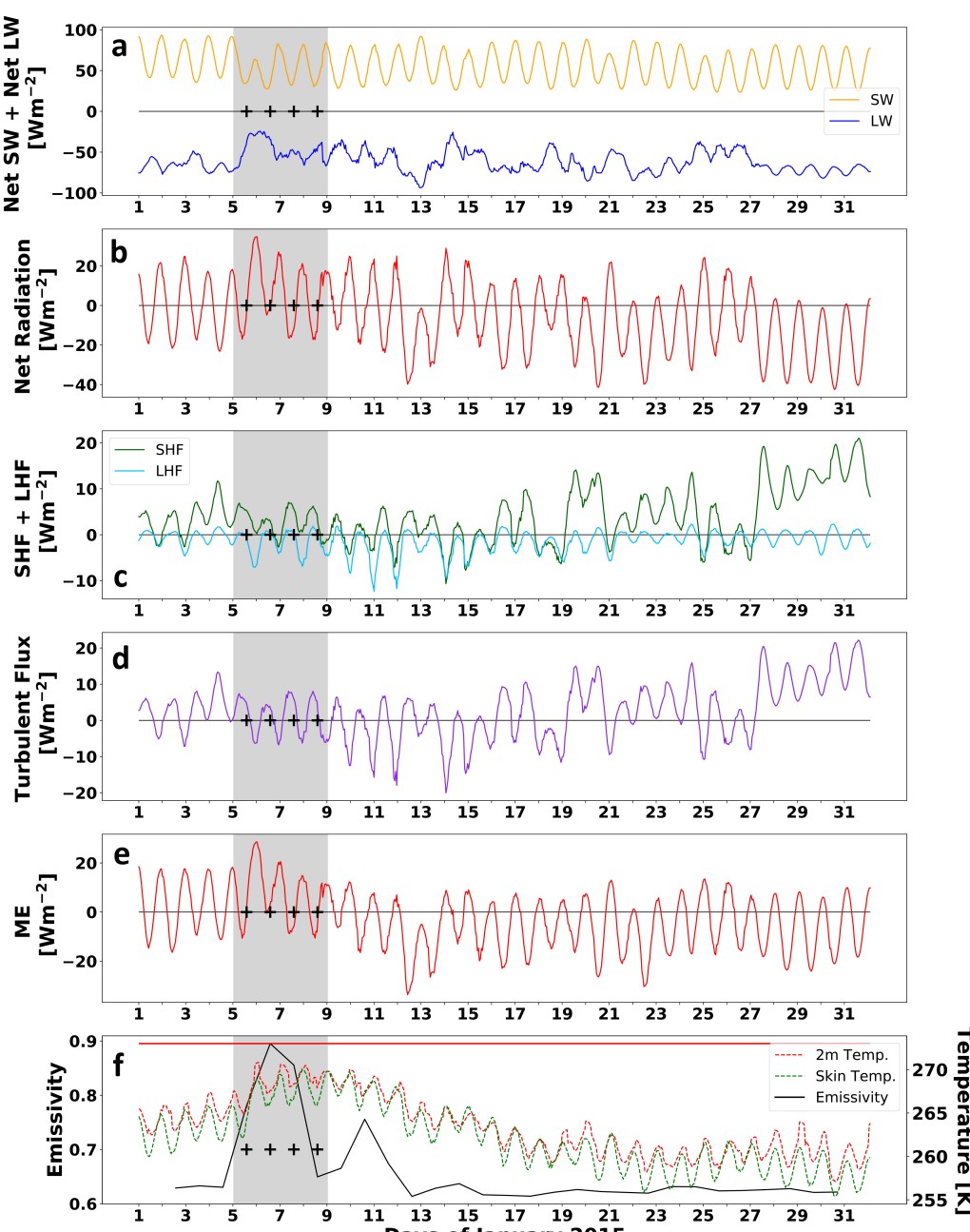

**Figure 4. Time series of the hourly *ME* components over the Siple Dome region throughout January 2015 from ERA5: (a) individual net SW and LW fluxes; (b) total net radiative flux (SW + LW); (c) individual SH and LW fluxes; (d) net turbulent flux (SH + LH); (e) total ME; (f) skin temperature (green), 2 m air temperature (red), and sampled 99th percentile emissivity from Figure 3b (black). The horizontal red line in (f) is at 273.15 K. The shaded region denotes the melt period of interest. Black crosses denote the satellite evening overpass times.**

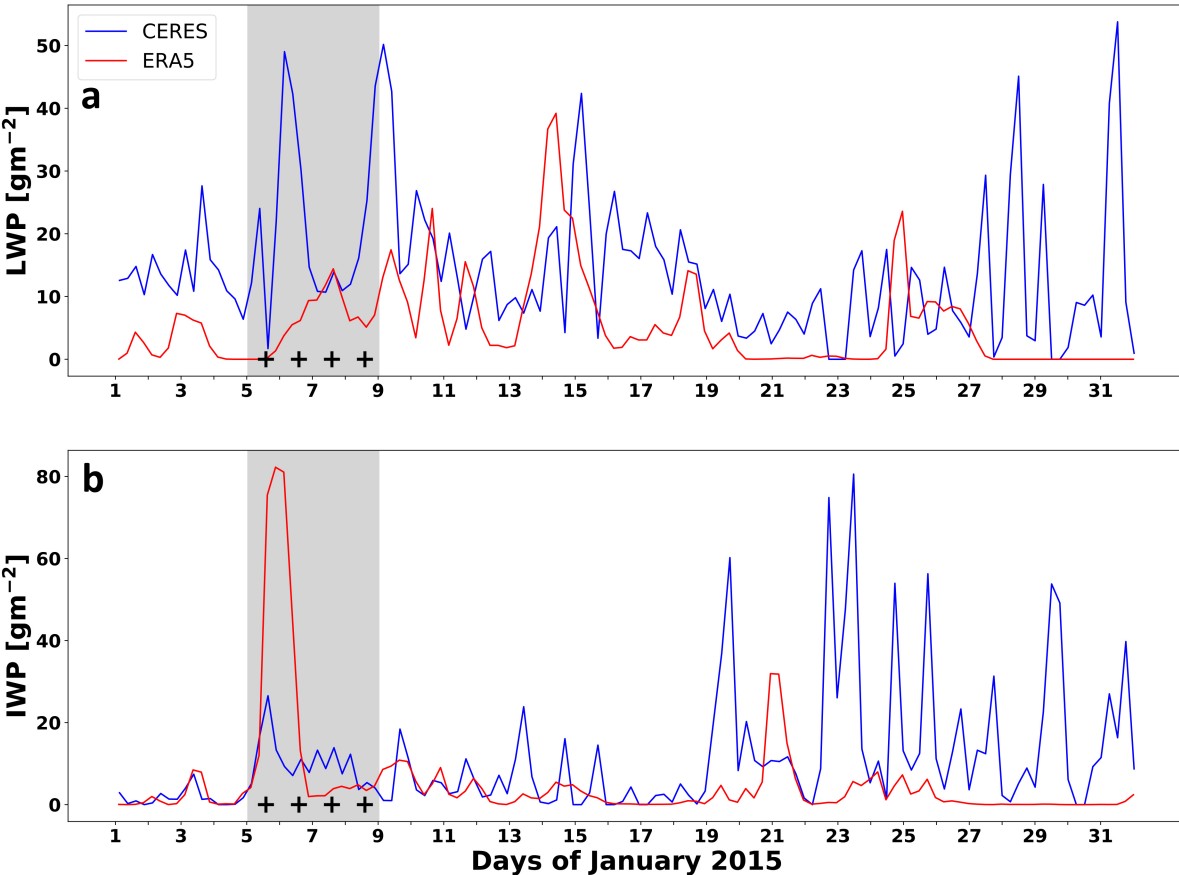

**Figure 5. Time series of hourly cloud *LWP* (a) and *IWP* (b) from the CERES SYN1deg data product (red) and ERA5 (blue), over the Siple Dome region throughout January 2015. Black crosses denote the satellite evening overpass times.**

that persist until 11 January (Figure 3). This may be an example of the refreezing inhibition proposed by Van Tricht et al. (2016).

Field camp observations between 5-9 January indicate mostly broken and overcast cloud cover with cloud bases between 900-1800 m and unrestricted visibility, occasionally dropping to ~250 m in reduced visibility with freezing fog or mist and light fog during 8-9 January. On 5-6 January the observer remarks "Sun dimly visible" through the overcast. These observations are qualitatively consistent with optically thin cloud cover.

### 3.2 Pine Island and Thwaites Glaciers January 2012

We next investigate a melt event that is clearly driven by clouds, during synoptic conditions that normally don't favour surface melt. Early January 2012 experienced strong positive Southern Annular Mode (SAM) conditions, as evidenced by the anomalously low geopotential heights over Antarctica in Figure 6. Such conditions, accompanied by strong circumpolar westerly flow, are associated with reductions in meridional heat exchange with lower latitudes. Therefore, this scenario is typically not conducive to melting on the ASE region (Scott et al. 2019). However, during the brief period of interest, a high-pressure ridge developed over the northern Amundsen Sea, off the tip of South America (not shown). This briefly diverted the large-scale flow toward the ASE region and provided an impulse of heat and moisture to the area.

Our period of interest shows a modest melt signal (Figure 7a), with the mean satellite PMW $T_b$ reaching the standard 30 K detection threshold only on 6 January. However, we note that throughout January 2012 the upper bound of the $T_b$ sample is near or slightly above the 30 K detection threshold. In Figure 7b we see that all the sampled surface emissivity estimates are larger than 0.8, in contrast to the lower values observed over Siple Dome outside of melt periods.

During our period of interest 4-8 January, the radiative fluxes show a strong modulation by cloud cover (Figure 8a), with net SW flux attenuated by nearly a factor of two relative to most of the rest of the month, and with net LW driven to nearly zero. The result is that the net total radiative flux remains positive across three diurnal cycles. The SH and LH fluxes (Figure 8c) are much smaller in amplitude and the net turbulent flux drops below zero every day (Figure 8d). It is primarily the radiative flux terms that keep the *ME* positive across nearly four diurnal cycles (Figure 8e). The corresponding 2m air and skin temperatures rise steadily during this interval to a monthly maximum on 7 January (Figure 8f), which is the second strongest day in the satellite melt detection signal (Figure 14a). We note that two other short periods, 16-17 January and 20-21 January, show *ME* > 0 across two diurnal cycles. However, the 2m air and skin temperatures are well below freezing during these periods, and satellite melt signatures are barely detectable (Figure 7). During 4-8 January the 2m air and skin temperatures approach freezing, which is generally necessary for melt onset even when the primary energy input is from a cloud radiative impulse.

The cloud *LWP* estimates (Figure 9a) show consistency between ERA5 and CERES during 4-8 January, although ERA5 appears to underpredict *LWP* for most of the rest of the month. ERA5 again appears to overpredict *IWP* during the melt period of interest, by more than a factor of two compared with CERES. During 4-8 January the CERES *LWP* is mostly within the thin cloud range (10-40 g m$^{-2}$) associated with the Bennartz et al. (2013) all-wave radiative effect. CERES *IWP* is almost as large as the *LWP*, which again may reflect errors in MODIS-based phase discrimination. Considering the CERES combined *LWP* and *IWP*, it remains unclear if the cloud radiative impulse (Figure 8a,b) is due to the Bennartz et al. (2013) all-wave effect or to thermal blanketing by optically thicker cloud cover. And the ERA5 radiative transfer algorithm

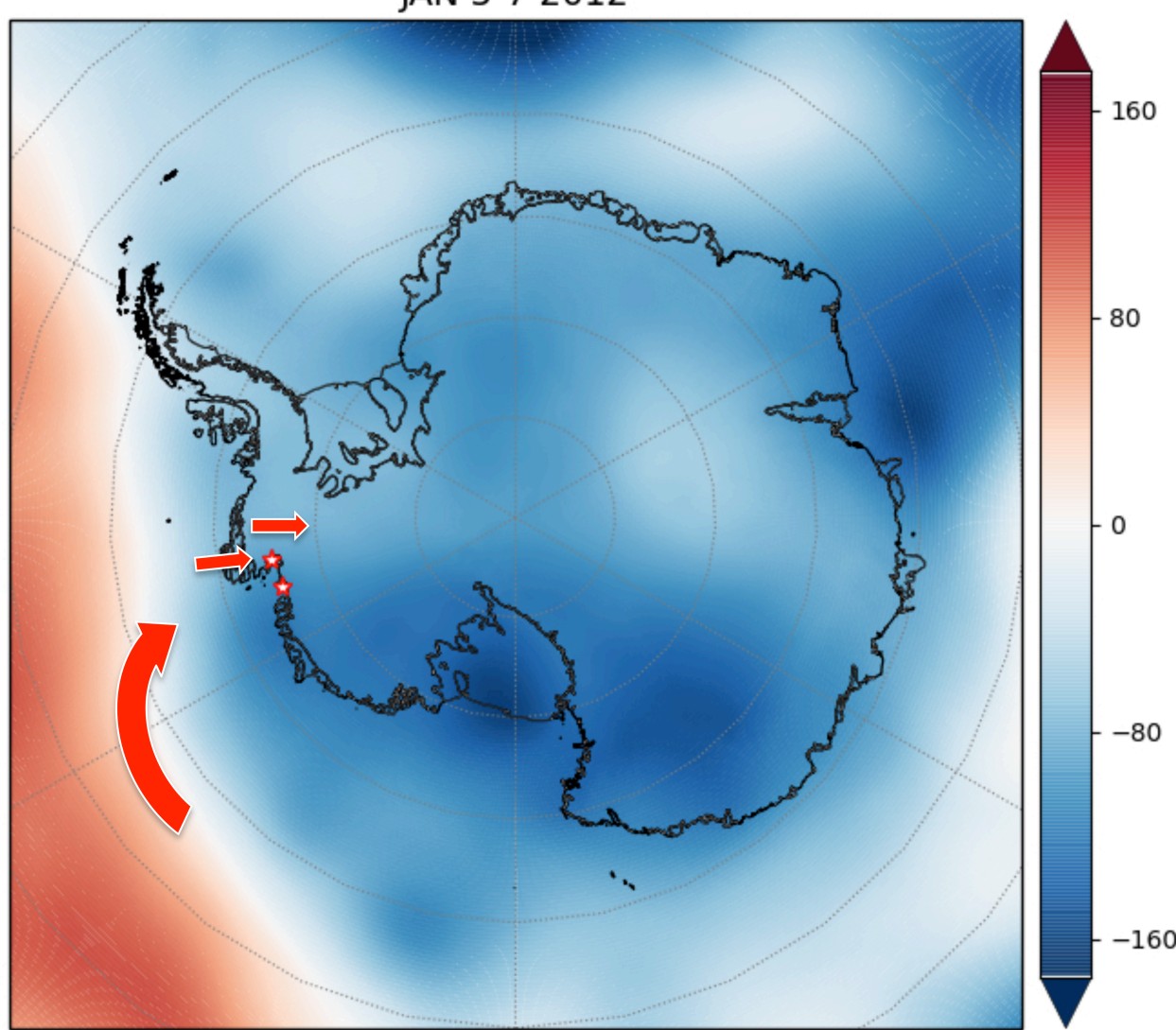

JAN 5-7 2012

$Z_{700}$ Composite Anomaly - meters

**Figure 6. Composite anomaly of ERA5 700 hPa geopotential height for 5-7 January 2012. The red stars depict the locations of Pine Island and Thwaites Glaciers. The red arrows depict the direction of 700 hPa winds relevant to the case study.**

produces the fluxes in Figure 8a using the large cloud *IWP* values that are almost certainly in error. This case study clearly shows the role of clouds in altering the *ME* to enhance surface melt, but also underscores the need to improve both satellite retrieval and reanalysis cloud microphysics to obtain a complete understanding.

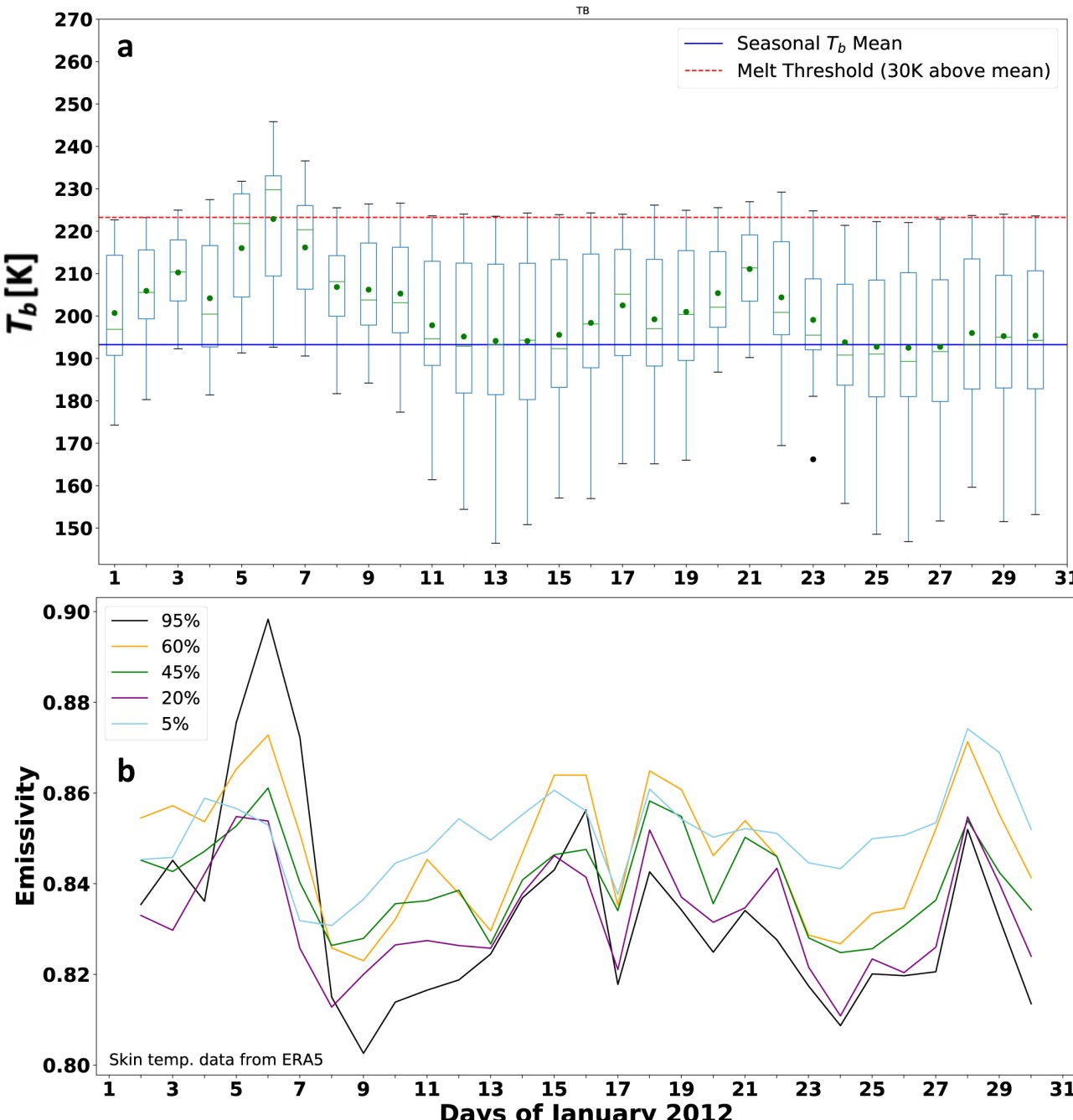

**Figure 7. As in Figure 3, but over the Pine Island and Thwaites Glaciers region throughout January 2012. The five estimates of surface emissivity in (b) sampled from the region are referenced to the maximum $T_b$ value in the region on 6 January.**


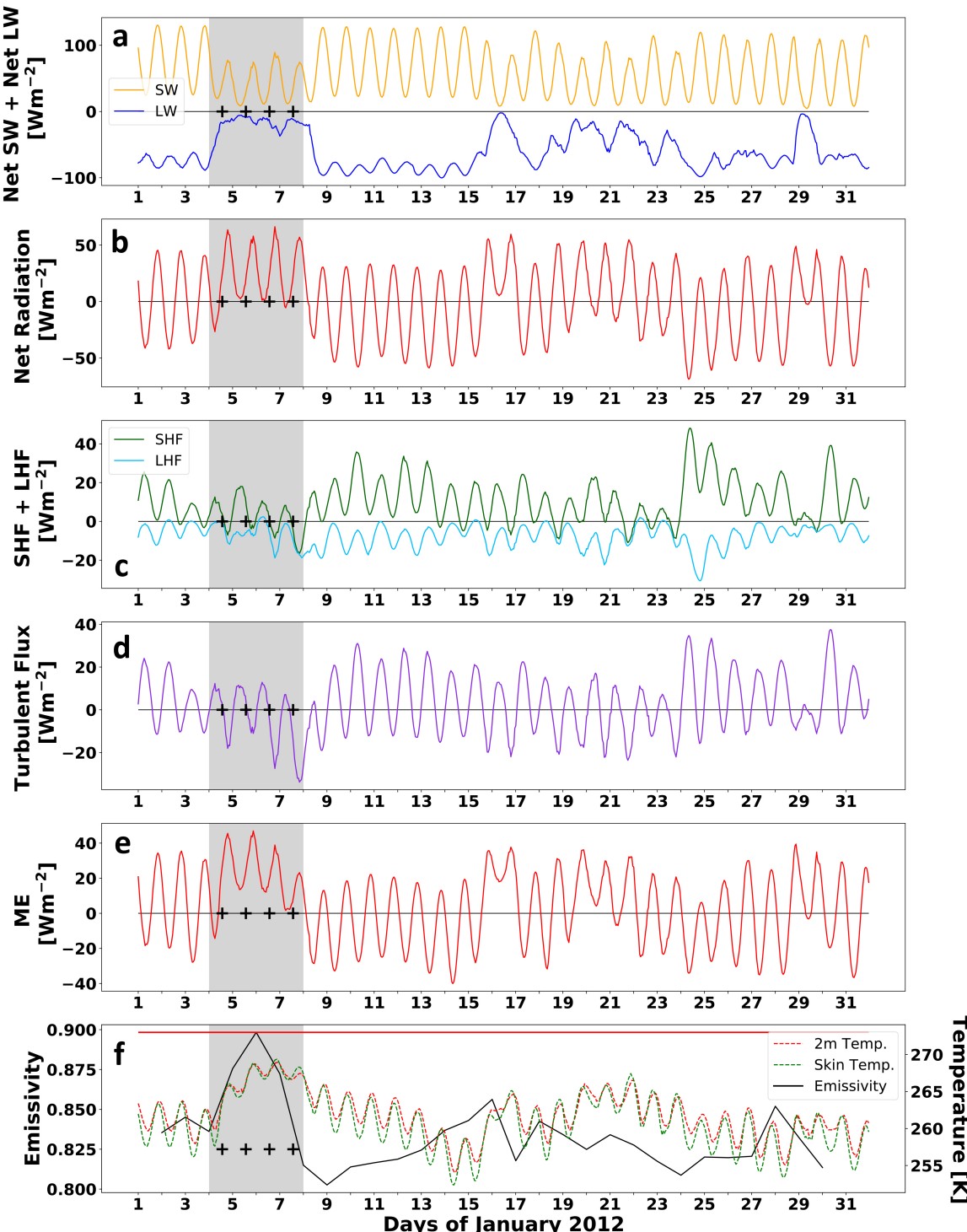

**Figure 8.** As in Figure 4, but over the Pine Island and Thwaites Glaciers region throughout January 2012.

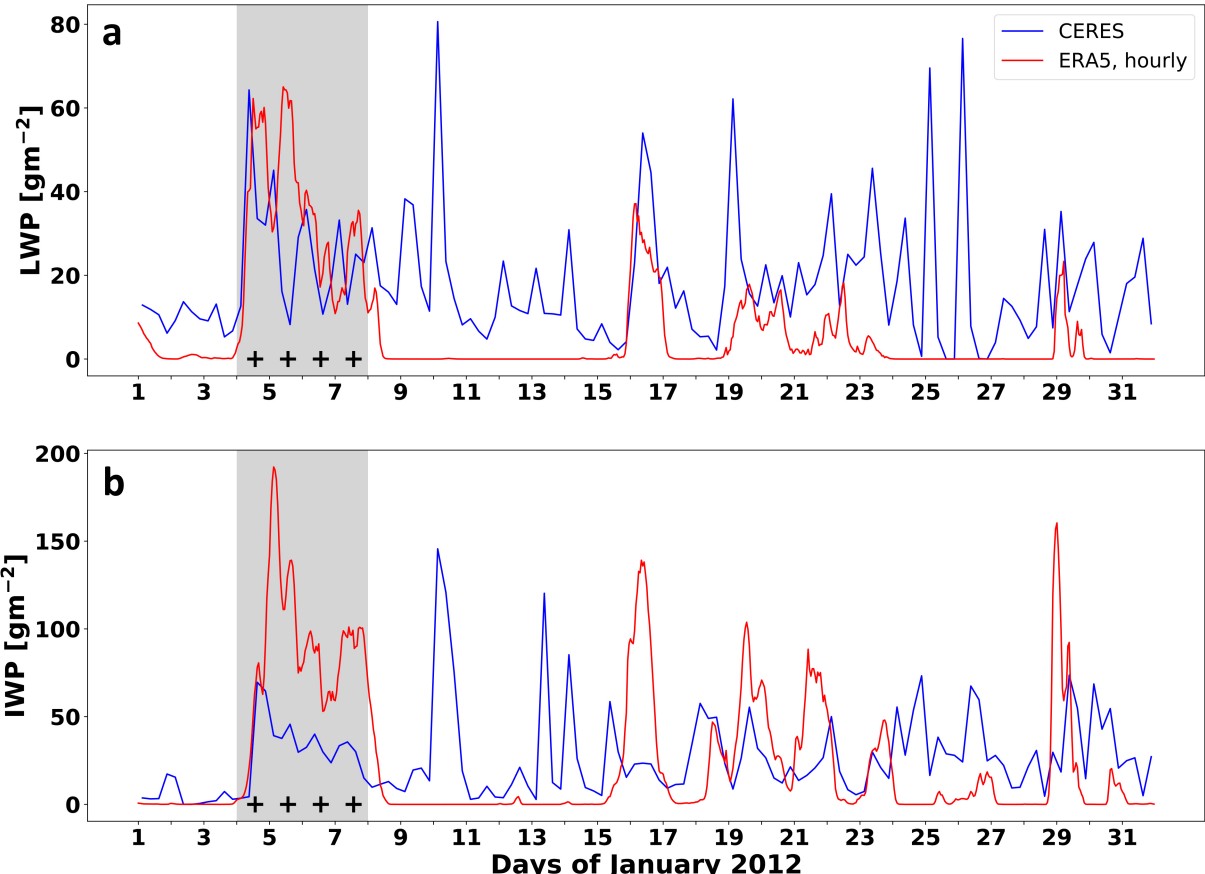

**Figure 9. As in Figure 5, but over the Pine Island and Thwaites Glaciers region throughout January 2012.**


The field camp on Pine Island Glacier recorded broken to overcast cloud cover with bases 300-600 m on 4 January, with ceilings dropping to 150 m on 5 January. On 6-7 January at least two cloud layers were observed, with variable ceilings mostly below 2000 m. Throughout 8 January sky coverage steadily reduces from broken to scattered/few. Light snowfall is the most consistent present-weather condition between 4-8 January, but there are also episodes of mist, freezing fog, drifting

snow and blowing snow. Qualitatively these observations might suggest optically thicker cloud cover.

### 3.3 Pine Island and Thwaites Glaciers February 2013

We now examine a late summer melt event driven by thermal blanketing on Pine Island and Thwaites Glaciers in February, when climatological surface and lower tropospheric temperatures are typically several degrees cooler than in January. During late February 2013, an amplified ridge of high pressure developed and remained stationary over the Amundsen-

Bellingshausen Seas (Figure 10). At the same time, a low-pressure system formed and deepened over the Ross Sea. This

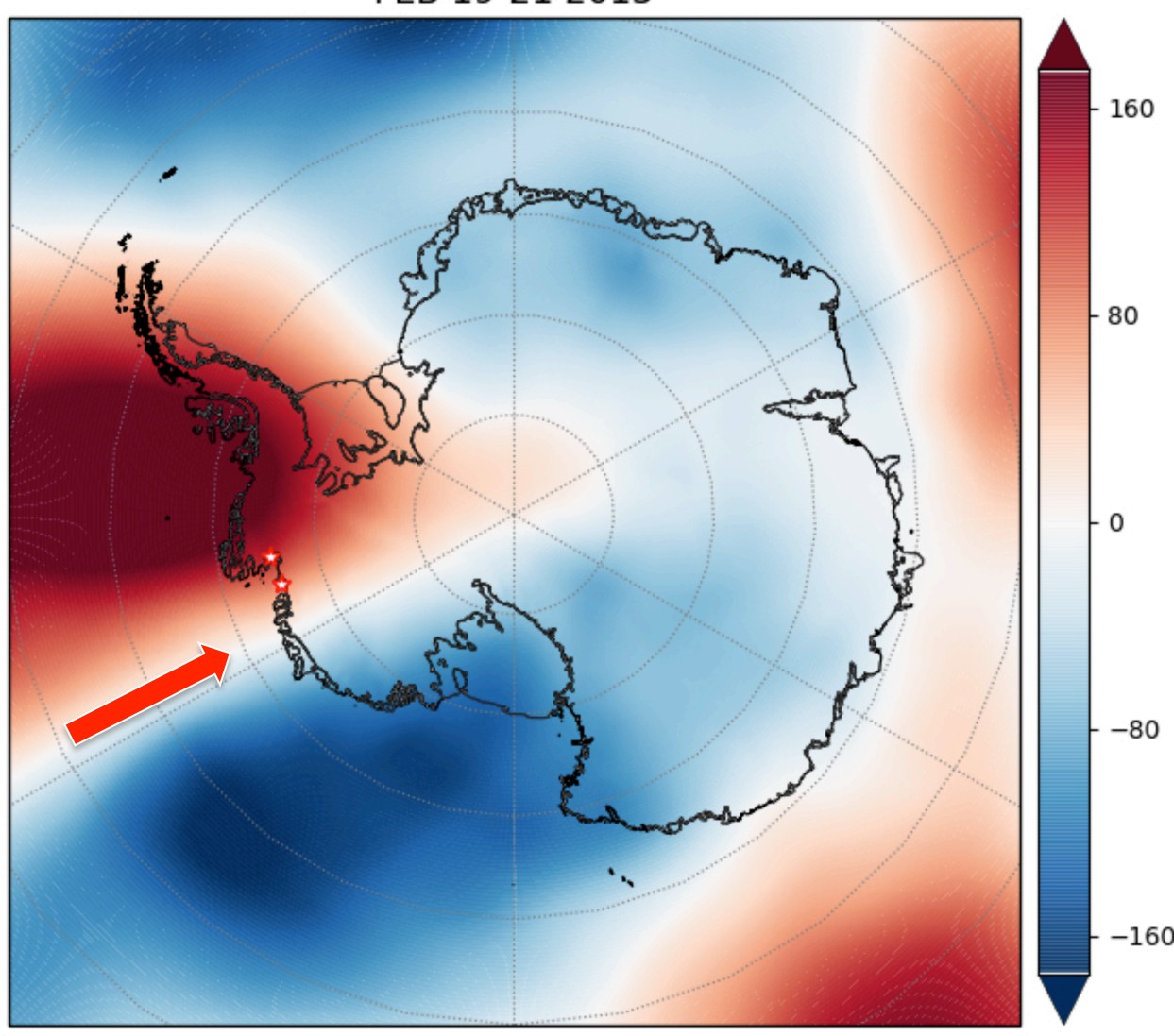

**FEB 19-21 2013**

$Z_{700}$ Composite Anomaly - meters

**Figure 10. Composite anomaly of ERA5 700 hPa geopotential height for 19-21 February 2013**. **The red stars depict the locations of Pine Island and Thwaites Glaciers. The red arrow depicts the direction of 700 hPa winds relevant to the case study.**


resulted in strong and sustained meridional flow of heat and moisture into West Antarctica, which lasted for 5 days. Such synoptic conditions are highly conducive to surface melting along the West Antarctic coastline and were likely critical for

causing the observed late-summer melt. This synoptic pattern is a signature of the Amundsen Sea Low (Turner et al., 2013; Clem et al. 2017), and is representative of frequent surface melting in the area (Scott et al., 2019).


In Figure 11a, satellite PMW data show a three-day, partial-surface-melt signature in the Thwaites and Pine Island Glaciers region from 20-22 February 2013. Surface emissivity (Figure 11b) has relatively large spatial variability throughout this local region. For our melt period of interest between 19-21 February, the radiative fluxes (Figure 12a) show a clear signature of thermal blanketing by optically thick cloud cover. The net SW flux is attenuated by a factor of three compared with the

earlier weeks in February, such that its diurnal amplitude is only ~20 W m$^{-2}$. The LW flux is positive, signifying optically thick clouds that are warmer than the surface. The net radiative flux (Figure 12b) is positive over the diurnal cycles 20-21 February. We also find positive SH flux (Figure 12c) that yields positive net turbulent flux (Figure 12d) across the entire melt period of interest. This positive turbulent flux is comparable in magnitude if not greater than the net radiative flux between 19-21 February. Then between 21-23 February, as the cloud radiative effect diminishes such that the net radiation

drops below zero each day, the SH flux doubles in magnitude to sustain the positive *ME* until 23 February (Figure 12e). The result is a steady rise in 2m air and skin temperatures from 20 February, when the satellite melt signature is first detected, to nearly the freezing point by 21 February and staying this warm for another four days. Even though these temperatures remain close to the freezing point for several days, the satellite melt signature decreases as the *ME* decreases and resumes a diurnal cycle that drops below zero.


The cloud properties during this melt period (Figure 13) are mainly consistent with large optical thickness. The CERES average *LWP* and *IWP* are 34.9±25.8 g m$^{-2}$ and 47.8±27.4 g m-$^{-2}$, respectively. While this larger *IWP* may reflect errors in phase discrimination, the suggested total cloud water content is higher than that associated the Bennartz et al. (2013) all-wave effect, and instead indicates primarily a longwave surface warming where a low cloud radiates as a blackbody, with a

muted SW diurnal signal. ERA5 *LWP* and *IWP* are significantly larger than the CERES retrievals, and may be overestimated due to microphysical errors, but their timing is consistent with the CERES detection of optically thick clouds. In this case study, we therefore see a thermal blanketing effect that is initiated in the first two days by a cloud radiative warming, and then sustained for another two days by elevated SH flux.


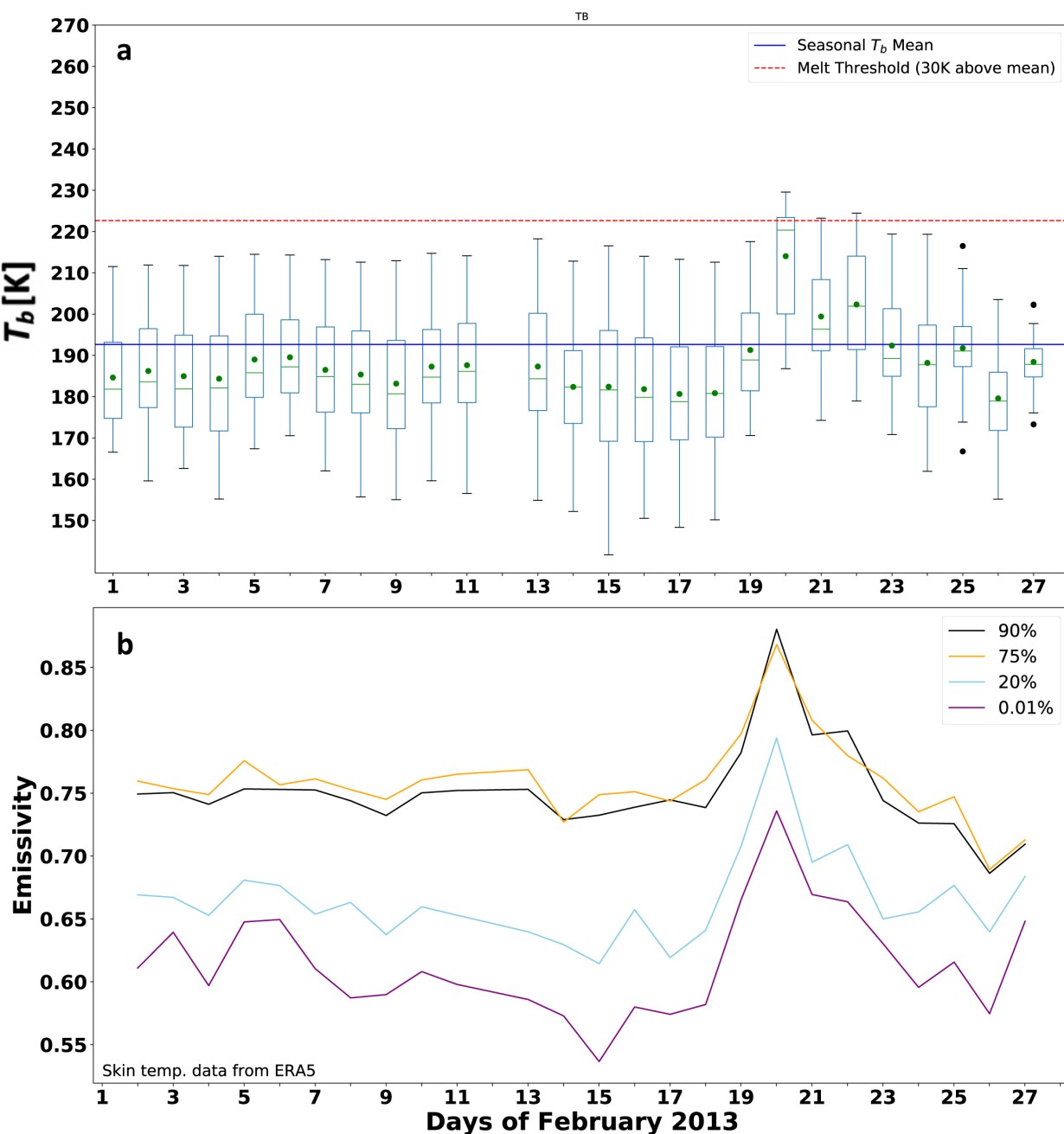

**Figure 11.** As in Figure 3, but over the Pine Island and Thwaites Glaciers region throughout February 2013. The five estimates of surface emissivity sampled in (b) from the region are referenced to the maximum $T_b$ value in the region on 20 February.

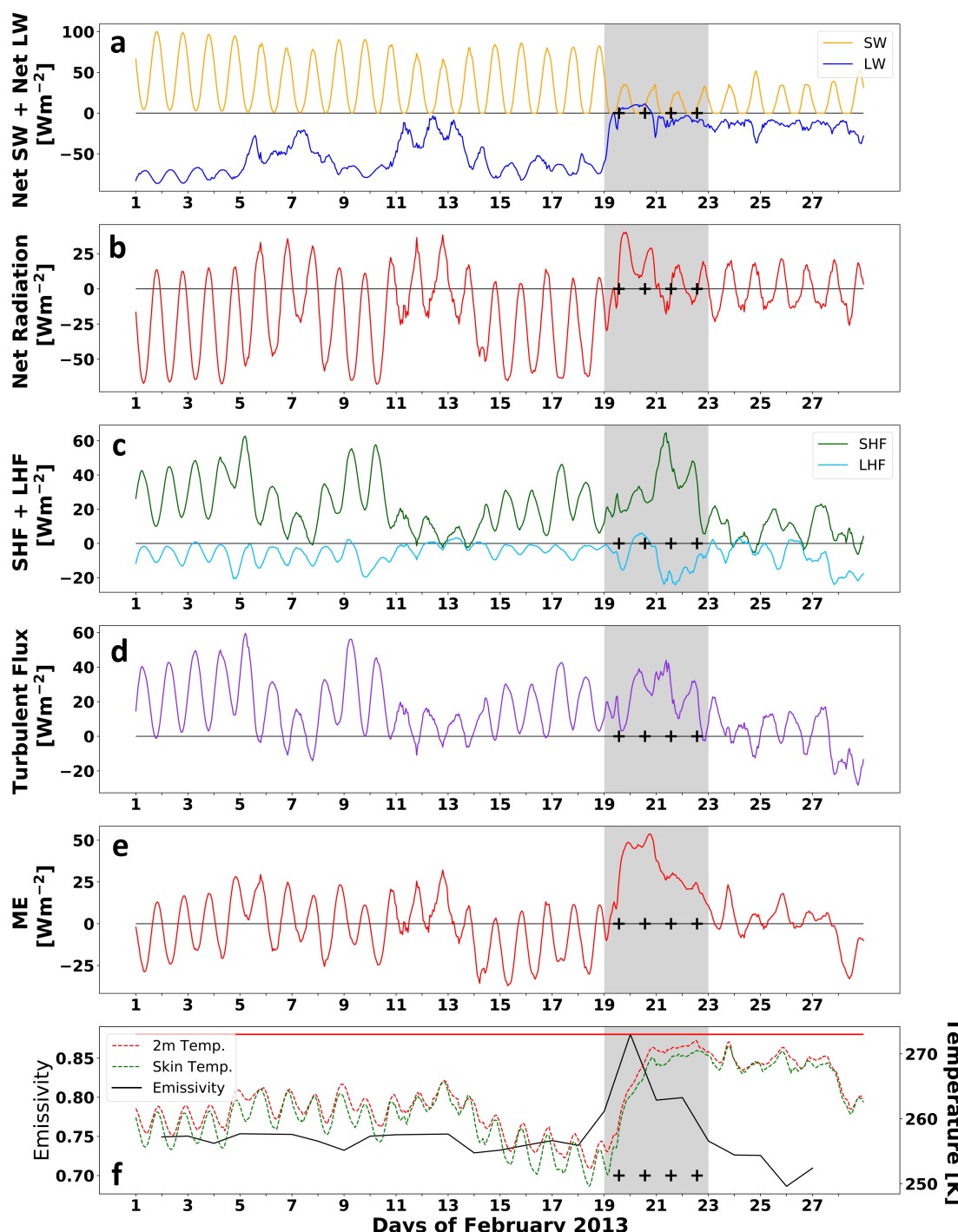

**Figure 12. As in Figure 4, but over the Pine Island and Thwaites Glaciers region throughout February 2013.**

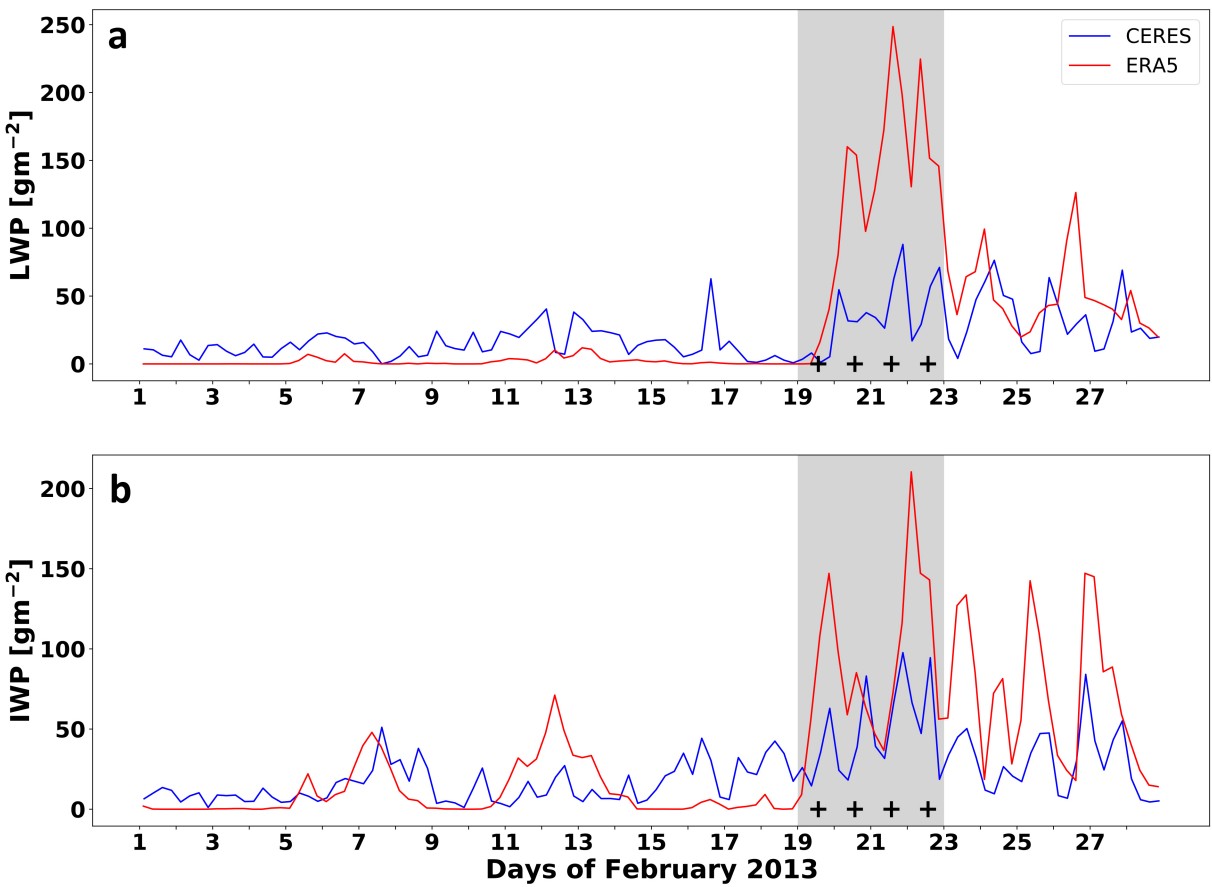

**Figure 13. As in Figure 5, but over the Pine Island and Thwaites Glaciers region throughout February 2013.**

### 3.4 West Antarctica and Ross Ice Shelf December 2011

We now consider a meteorological event that triggered surface melting at all three regions considered in this study. In late December 2011, a low-pressure system (Figure 14) propagated eastward over the Ross Sea as a ridge of high pressure built over the Amundsen-Bellingshausen Seas. This conjunction favoured an intrusion of a warm, moist airmass far into the interior of West Antarctica. The marine air intrusion maintained an optically thick liquid cloud presence over the ASE region, where surface melting first began and signatures of surface snow melt persisted for several days. After crossing over the WAIS, the airmass then descended onto the Ross Ice Shelf, producing widespread föhn effects over Siple Coast. Relatively weak melt signatures were observed at Siple Dome. Föhn warming was most pronounced parallel to the Transantarctic Mountains in

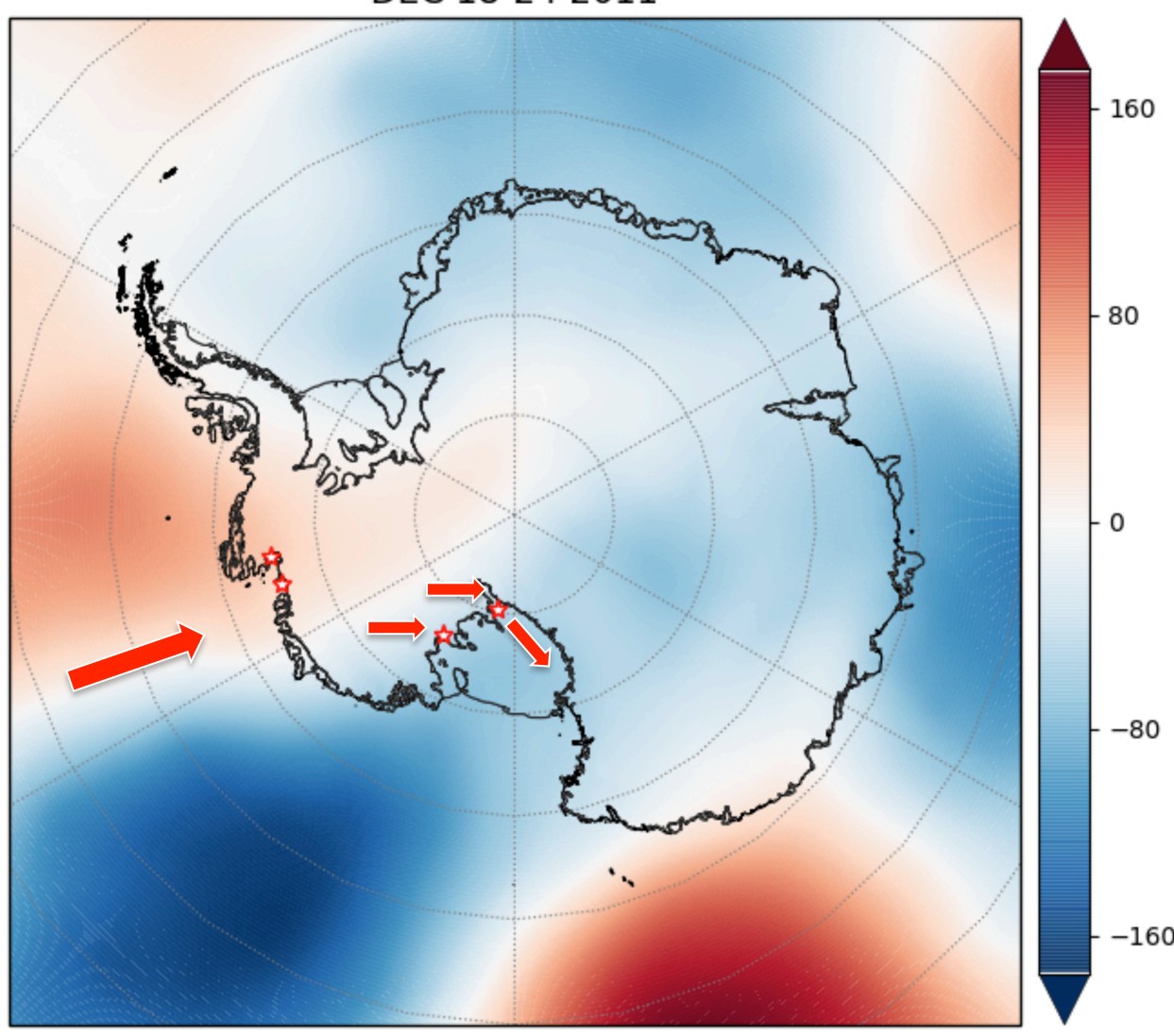

**Figure 14. Composite anomaly of ERA 700 hPa geopotential height for 18-24 December 2011**. The red stars depict the locations of all three case study locations. The red arrows depict the direction of 700 hPa winds relevant to the case study.

association with a summertime Ross Ice Shelf airstream event. This gave rise to melting at the Tom and Sabrina automatic weather stations.

At Pine Island and Thwaites Glaciers the melt period of interest is between 19-25 December (Figure 15), and on 20-21 December most satellite $T_b$ measurements are consistent with unambiguous surface melt (also see Appendix B). Examining the surface emissivity samples (Figure 15b) we see considerable spatial variability throughout the month. Between 2-18 December some of the grid cells show surface emissivity in the "dry snow" range (<0.80), while others are in a range (>0.80) that may signify wet or otherwise altered firn (e.g., Mätzler, 1987). We notice in Figure 15a that the top of the $T_b$ range in all days between 1-18 December is near or slightly above the standard 30 K melt detection threshold. In Figure 15b the sampled percentiles are referenced to the maximum $T_b$ on 21 December. We notice that the sampled grid cells reaching the 75th and 99th percentiles had very low surface emissivity earlier in the month. Figure 15b therefore illustrates complexity in local-scale surface properties at these low elevation locations near the coast. This complexity might arise from repeated melting and re-freezing episodes, combined with more intense episodes of precipitation, as well as varying topography especially near Pine Island and Thwaites Glaciers.

Over Siple Dome this synoptic condition led to several satellite $T_b$ measurements in the > 30-K threshold melt detection range between 22-26 December 2011 (Figure 16), a less pronounced melt signature than in January 2015 but nevertheless detectable. In Figure 16b, we again see spatial uniformity in sampled surface emissivity throughout the prior three weeks; then during the melt period of interest, some surface emissivity values remain low and within the "dry surface" range (e.g., Mätzler, 1987) while others become elevated by as much as 0.16. At our location on the RIS adjacent to the Transantarctic mountains (Figure 1), Figure 17a indicates that between 23-25 December, some grid cells show a strong satellite PMW melt signature, and a few continue to show a melt signature as late as 27 December. Similar to Siple Dome, surface emissivity is spatially uniform and consistent with a dry snow surface throughout the previous three weeks of December (Figure 17b).

Examining the *ME* components at these three locations reveals contrasting mechanisms for inducing and sustaining surface melt. At Pine Island and Thwaites Glaciers during 19-25 December the radiative fluxes (Figure 18a,b) are consistent with optically thick clouds attenuating the SW flux and driving the new LW flux to nearly zero, particularly on 20-21 December and 23-24 December. SH flux (Figure 18c) is small but positive over two diurnal cycles 19-20 December, but this is partially offset by negative LH fluxes, such that the net turbulent flux (Figure 18c) drops below zero every day between 19-25 December (and throughout the month). The total *ME* > 0 across the diurnal cycles 20-22 December and 23-24 December, mainly due to the impact of cloud cover on the radiative fluxes (Figure 18e). This induces a steadily rising 2m air and skin temperatures (Figure 18f), with corresponding rise in the fraction of grid cells showing satellite melt detection signatures (Figure 15a).

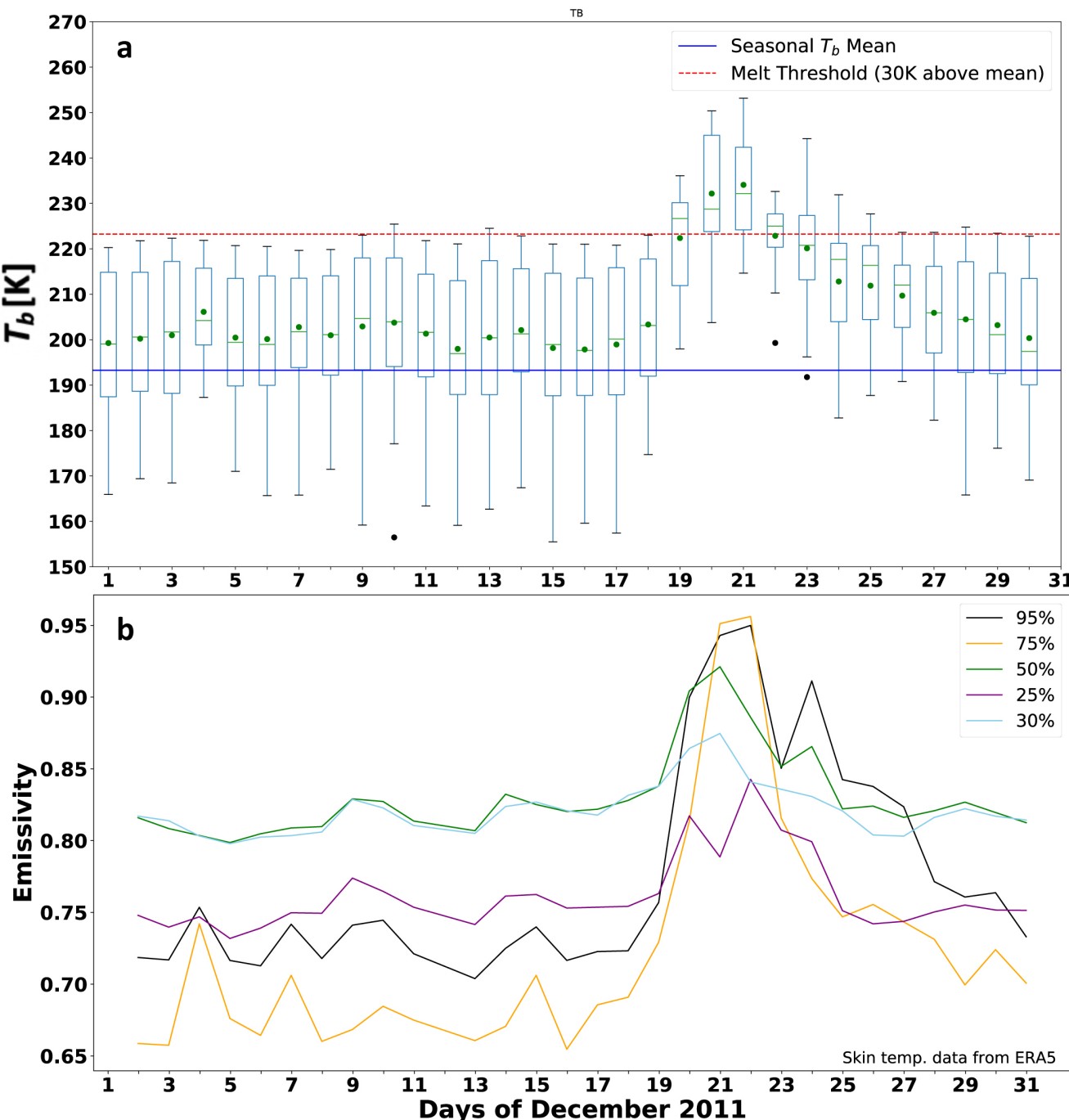

**Figure 15. As in Figure 3, but over the Pine Island and Thwaites Glaciers region throughout December 2011. The five estimates of surface emissivity in (b) sampled from the region are referenced to the maximum $T_b$ value in the region on 21 December.**

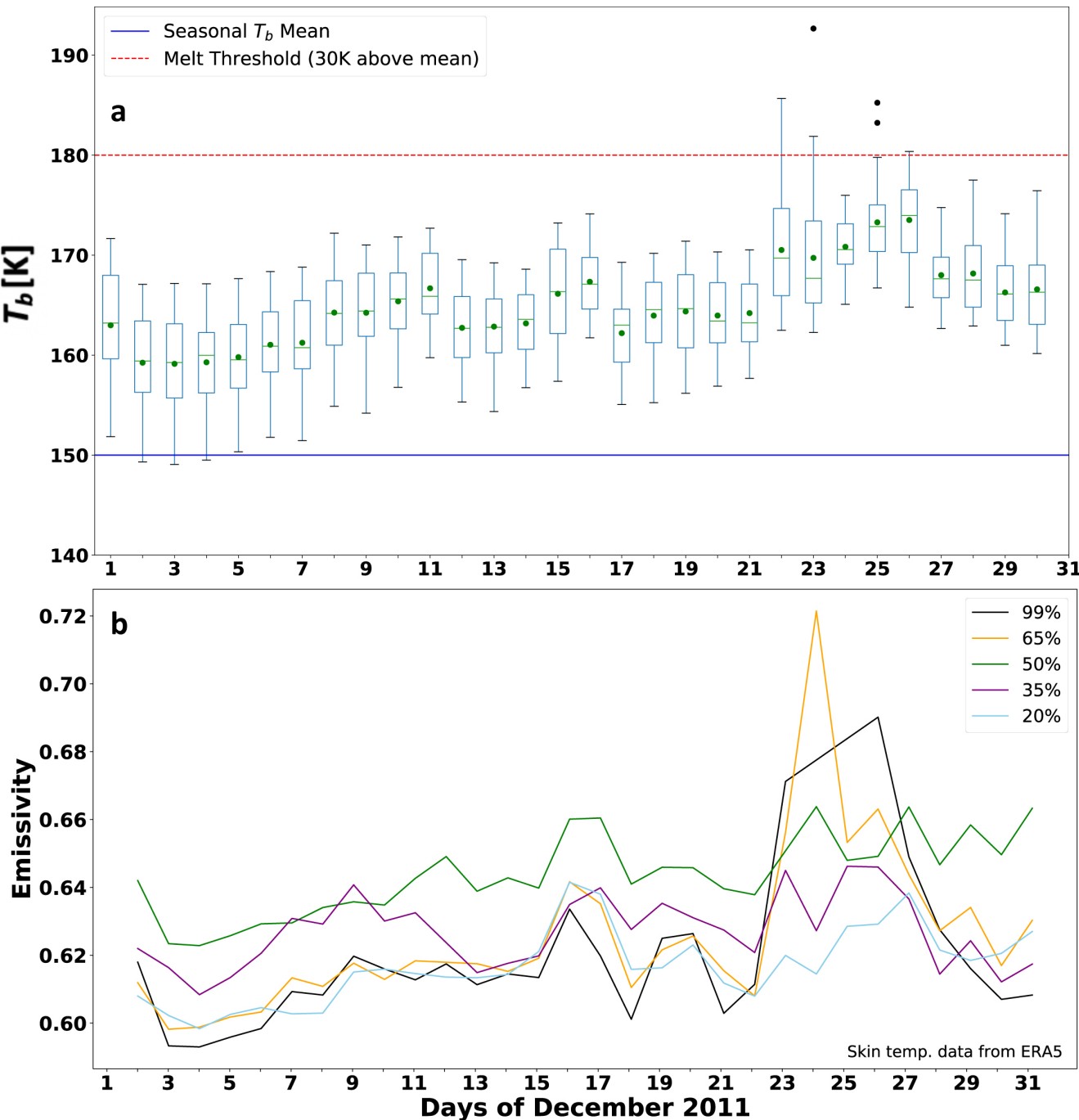

**Figure 16.** As in Figure 3, but over the Siple Dome region throughout December 2011. The five estimates of surface emissivity in (b) sampled from the region are referenced to the maximum $T_b$ value in the region on 23 December.

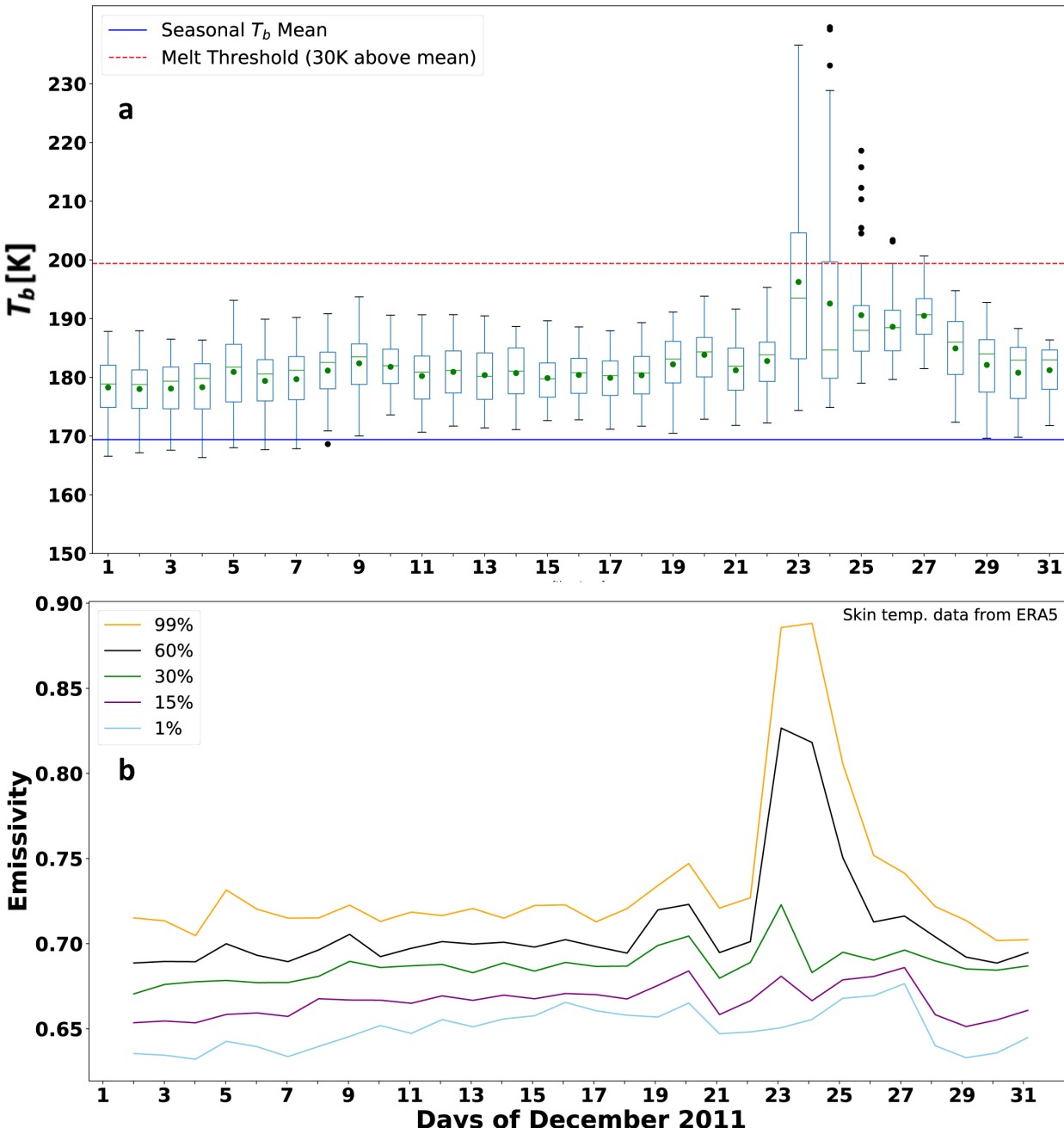

Figure 17. As in Figure 3, but over the RIS region containing the Tom and Sabrina AWS, throughout December 2011. The five estimates of surface emissivity sampled in (b) from the region are referenced to the maximum $T_b$ value in the region on 23 December.

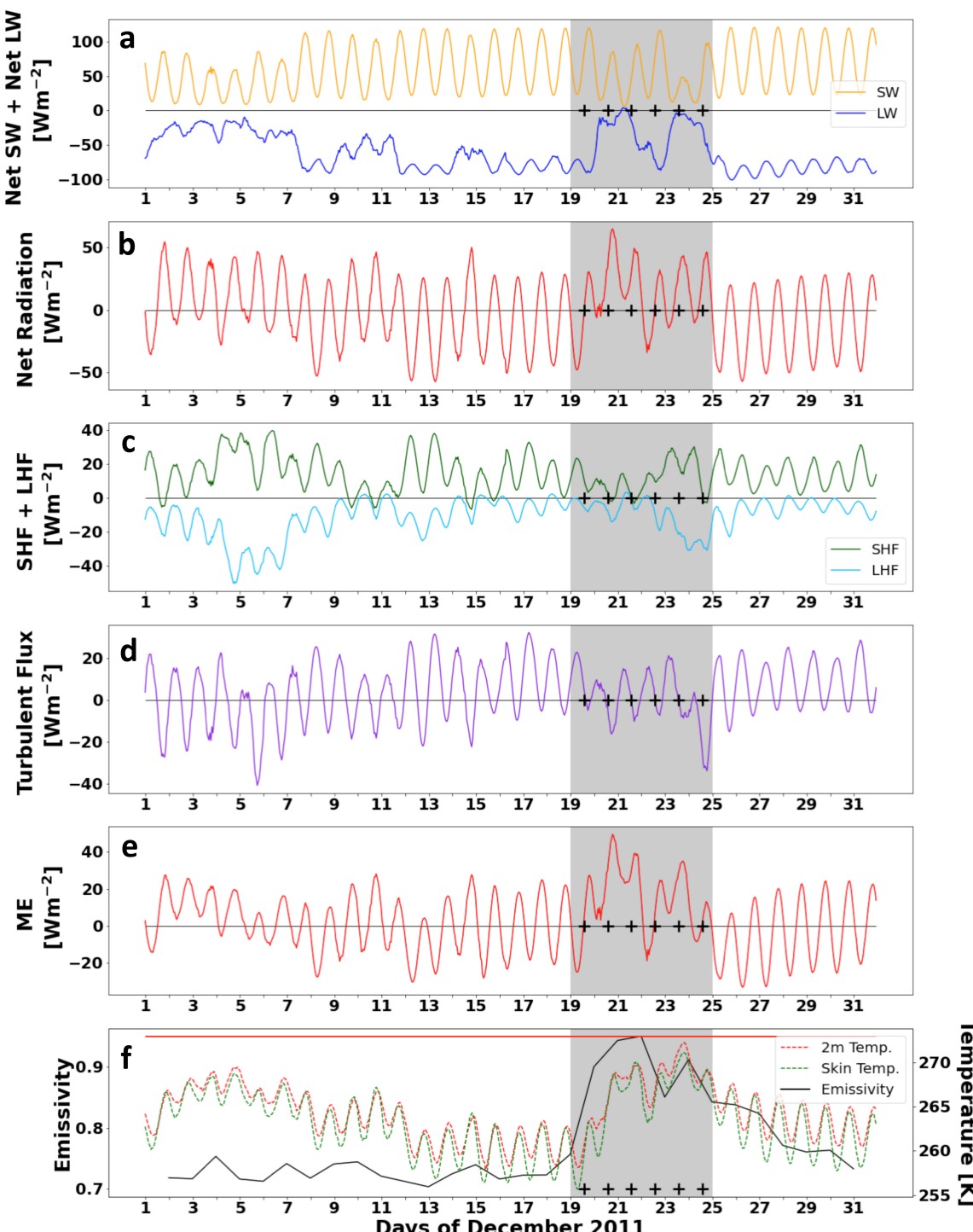

**Figure 18. As in Figure 4, but over the Pine Island and Thwaites Glaciers region throughout December 2011.**

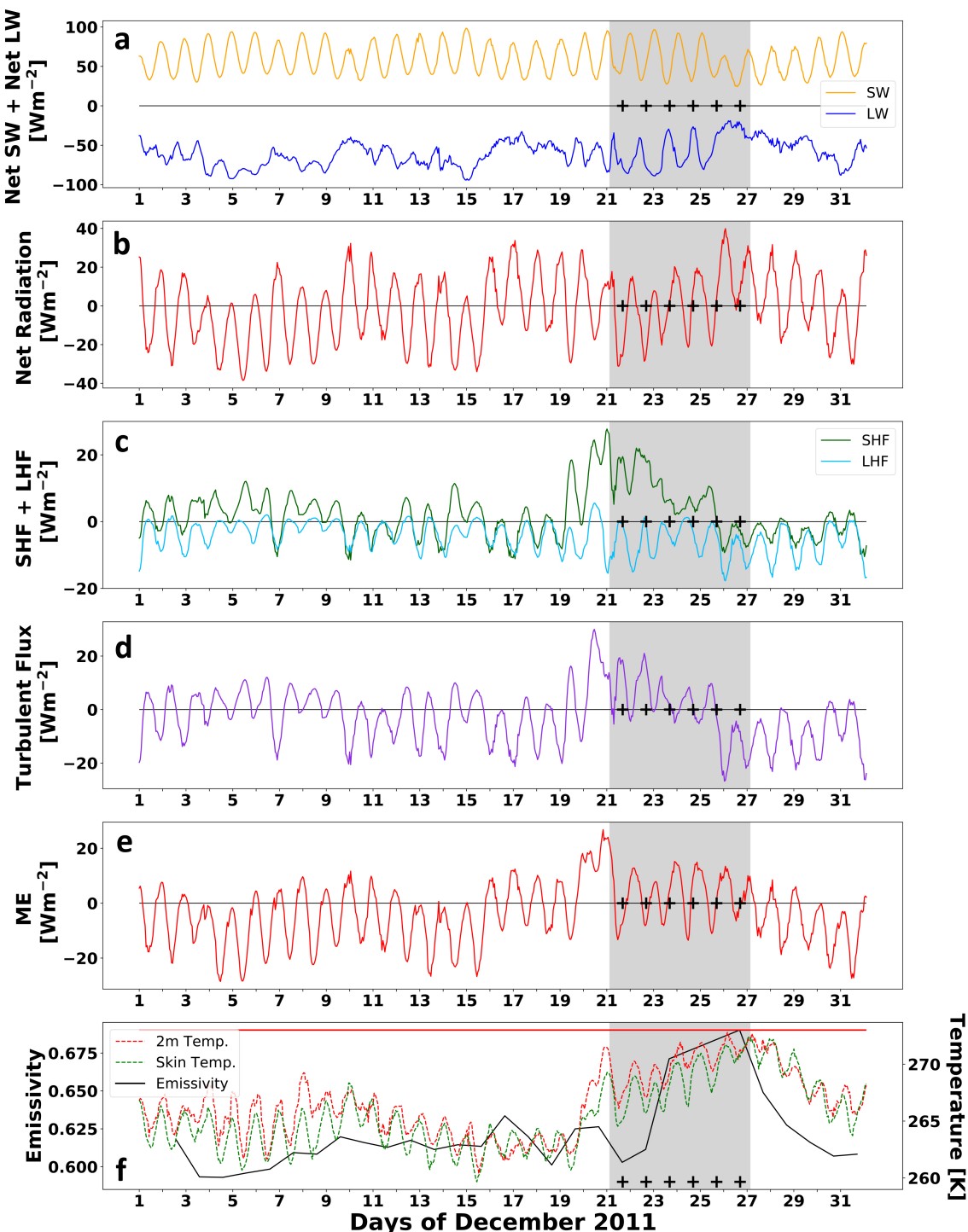

**Figure 19. As in Figure 4, but over the Siple Dome region throughout December 2011.**

Examining the SEB components at Siple Dome, we see that cloud radiative effects (Figure 19a,b) do not substantially alter the SEB until late in the melt period of interest (22-26 December). This melt event instead appears to be induced and dominated by an impulse of SH flux that begins on 19 December (Figure 19c), associated with the warm air intrusion, and causes the net turbulent flux (Figure 19d) and the total *ME* (Figure 19e) to remain positive through two diurnal cycles before the satellite PMW data show signs of surface melt. During the satellite melt detection period, the *ME* actually drops below zero at the lowest Sun elevations, even as the ERA5 2m air and skin temperatures rise steadily (Figure 19f).

At the RIS location the relevant energy inputs appear to precede the satellite melt signature detection by approximately two days (similar to Siple Dome). For the SEB components (Figure 20) we have identified the study period of interest as 19-23 December, while the satellite melt signature occurs mainly on 23 December and later. The radiative fluxes during 20-22 December (Figure 20a) show net SW attenuation and LW increase toward zero that appear consistent with all-wave enhancement from optically thin cloud. During this interval the net radiative flux is mostly positive, but does briefly drop to zero each diurnal cycle (Figure 20b). A strong impulse of positive SH flux (Figure 20c) is partly cancelled by a LH flux of opposite sign, but the net turbulent flux is positive across two diurnal cycles 20-21 December (Figure 20d), as is the total *ME* (Figure 20e). This signature, positive SH and negative LH fluxes, is frequently indicative of föhn wind conditions (e.g., Kuipers Munneke, 2012b; 2018; Datta et al., 2019; Elvidge et al., 2020). The maximum in *ME* on 21 December corresponds with a local maximum in 2m air and skin temperatures (Figure 20f), which increased by nearly 10K until they are close to freezing. The ERA5 daily maximum in 2-m air temperature continues to rise to above freezing on the 24[th] and peaking on the 25[th], before returning to sub-zero temperatures.

The cloud properties at Pine Island and Thwaites Glaciers (Figure 21) show impulses of high *LWP* and *IWP* simultaneously detected in CERES remote sensing data and simulated by ERA5. The LWP simulated by ERA5 is twice as large as that retrieved by CERES, and the radiative transfer model providing the fluxes in Figures 18a,b responds to this high *LWP*. The *IWP* is consistent between ERA5 and CERES, but we note that both could be artefacts: the ERA5 values might be an overestimate per Silber et al. (2019), and the CERES retrievals could also be an overestimate based on occasional difficulties in phase discrimination when using MODIS spectral reflectances (e.g., Platnick et al., 2017). Nevertheless, the information within the melt period of interest in Figure 13, specifically the total cloud water path (liquid plus ice), is highly consistent with optically thick clouds that provide most of the thermal blanketing effect in this case study. A field camp on Pine Island Glacier recorded mostly few and scattered clouds between 20-27 December. The timing of the two periods of increased sky coverage is consistent with the maxima in *LWP* and *IWP* of Figure 13. Late on 20 December and early on 21 December, the sky became broken to overcast with cloud base 1800 m. During 24 December the visibility dropped to 100-800 m in freezing fog and blowing snow. These observations do not definitively indicate optically thick clouds, and it is possible that this specific field camp location had lighter cloud cover than average for the entire region considered in this case study.

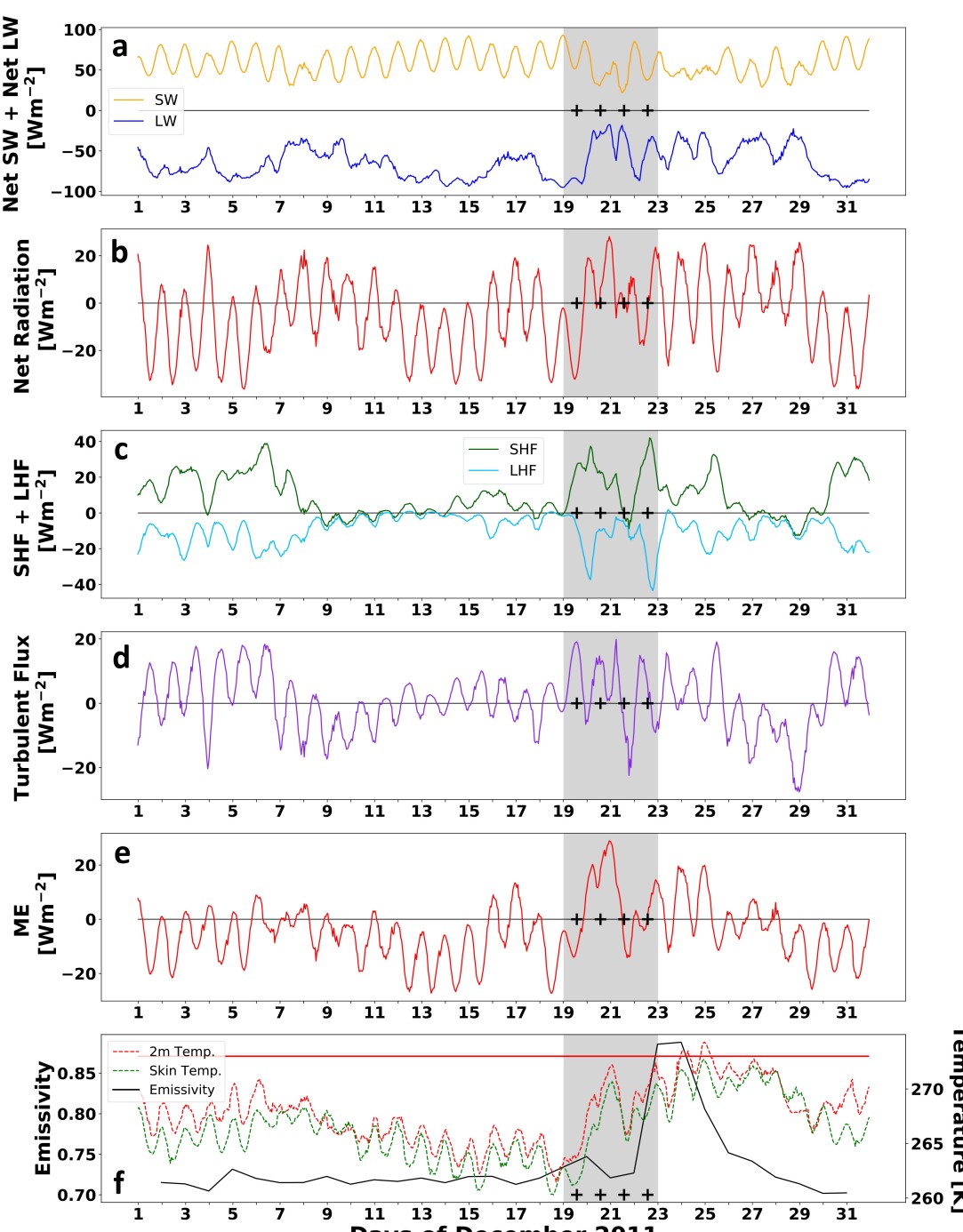

**Figure 20. As in Figure 4, but over the RIS region containing the Tom and Sabrina AWS, throughout December 2011.**

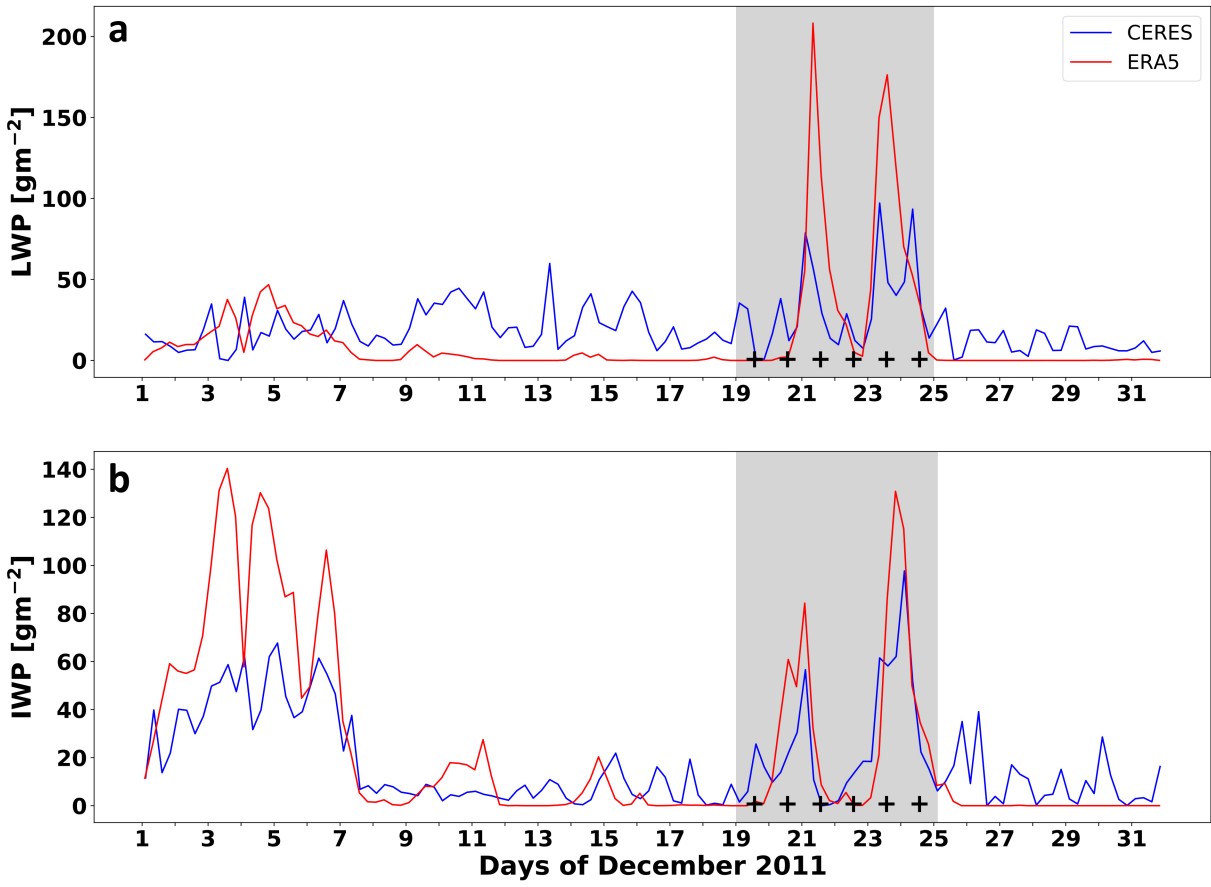

**Figure 21. As in Figure 5, but over the Pine Island and Thwaites Glaciers region throughout December 2011.**

At Siple Dome the cloud properties (Figure 22) during the melt period of interest are comparable with the rest of the month,
with average *LWP* values between 21-25 December of 14 and 5 g m$^{-2}$ from CERES and ERA5, respectively, and corresponding average *IWP* values 14 and 19 g m$^{-2}$ from CERES and ERA5, respectively. After 25 December there is an impulse of *LWP* > 80 g m$^{-2}$ that is simulated by ERA5 two days before it is detected in the CERES MODIS-based remote sensing data. This cloud intrusion having moderate to large optical thickness may help explain the skin and 2m air temperatures between 27-28 December, which are very close together, at or just below freezing, and at monthly maximum
values. Overall, this case study suggests a thermal blanketing episode at Siple Dome driven primarily by a positive SH flux impulse that began on 19 December, that caused a delayed melt onset as detected by satellite PMW data three days later, and that may have been prolonged by a cloud radiative effect 5-7 days later. The slow melt onset may be the result of smaller total *ME* during 21-25 December, including the drops below zero, as compared with all the other cases considered in this work. Field camp observations at Siple Dome indicate cloud cover ranging from scattered to overcast between 22-25

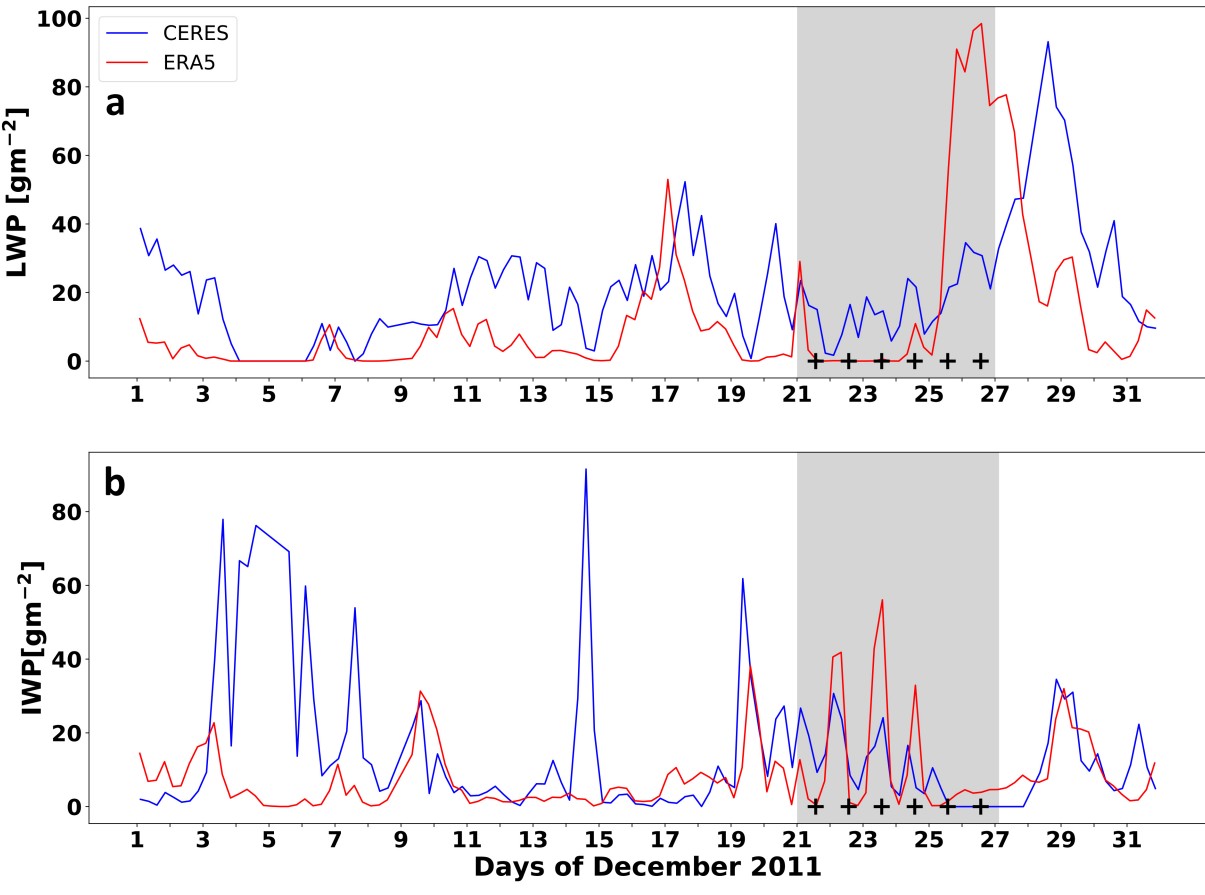

**Figure 22. As in Figure 5, but over the Siple Dome region throughout December 2011.**

December, with most observations also showing reduced visibility in freezing fog, drifting snow or blowing snow. On 24 December an overcast layer is noted at 2350 UTC with cloud base 1400 m and light snowfall. Throughout 26 December the ceiling is obscured by mist, freezing fog or drifting snow. On 27 December low cloud and overcast conditions are recorded throughout most of the day with cloud base ~300 m. These observations are qualitatively consistent with the radiative flux components of Figure 19a,b and the cloud properties of Figure 22.

At the RIS location cloud properties between 19-23 December (Figure 23) show ERA5 simulated and CERES detection of cloud cover that are consistent in time, and subject to the same potential uncertainties and errors as in the previous case studies. With CERES *LWP* and *IWP* values of 14.7±11.0 g m$^{-2}$ and 32.7±16.1 g m$^{-2}$, respectively, the clouds are likely to be optically thin and causing an all-wave radiative enhancement. Here we should consider the possibility that the clouds might

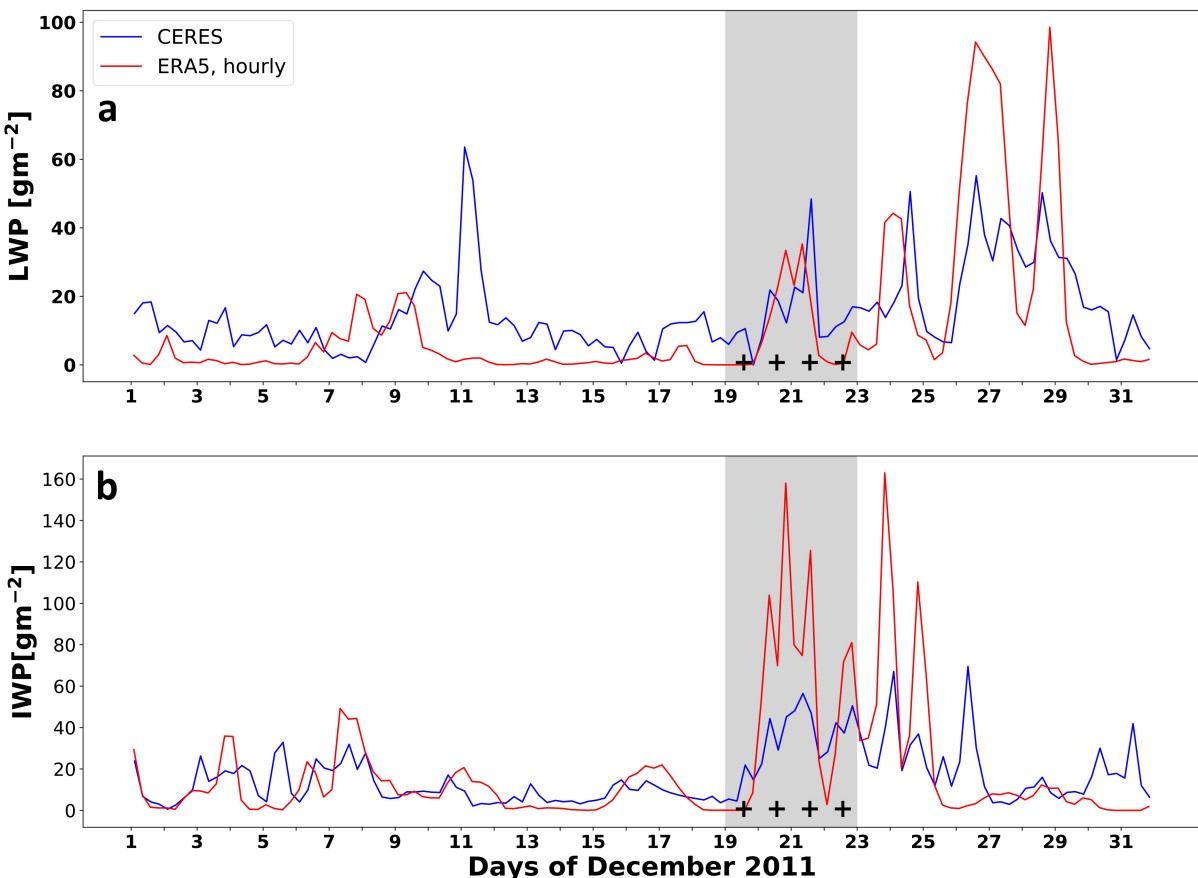

**Figure 23. As in Figure 5, but over the RIS region containing the Tom and Sabrina AWS, throughout December 2011.**


be optically thin in part due to a cloud-clearing effect of the föhn wind; the other two locations during late December 2011 saw optically thicker clouds during the surface melt conditions.

We now examine the local-scale meteorology at the RIS location in more detail, to illustrate the föhn wind effect. Figure 24 shows ERA wind speed and direction at the surface and at 850 hPa. Between 9-19 December winds are light to moderate, and have a variety of directions but are mostly northerly between 9-14 December and 18-19 December. During the melt period 23-25 December, surface and lower troposphere winds strengthen and their directions become more spatially uniform, mainly easterly to southeasterly, consistent with descent into the region from the Transantarctic mountains.


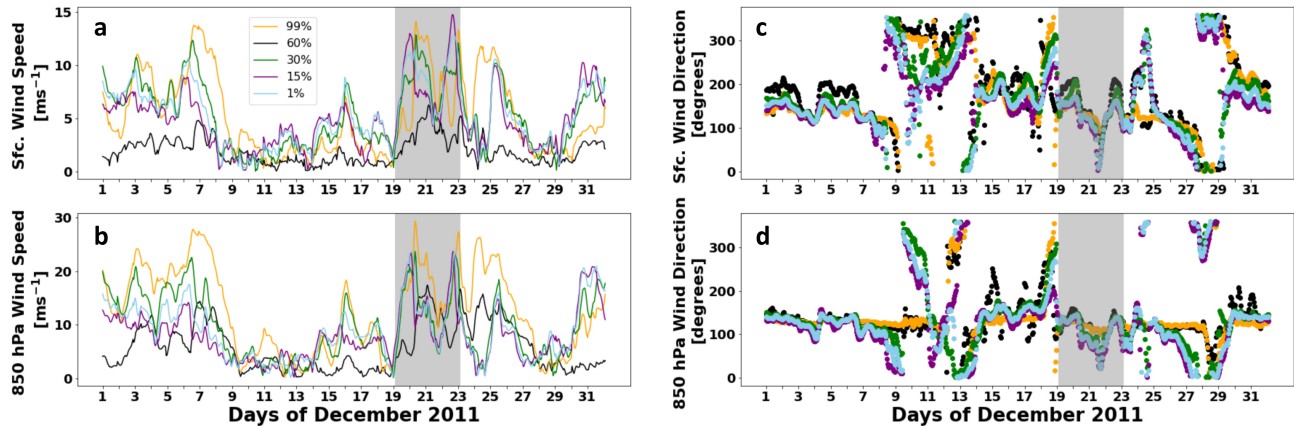

**Figure 24. Time series of sampled ERA5 (a) near-surface wind speed, (b) 850 hPa wind speed, (c) near surface wind direction, (d) 850 hPa wind direction over the RIS region containing the Tom and Sabrina AWS, throughout December 2011. Percentiles sampled correspond to the locations of Figure 18b.**


In Figure 25 we examine in situ measurements of 2m air temperature, relative humidity, wind speed and wind direction from the Tom and Sabrina AWS. During the time intervals 9-19 December and 19-23 December, these measurements are very consistent with the ERA5 values (Figures 20 and 24). Between 19-23 December, the two local maxima in wind speed (Figure 25c) correspond to minima in relative humidity (Figure 25b), along with a slight westward shift in wind direction

(Figure 25d), and these changes are consistent with föhn wind occurrence. However, between 23-26 December surface wind speed is consistently stronger and wind direction is more consistently southeasterly at both AWS than in the ERA5 reanalysis data, although the 2m surface air temperatures compare well. A possible cause of this discrepancy might be the coarse spatial resolution in ERA5, yielding an underprediction of föhn winds (e.g., Trusel et al., 2013). The ERA5-based analysis (Figures 20 and 23) suggests that the initial föhn wind onset combined with a cloud radiative enhancement gradually

set up the conditions starting on 20 December that lead to satellite PMW melt signature detection on 23 December. Absent the cloud radiative enhancement after 22 December, the AWS data suggest that persistent föhn winds alone can sustain the surface melt conditions for several more days. We do note that the underprediction of föhn winds in ERA5 might be offset by larger LWC and IWC that are retrieved in the CERES data (Figure 23).

Finally, we note that between 1-9 December there are strong surface and lower troposphere winds from a southeasterly direction, seen in both ERA5 and AWS, that induce consistently positive SH flux and positive net turbulent flux over at least three diurnal cycles. These observations would also be consistent with föhn winds from the Transantarctic mountains. Skin temperatures and 2 m air temperatures are also 3-5 K warmer than during the subsequent time interval 9-19 December. However, cloud cover appears to be consistently light in both the ERA5 simulations and CERES retrievals (Figure 23) that

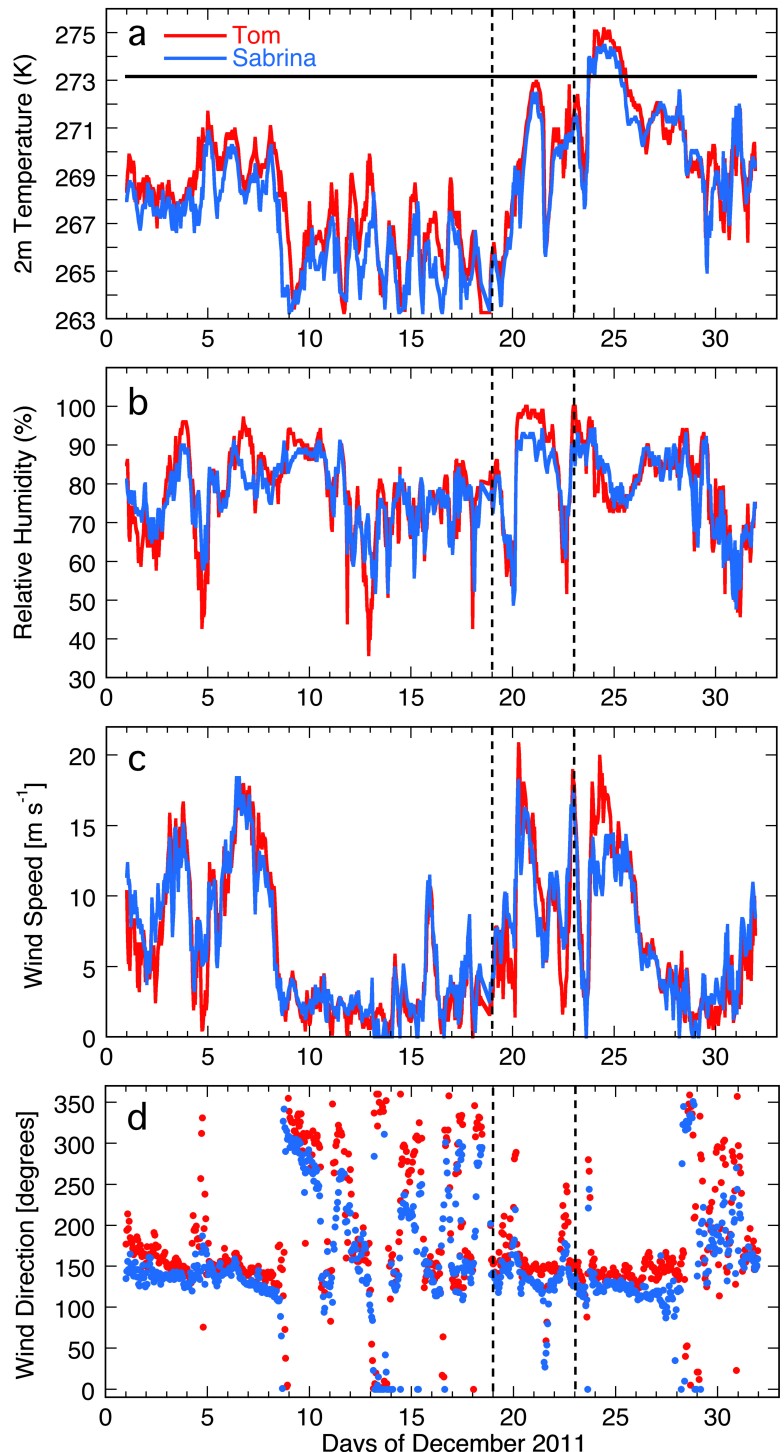


Figure 25. Time series of 2 m meteorological measurements from the Tom and Sabrina AWS throughout December 2011: (a) air temperature, (b) relative humidity, (c) wind speed, (d) wind direction.

allows for LW cooling (Figure 20a), and the total ME remains mostly negative before 19 December. Early in December the
synoptic conditions discussed above have not yet set up the warm air intrusion that brings moisture and cloud cover to all
three locations. A downslope wind by itself may not be sufficient to cause a detectable surface melt event (e.g., King et al.,
2017), but may need to operate in conjunction with additional conducive atmospheric conditions.

**Conclusion**

In this study we demonstrate that readily available climatic data, including meteorological reanalysis and satellite remote
sensing, can be used to examine and diagnose individual episodes of surface melt over Antarctic ice sheet and ice shelf
locations that are of significant concern in a steadily warming climate. We demonstrate examples for each of three
thermodynamic mechanisms that induce surface melting. The case study from January 2015 over Siple Dome very likely
involves the same all-wave cloud radiative enhancement discovered over the GIS (Bennartz et al., 2013; Van Tricht et al.,
2016). In contrast, Pine Island and Thwaites Glaciers during December 2011 experience a thermal blanketing effect where
the positive *ME* impulse comes mainly from optically thick clouds. Over the Tom and Sabrina AWS locations on the RIS,
we identified a föhn wind effect that might be augmented by an all-wave cloud radiative enhancement. Other examples when
two of the mechanisms are at work include the December 2011 thermal blanketing case over Siple Dome, where an impulse
of positive SH flux set up the surface conditions for melt onset followed by additional energy input from a cloud radiative
enhancement, and the February 2013 thermal blanketing case over Pine Island and Thwaites Glaciers, when optically thick
cloud cover initiates a melt event that is subsequently prolonged by positive SH flux.

For the Pine Island and Thwaites Glaciers region we notice considerable local-scale variability in surface $T_b$ and emissivity,
possibly related to microwave signatures dominated by new snow in some grid cells and by older snow or prior melt events
in others. Bell et al. (2017) show that local-scale variability on Antarctic ice shelves influences whether surface meltwater
filters into the ice as a source or hydrofracturing or runs off in temporary rivers. Local-scale spatial inhomogeneity on the ice
shelves probably requires further investigation to make reliable projections regarding multi-year stress.

Two limitations stand out with the present level of analysis. First, improvements are needed in cloud microphysics and
related optical properties in both the reanalysis models and in the satellite remote sensing retrievals. AWARE ground-based
remote sensing data have fostered some progress in this respect, in providing confidence in MODIS retrievals of cloud
microphysical properties (Wilson et al., 2018), and in providing unique data for modelling case studies (Hines et al. 2019;
Silber et al., 2019; Lubin et al., 2020). Presently throughout the ASE, although the presence of cloud in a case study is
reliably detected, the microphysical uncertainties sometimes prevent a full diagnosis of the melt event mechanism. For
example, in the January 2012 case study over Pine Island and Thwaites Glaciers, a cloud radiative effect is clearly indicated

but it is not clear if this is a thin cloud all-wave effect or an optically thick thermal blanketing effect. In atmospheric models, the use of double-moment cloud microphysical parameterizations makes noticeable improvements over Antarctica (e.g., Hines et al., 2019). However, these more rigorous parameterizations are found mainly in global climate models. Numerical weather prediction models, which are used to generate reanalysis data, must run on an operational forecast schedule and may not be able to accommodate the time-consuming rigorous parameterizations.

We mention that one regional model is known to be useful for this type of work. This is the European Regional Atmospheric Climate Model second version (RACMO2; van Wessem et al., 2018). Lenaerts et al. (2018) have used RACMO2 to accurately simulate West Antarctic melt events between 1979-2015. In RACMO2, van Wessem et al. (2014) addressed the common cloud *LWP* deficiency over Antarctica by altering the model cloud microphysics to allow for more extensive cloud liquid water transport. This is done primarily by making simple but defensible adjustments to the threshold for ice supersaturation (Tompkins et al., 2007), and the critical cloud content for efficient precipitation (Lenaerts et al., 2018). While these simple alterations allow for sufficient cloud liquid water to contribute radiatively to positive *ME* and surface melt onset, the simulated *LWP* values have yet to be thoroughly validated against other data such as SYN1deg. It is therefore not clear if RACMO2 simulations by themselves can discriminate between the mechanisms involving optically thick versus optically thin clouds, and supplementing RACMO2-based analysis with SYN1deg data is therefore recommended.

In the MODIS-based retrievals contained in the CERES SYN1deg data product, we suspect that some of the higher *IWP* values may actually be liquid water clouds. Chylek et al. (2006) suggest that cloud phase discrimination that relies on differential backscatter in MODIS near-infrared channels can be biased toward the ice phase. The MODIS retrieval algorithms for cloud phase discrimination generally use both near- and mid-infrared bands, and further investigation is needed specific to clouds over West Antarctica to identify possible errors. Additionally, the CERES-MODIS approach can retrieve unrealistically high *IWP* values over ice sheets, mainly over the Antarctic interior. An issue with this approach is that over these areas, where the contrast between the surface and cloud albedo is small, a large correction of cloud water path is necessary to match the TOA fluxes since they are insensitive to small changes. Furthermore, since *LWP* has limited observational constraints over Antarctica, the algorithm likely has to resort to increasing the *IWP* dramatically to compensate for any lack of brightness owing to missing liquid (e.g., Lenaerts et al., 2017).

A second limitation involves quantifying the effect of föhn winds. In the RIS example the AWS data indicate more persistent föhn winds than are simulated by ERA5. This is most likely related to the coarse spatial resolution in the reanalysis model. While ERA5 can identify the likely presence of a föhn wind effect based on its generally accurate lower troposphere wind direction relative to varying high terrain, a more quantitative analysis might need to incorporate detailed knowledge of the actual terrain elevation (Dreschel and Mayer, 2008; Elvidge et al., 2015; King et al., 2017).

Over the modern satellite record spanning nearly four decades, it should be possible to make projections regarding future atmospheric stress on the West Antarctic ice shelves by identifying the specific mechanisms, their frequency of occurrence singly or concurrently, their relationships with large-scale meteorological drivers (Nicolas and Bromwich, 2011; Scott et al., 2019) and transport and abundance of atmospheric precipitable water (e.g., Suzuki et al., 2013; Wille et al., 2019). The analysis methods presented here, in which the energetics of individual melt events are diagnosed from satellite observations and reanalysis data, can supplement recent large-scale analysis using regional modelling (e.g., Deb et al., 2018). Our individual cases and their meteorological drivers are qualitatively consistent with the large-scale modelling analysis of Deb et al. (2018). In conjunction with increasing understanding of shelf basal melting and its time variability (Adusumilli et al., 2020) and understanding the disposition of surface meltwater either within the structure of Antarctic ice shelves or as runoff (e.g., Bell et al. 2017), one can also envision a quantitative assessment of ice shelf resilience in a warming climate based on analysis of the surface energy balance.

## Appendix A: Supplementing ERA5 Melt Energy Calculation with Satellite-Retrieved Cloud Microphysical Properties

Silber et al. (2019) compared ERA5 data with AWARE data from the WAIS Divide Ice Camp and found a tendency for ERA5 to overestimate cloud ice water content and underestimate cloud *LWP*. We therefore compare the ERA5 skin temperature, downwelling SW flux and downwelling LW flux with the AWARE measurements at WAIS Divide in Figure A1, to estimate how errors in ERA5 cloud microphysics might impact a time series of the *ME* before and during a melt event. The AWARE flux measurements were made using the ARM user facility pyranometers and pyrgeometers (Mather & Voyles 2013; Lubin et al. 2020). Figure A1a shows that ERA5 consistently underestimates skin temperature except on occasions when the Sun is at its lowest elevation, but that the temperature discrepancy varies from day to day. The instantaneous discrepancies between ERA5 and the measured downwelling SW flux (Figure A1b) can sometimes be on the order of 100 W m$^{-2}$, but the similarity in amplitudes of the diurnal cycles suggest that ERA5 is reliably simulating the presence of clouds on a daily basis. Much more striking discrepancies appear between ERA5 and measured downwelling LW flux (Figure A1c). Here there are many periods, sometimes a day long, where ERA5 underestimates the LW flux by ~50 W m$^{-2}$, which would be expected if modelled *LWP* is too low (see Figure 14 in Lubin et al. 2020). There are, however, other periods when the ERA5 and measured LW fluxes are consistent. This episodic nature of the LW flux discrepancies, in which errors can persist throughout a day, suggest that we should find alternative estimates of the cloud *LWP* and ice water path (*IWP*) to evaluate the realism of LW flux calculations in the *ME* based on ERA5 data.

Our goal is to be able to evaluate the energetics of surface melt events anywhere in Antarctica, rather than be tied to the few instances such as AWARE where corroborating surface measurements are available. We therefore examine the contrasts between ERA5 and CERES SYN1deg cloud properties and radiative fluxes during the AWARE January 2016 melt event but at Siple Dome instead of WAIS Divide. From Nicolas et al. (2017) we know that clouds should be optically thick and that

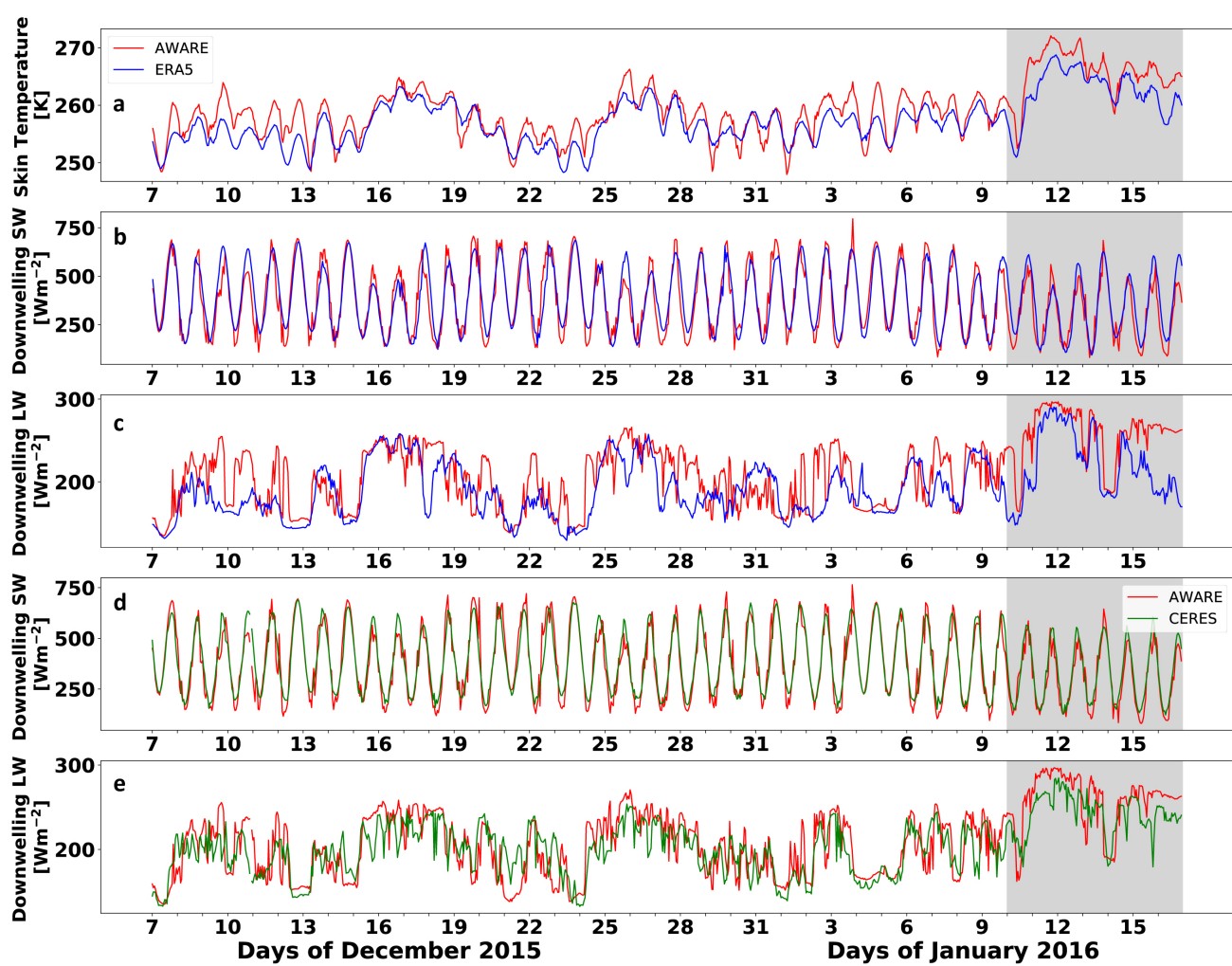

**Figure A1. Time series comparison of hourly SEB components from ERA5 at the WAIS Divide Ice Camp with surface measurements from AWARE (Lubin et al. 2020; red curve in all plots): (a) skin temperature from ERA5; (b) downwelling SW flux from ERA5; (c) downwelling LW flux from ERA5; (d) downwelling SW flux from CERES SYN1deg; (e) downwelling LW flux from CERES SYN1deg. The shaded region indicates the WAIS January 2016 melt event period (Nicolas et al. 2017). One hour of surface radiometric data is missing on 10 December 2015, but the data are continuous afterward.**

the *ME* should be positive over several diurnal cycles after 10 January 2016. Over Siple Dome during the melt event, both ERA5 and CERES indicate *LWP* > 50 g m$^{-2}$ (Figure A2). However, ERA5 cloud *IWP* is sometimes twice as large as the CERES retrieval.

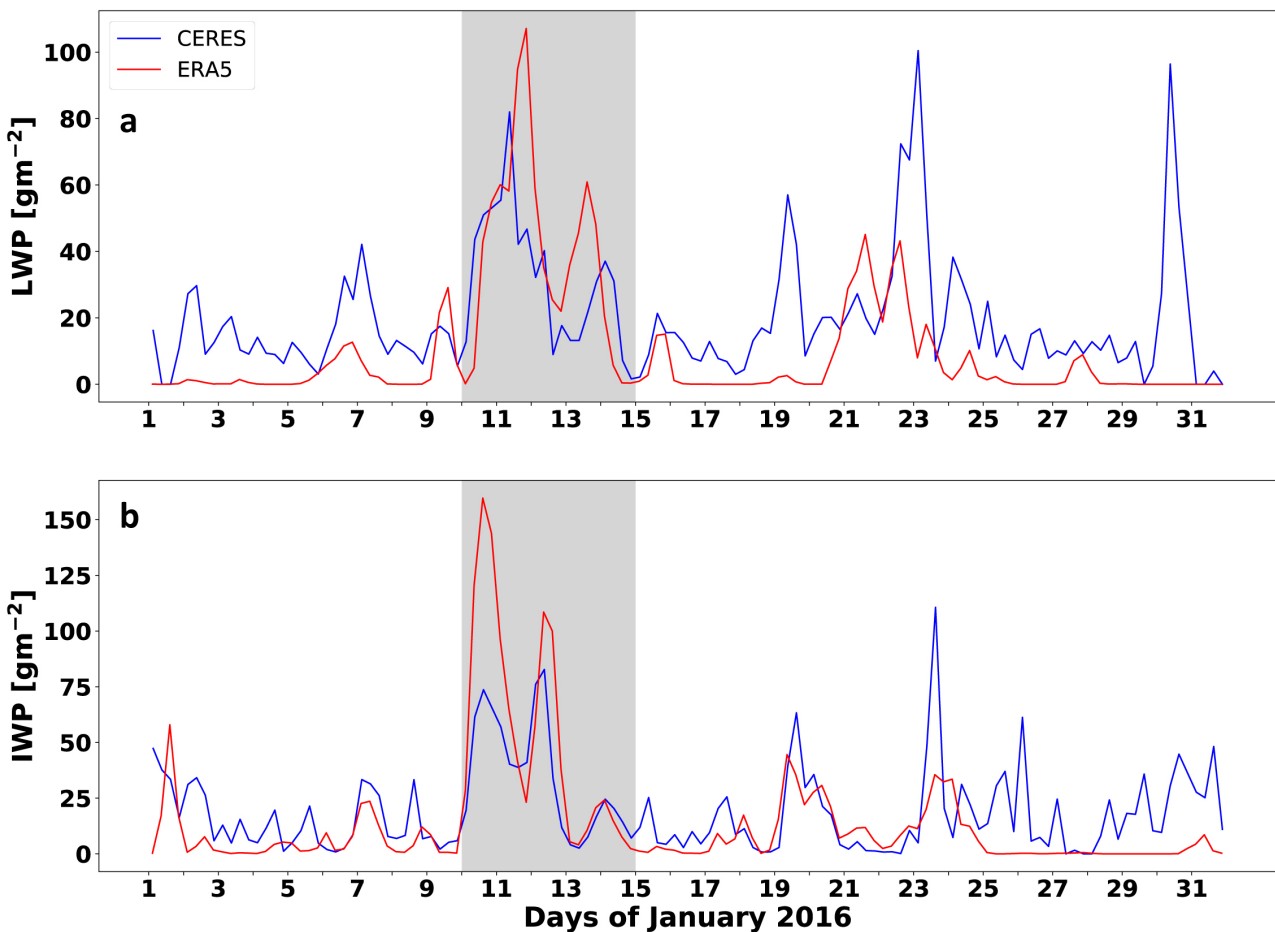

**Figure A2. Time series of hourly cloud *LWP* (a) and *IWP* (b) from the CERES SYN1deg data product (red) and ERA5 (blue), over the Siple Dome region throughout January 2016. The shaded region denotes the time period of the WAIS large-scale melt event (Nicolas et al. 2017).**

If cloud microphysics are more realistic in the CERES data product, one might be tempted to calculate the *ME* by replacing ERA5 net radiative fluxes with their CERES counterparts, while retaining the ERA5 turbulent fluxes. We tried this approach for January 2016 over Siple Dome (Figure A3) and the result is unsatisfactory. The diurnal amplitude of the CERES net SW flux is up to twice as large as that modelled by ERA5, and is also qualitatively less consistent with the AWARE measurements from WAIS Divide. There are substantial differences of order 50 W m$^{-2}$ between ERA5 and CERES net LW fluxes, with CERES appearing to be an improvement compared with ERA5's known tendency to underpredict the net LW flux over Antarctica (Silber et al., 2019). However, the *ME* calculation using ERA5 for all flux terms is basically realistic in that *ME* > 0 over three diurnal cycles after 10 January, and almost always drops below zero at lowest Sun elevation for the

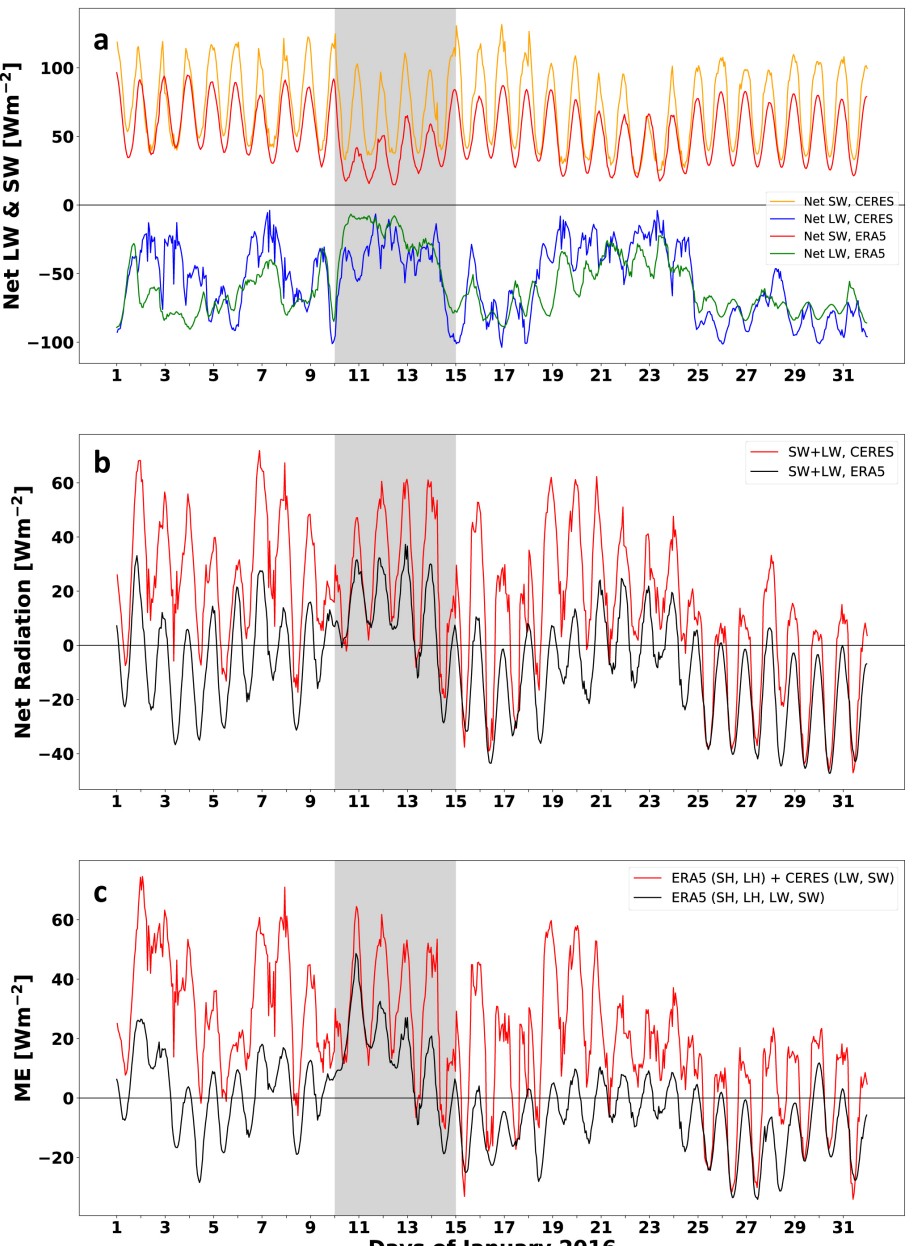

**Figure A3. Radiative flux components and alternative estimates of the *ME* over Siple Dome during January 2016: (a) Individual net SW fluxes from ERA5 (red) and CERES SYN1deg data (yellow) and net LW fluxes from ERA5 (green) and CERES SYN1deg data (blue); (b) total net radiative flux from ERA5 (black) and CERES SYN1deg data (red); (c) *ME* computed entirely from ERA5 (black) and using ERA5 turbulent fluxes but substituting the CERES SYN1deg radiative fluxes (red).**

rest of the month. When we substitute the CERES radiative fluxes, both the net (SW + LW) radiative flux and *ME* are
positive over several diurnal cycles for about half the month, including before 10 January when we know that meteorological
conditions were not conducive to surface melt (Nicolas et al., 2017). We therefore conclude that a "mix and match" approach
to evaluating the *ME* is unsuitable, and this is not surprising given that ERA5 and CERES use different radiative transfer
algorithms. Instead, we proceed by calculating the *ME* with ERA5 radiative and turbulent fluxes, and then examine the
CERES SYN1deg cloud *LWP* and *IWP* as a separate check on the realism of cloud properties simulated by ERA5.

**Appendix B: Examples of Satellite Passive Microwave Brightness Temperature Spatial Variability**

To illustrate the spatial variability in the surface melt signature, we provide examples of the SSMIS horizontally polarized
19.35 GHz (K-band) brightness temperature $T_b$ measured on the days during each of the case studies when surface melt
reached maximum frequency within the bounding region. At Siple Dome on 6 January 2015 (Figure B1) the extensive
surface melting also appears over the eastern edge of the RIS and throughout most of the ASE. On 6 January 2012, there is
considerable spatial variability in $T_b$ over Pine Island Glacier and more uniformity over Thwaites Glacier (Figure B2), in
response to the synoptic situation that normally doesn't favour surface melt. Similarly, during the late summer melt event of
February 2013, there is noticeable spatial variability in $T_b$ over both Pine Island and Thwaites Glaciers (Figure B3), even
though this melt event is driven by pronounced thermal blanketing. During the December 2011 synoptic conditions that
strongly favour melt, spatial variability in $T_b$ over Pine Island Glacier is still apparent (Figure B4). Over Siple Dome during
late December 2011 (Figure B5) the measured $T_b$ exhibits spatial uniformity and values ~50 K smaller than over Thwaites
Glacier (Figure B5). At the RIS location on 23 December 2011, spatial variability in Tb is consistent with a föhn effect, as $T_b$
is above the melt detection threshold close to the Transantarctic mountains and decreases throughout the bounding region
moving away from the mountains.

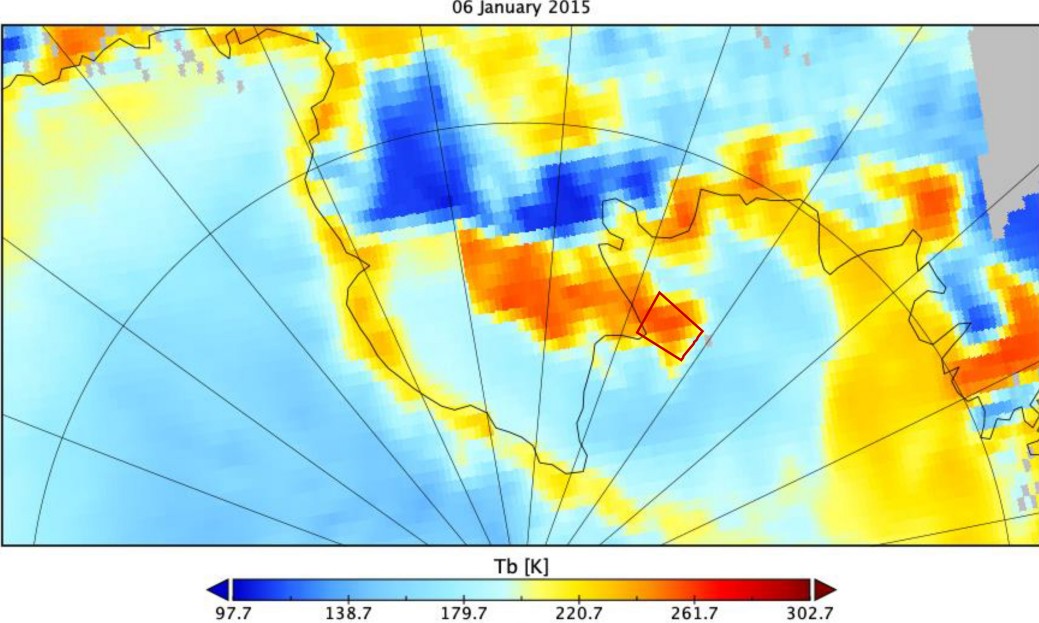

**Figure B1. SSMIS horizontally polarized 19.35 GHz brightness temperature over West Antarctica and the RIS on 6 January 2015, with the red box denoting the Siple Dome bounding region used in the case studies.**

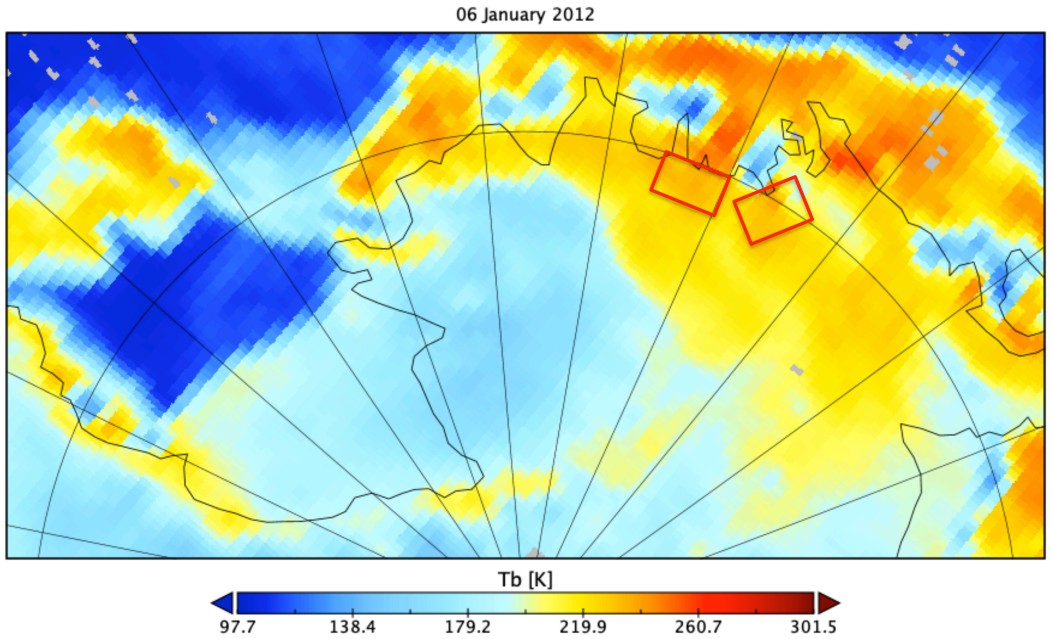

**Figure B2. SSMIS horizontally polarized 19.35 GHz brightness temperature over West Antarctica and the RIS on 6 January 2012, with red boxes denoting the Thwaites Glacier (left) and Pine Island Glacier (right) bounding regions.**

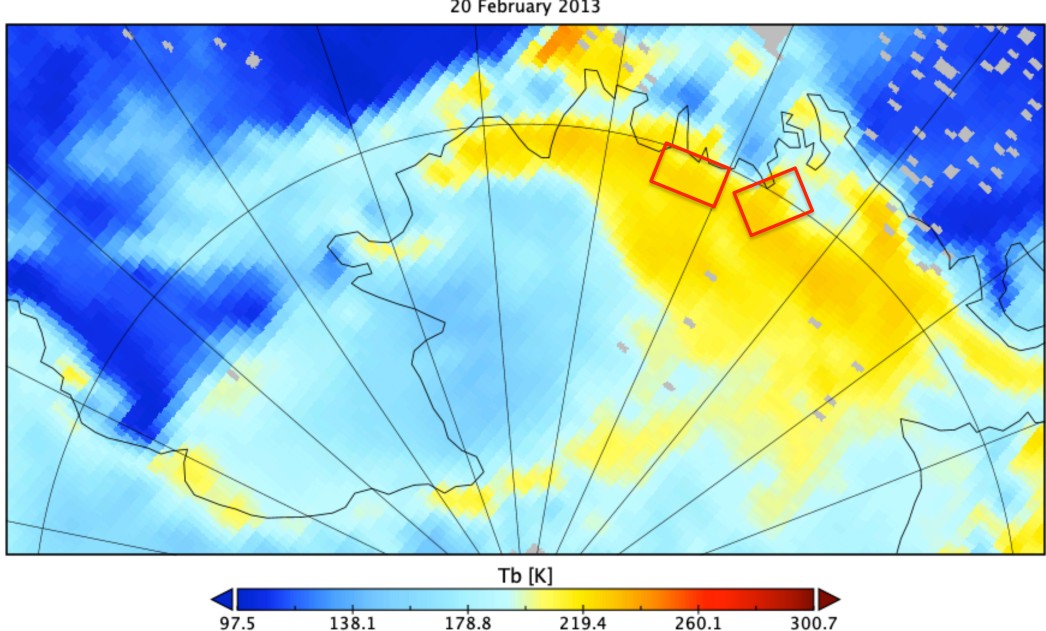

**Figure B3. SSMIS horizontally polarized 19.35 GHz brightness temperature over West Antarctica and the RIS on 20 February 2013, with red boxes denoting the Thwaites Glacier (left) and Pine Island Glacier (right) bounding regions.**

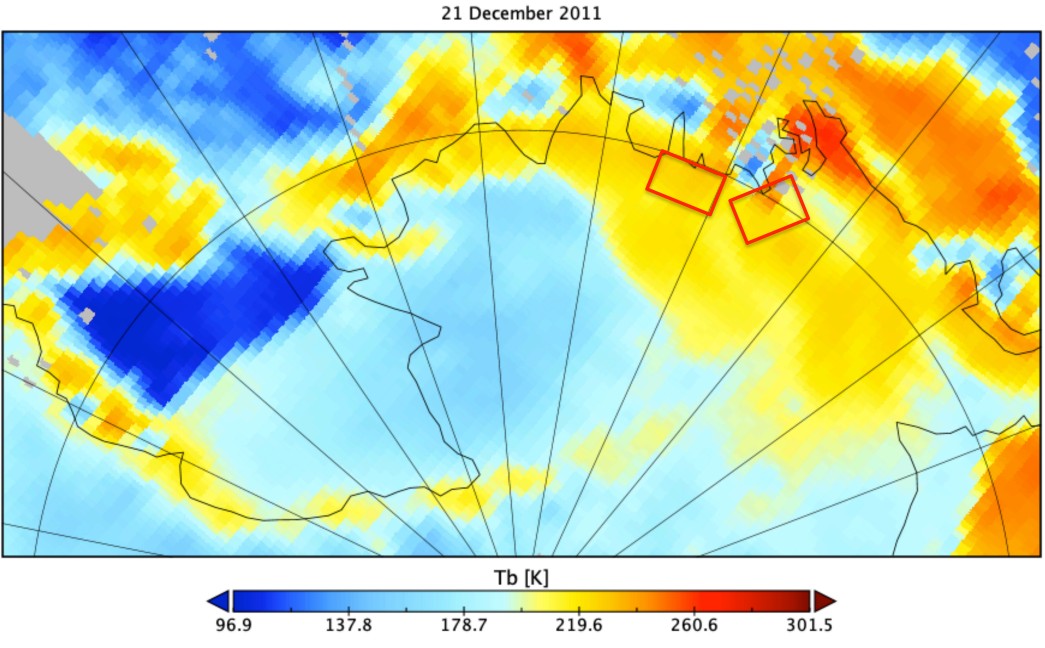

730

**Figure B4. SSMIS horizontally polarized 19.35 GHz brightness temperature over West Antarctica and the RIS on 21 December 2011, with red boxes denoting the Thwaites Glacier (left) and Pine Island Glacier (right) bounding regions.**

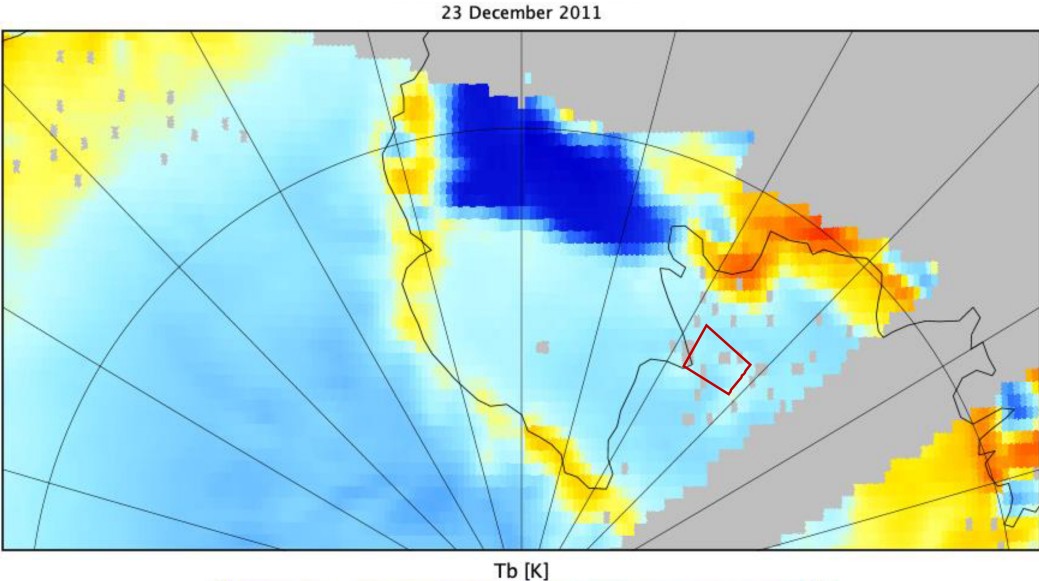

**Figure B5. SSMIS horizontally polarized 19.35 GHz brightness temperature over part of West Antarctica and the RIS on 24 December 2011, with red box denoting the Siple Dome bounding region.**

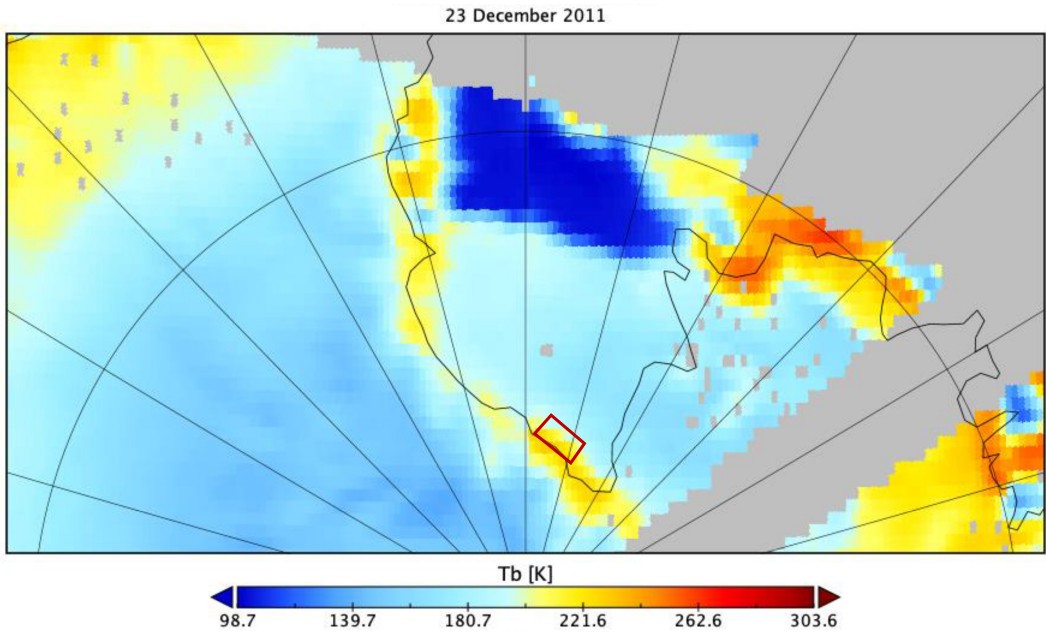

**Figure B6. SSMIS horizontally polarized 19.35 GHz brightness temperature over part of West Antarctica and the RIS on 24 December 2011, with red box denoting the bounding region containing the Tom and Sabrina AWS.**

## Data Availability

ERA5 data were obtained as provided by ECMWF, using the Copernicus Climate Change Service (C3S) Climate Data Store (https://cds.climate.copernicus.eu). NASA CERES SYN1deg data were obtained from NASA Langley Research Center Atmospheric Science Data Center (https://asdc.larc.nasa.gov/project/CERES). MEaSUREs EASE-Grid 2.0 data were obtained from NSIDC (https://nsidc.org/data/NSIDC-0630/versions/1). AWS and Field Camp Observations are archived at the University of Wisconsin Antarctic Meteorological Research Center (https://amrc.ssec.wisc.edu/data).

## Author Contribution

MG performed the data analysis and interpretation as part of her Master of Science thesis at the Scripps Institution of Oceanography. RS provided synoptic and local scale meteorological analysis. AV provided the AWARE surface energy balance data analysis and contributed to manuscript preparation. JL provided interpretation of the surface energy balance in the case studies and contributed to manuscript preparation. ML provided the AWS and field camp data and their interpretation for this work. DL served as thesis advisor for MG and contributed to manuscript preparation.

## Acknowledgments

MG and DL were supported by NSF Grant OPP-1744954 and US Department of Energy Grant DE-SC0017981. AV was supported by DOE ASR under contract DE-SC0012704. JL was supported by the National Aeronautics and Space Administration under grant 80NSSC18K1025. ML was supported by NSF Grant OPP-1924730. We thank Dr. Walt Meier of NSIDC for assistance with the MEaSUREs EASE-Grid 2.0 data.

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
