# Peer review of "Energetics of Surface Melt in West Antarctica"

_The Cryosphere, 2020_

## Referee Comment (RC1) · Anonymous Referee #1 · 9 Nov 2020

Summary

The manuscript uses multiple data sources (satellite, AWS, reanalysis) to explore the contributing meteorological factors influencing the SEB and surface melting over West Antarctica. Specifically, the authors examine the influence of optically thick cloud that induces a thermal blanketing effect, optically thin clouds that enhance all-wave radiation, and foehn winds. They explore these mechanisms by evaluating several case studies from various locations in West Antarctica.

Adherence to evaluation criteria

This paper specifically examines the effects of processes that have previously been considered in isolation or in a less direct manner, and quantifies their impact on melting by unravelling their effect on the surface energy balance (SEB). This represents new insight and advances our understanding of cloud processes and foehn events on the

SEB of West Antarctica, and so demonstrates the originality required for publication in The Cryosphere and fits well within the aims and scope of the journal.

Overall, I am convinced of the scientific rigour of the methodology and believe that the manuscript contains enough detail to make the methods reproducible. I have some minor comments and clarifications on certain aspects of the methods, which are given below. The specific contributions of this work in relation to the existing literature could be outlined more explicitly, but generally there is adequate consideration of related work. The limitations of the method are discussed and addressed appropriately.

The results and conclusions are presented clearly, although the text can be overly descriptive and somewhat repetitive in places. I have provided some suggestions for re-structuring some of the case study sections to improve readability. Overall the language is satisfactory, but there are still some outstanding typos and errors that could be rectified with a thorough proof-read. I have highlighted some below in my comments. The figures used are appropriate but are again repetitive. I have offered some suggestions in my comments that may enhance the reader's understanding of the results.

The authors reach substantial conclusions regarding the importance of the three mechanisms outlined for determining melt. These findings are likely applicable to other parts of Antarctica, which gives them broader significance. The results fit within the current literature regarding Antarctic SEB and the drivers of melt, and represent a development in our scientific understanding of the effect of cloud properties and atmospheric dynamics on melting.

General points

- The manuscript is very descriptive, at times overly so. It may help the reader if you more explicitly draw the links between the general synoptic conditions that are described at the start of each case study and their impact on the mechanisms involved and the SEB/melt.

- Each of the case studies is comprehensively evaluated and a great deal of analysis has evidently been done. However, it does get a little repetitive reading each of the case studies in turn because they all follow the same structure. My suggestion would be to combine the discussion of e.g. Siple + PIG/Thwaites (2nd and 3rd cases), and possibly the Ross Ice Shelf (6th case) into a 'December 2011' case, which might help the reader and facilitate better discussion of the influence of synoptic meteorology on melting at these two/three locations. You can then discuss the ways the same event affects the SEB similarly/differently at the different sites.

- The similarity of the figures is also a little repetitive. Whilst I understand that this is difficult to avoid because of the nature of your analysis, I wonder if there is a way of combining some elements of the figures to communicate the differences between case studies/mechanisms more concisely? For example, could you show a scatter plot of IWP/LWP vs melt or Rnet for the optically thin/thick cloud cases to see how the optical depth influences these? Or contrast the time series of melt/emissivity for different case studies in one figure with multiple sub-plots? Some spatial representation of these case studies, e.g. maps of ERA-5 modelled ME, synoptic meteorology, emissivity or specific fluxes might also help the reader understand these cases better.

- It would be helpful for the reader to more clearly indicate which mechanism you believe is responsible for driving melt at the start of each case study section, possibly even in the subtitle heading.

- I would like to see more discussion of the impact of cloud phase on melting. Ice and liquid have quite different radiative impacts, so this might affect your results.

- More concrete linkages should be drawn between the melt mechanisms and large-scale drivers, such as those identified by Scott et al. (2019).

- If you only refer to something once or twice, you probably don't need to define an acronym for it (e.g. ARM, ARs, GIS, AMRC, DMSP). This helps the reader because they don't have to refer back to remind themselves what the letters stand for.

[Figure]

- Do you see the effect of the Amundsen-Sea Low (ASL) in your results? This has been shown to be linked with the SAM (and ENSO) and can influence the advection of maritime air onto the continent (Hosking et al., 2013 – doi:/10.1002/joc.3558 ; Clem et al., 2017 doi: 10.1175/JCLI-D-16-0891.1)

- How do your results compare with high-resolution regional modelling? Are there implications for e.g. surface mass balance? E.g. Lenaerts et al. 2017 doi: 10.1017/aog.2017.42; Deb et al., 2018 doi; 10.1029/2018GL077092)

Specific points

[L35-37] This would be a good place to reference Lhermitte et al. (2020) on structural destabilisation of West Antarctic ice shelves. doi: /10.1073%2Fpnas.1912890117

[Para starting L80] quite verbose, could be revised to be clearer

[L83-85] Does the fact that the satellite retrievals are instantaneous affect melt detection?

[L105] define GIS on first usage

[L106-107] Suggest including a citation to support this point (frequency of cloud LWP 10-40 g m-2 in the Antarctic, plus radiative effect of this cloud)

[L111-116] Citation for definition of foehn – e.g. Elvidge & Renfrew (2016) for various mechanisms doi:/10.1175/BAMS-D-14-00194.1

[Section 2] How are foehn detected in the AWS data? Further detail is required here.

[L186-194] Discrepancies of 50-100 W m-2 are quite considerable, and certainly could mean the difference between melt and no melt, especially in non-summer seasons. All reanalyses struggle with cloud properties, so you would expect some level of disagreement in terms of radiative fluxes, but I wonder if this could be improved upon. Did you look at any other reanalysis products to find out whether any others perform better?

[L206] comma should be after 'event' not 'melt'

[Fig 2] Is there missing data in panels d and e between December 10-11?

[L263] missing full stop after parenthesis

[L276] misplaced comma – add after 'daily statistics'

[L 280-281] Explanation of Fig 5b unclear. Suggest that you explain which 5 cells were selected and/or how they were chosen, and rephrasing to more clearly state that these are expressed as percentiles of the max Tb observed with satellite retrievals.

[L292] Some comment on how you might expect the high IWP values used in the ERA5 radiative transfer scheme to influence the radiative fluxes would be informative here. e.g. competing SW/LW effects

[293-295] What is the implication of the greater vertical extent of cloud ice vs liquid regions? How might this influence the surface energy balance? I am wondering about e.g. temperature and altitude effects, as well as how the ice/liquid water paths influence LW/SW scattering and emission of the clouds. Cloud vertical profile is shown to be important in determining radiative flues and melt in e.g. Gilbert et al. (2020) doi:10.1002/qj.3753 on the peninsula in summer.

[L322] missing 'the' before 'lowest sun' ?

[L323-325] Quantifying these statements would be helpful. What, specifically does 'unexceptional' mean? Perhaps give an average. Similarly for 'moderate to high LWP'.

[para beginning L318] Presumably the source of the SH flux is warm air advection associated with the warm air intrusion during Dec 2011? Possibly worth re-stating this here.

[327-330] Further explanation of the mechanisms causing delayed-onset melt and then sustained melt would be informative.

[L331-332 (Fig 8 caption)] See comment above at line 280-281.

[L344] Not sure 'low overcast' is a noun. Please revise (e.g. low cloud/ overcast conditions/ similar)

[L358-361] Can you speculate about sources of homogeneity in the emissivities? E.g. snow properties, surface albedo, melt ponding, topography etc.?

[L381-383 (Fig 11 caption)] See comment above at line 280-281.

[Section 3.3] – I would consider combining this section with 3.2 (and possibly 3.6) in order to better discuss the influence of synoptic conditions on the cloud properties, emissivity and SEB at these locations. As it is, the manuscript is quite repetitive as each case study follows the same format with very similar figures, so this may help the reader draw parallels and distinctions between the same event manifested at different locations.

[L435-436 (Fig 14 caption)] See comment above at line 280-281.

[para beginning 445] – this sounds like the signature of the ASL in the Ross Sea. Could be worth tying this in, to make more connection between large-scale synoptic conditions and local effects.

[L499-501] This signature (positive SH, negative LH) is frequently indicative of foehn conditions. Adding a statement + citation to this effect would strengthen your argument (e.g. Datta et al., 2019; Elvidge et al., 2020, 10.1029/2020JD032463; Kuipers Munneke et al., 2012, doi: 10.5194/tc-6-353-2012; 2018, doi:10.1029/2018GL077899).

[L514-516] Might the optically thin clouds suggested by your results be indicative of the 'cloud clearance' effect of foehn? i.e. clouds that might have been optically thicker in the absence of foehn (such as those you have shown at PIG/Thwaites/Siple Dome) due to the presence of warm, moist air from the Southern Ocean are thinned by the warming/drying effect of foehn?

[Figure]

[para beginning L566] Might ground-based remote sensing from the AWARE project be useful for examining cloud microphysics? Could this be a way of improving the detection of these sorts of cloud-driven melt events?

---

## Referee Comment (RC2) · Anonymous Referee #2 · 19 Nov 2020

This research looks at how various meteorological conditions alter the surface energy balance (SEB) and produce melt at a number of locations in the West Antarctic region. Surface melting, even during summer, is still spatially limited in the Antarctic, compared to Greenland. However, recent studies, including this one, have highlighted the potential for intense melt events on the surface of ice shelves, which has important implications for ice shelf stability and dynamics. The authors present a number of case studies and explain the causes for the melt, and which SEB components initiated it. A range of processes and mechanisms are included, such as thick thermal-blanket clouds, thin cloud cover and föhn winds. The authors use a range of available data and evaluate it thoroughly. I believe this work is novel, relevant and an important contribution to the field. It also fits the scope of the journal very well. An interesting aspect was the inclusion of manual observations of clouds and visibility, which you don't regularly read in manuscripts now, but undoubtably offer important insight into cloud characteris-

tics as they are so poorly represented in reanalysis and models. Whilst the analysis is robust and thorough, the manuscript suffers from some presentation issues as well as being overly long and descriptive. Some of this is due to the number of case studies, which leads to repetition of both figure types and meteorological description. However, I think with some re-structuring and inclusion of some maps, the manuscript is a good scientific publication which should be accepted. I have outlined the larger issues below and also included line-references for more specific edits. Whilst I give specific areas to cut out and reduce the length, I do urge the authors to also go through and decide on any other areas which can be slimmed down.

Minor suggestions: The length of the manuscript and the repetition in results and their presentation currently makes it feel a little harder to read than it needs to be. There are 6 case studies provided, which is quite a lot for a 'case study' style paper. However, all are valid and important. I therefore suggest restructuring or combining some case studies. Here I have some potential ideas, but I think the authors should decide what works best for them and the manuscript. 1) Combine the December 2011 case studies. Whilst different processes are at play in these case studies, the larger-scale processes and synoptic situation will be the same/similar. Combining them to discuss the elements which are the same will reduce repetition, whilst highlighting the differences between them which show how local-scale processes are important for melting. 2) Combine the case studies by location. E.g combine the PI and Thwaites case studies for December 2011, January 2012 and February 2013. Again, I know that different processes are at play, but it could be advantageous to plot these together to see how melt magnitude/length etc vary between the different meteorological conditions. 3) Combine by melt-mechanism: combine the 'thermal blanket' cases, the ones dominated by turbulent fluxes etc. This would highlight which situation happens more frequently, and which ones are less frequent (but could perhaps generate more melt). The conclusion summarises the case studies very well, and groups them into the cases that have an individual mechanism, and ones that have interplaying mechanisms. Perhaps you could try and include this structure and succinct approach throughout the results.

I really like the conclusions section: it is clear, rounds the paper off well and includes the limitations to the study.

The introduction is quite long and currently reads a bit like a thesis chapter or review paper. This hides your motivation a little, even though there is a need for your research. I would move the section on SEB modelling into the data/methods section, and slim it down. Currently your objectives are at the end of the fourth page, but could be at the forefront if some lines/paragraphs were removed.

Be clearer with which AWS observations you use. As they are mentioned in the data section, but only in passing (e.g line 155/156). And they are mentioned in figure captions (e.g Fig 20 'over the RIS region containing Tom and Sabrina'), but I couldn't be sure where exactly the data are used besides Figure 24. If you do use them elsewhere, you need to make it clearer which locations and which variables. I would also include them on your Figure 1 map.

Figures: • Figure 1 should be improved to include more information. For instance, an insert of the whole of Antarctica is needed with indication of where this zoomed in section is. Number 3 is missing. Could you add the AWS locations? Some schematics or overlays could aid the reader in understanding the large-scale flow in the region, or some arrows to highlight the main air flow directions in your case studies. • Additional figures would also be preferential in highlighting main weather systems for the different case studies. Often in case study analysis (especially in atmosphere-SEB studies), there is a modelling aspect which provides some cross sectional or 2d plots with maps of airflow and such things, which aids understanding for the reader. I am not suggesting you do extra work, but only that you use figures to better represent more useful information. • Perhaps some synoptic charts would help show information, such as the Dec 2011 case with the low-pressure system and föhn winds. Such maps could also indicate the locations of the grid cells used to look at spatial patterns of emissivity. • Many figures of the same style and type- a consequence of a multi-case study analysis- but could be panelled together to avoid repetition. Or move some

to a supplement if some of them are not definitely useful. For instance, is the LWP and IWP plot needed for the case studies where cloud properties are less relevant?

Some of the decision making on ERA5 vs CERES could be moved to a supplement. You could state only what you finally use in the manuscript and move the evaluations and mix-match tests to supplement. Otherwise, it is quite hard to follow which method/dataset you are actually using. For example, after line 200/page 7, I assumed that you weren't using ERA5 anymore. However, you state at the end of page 9 that you do indeed use ERA5.

Is there a timing offset between ERA5 and CERES LWP? In some of the figures, the peaks appear to be always a day or two later.

Specific Suggestions:

Ln 105: GIS hasn't been defined previously Ln 106: Do you have a reference or evidence that these clouds are common in the polar regions? Ln 110: Föhn winds are only common in areas with mountain ranges and perpendicular air flow, which is a relatively small area of the Antarctic. I would say, common in some regions within the Antarctic. Ln 118,119: Föhn winds can't occur anywhere or on any ice shelf- there are many ice shelves in the East where föhn winds do not occur. These may be affected by katabatic winds, but evidence of föhn is lacking. Zhou et al. 2018 only looked at W. Antarctic ice shelves and föhn. Föhn are only common on Antarctic Peninsula ice shelves and Ross Ice Shelf. Throughout: do you have any DOIs or URLs to point the reader to the sources of your data? Ln 206: Add 2016 to the date here. Ln 250-260: I have a hard time knowing how spatially uniform (ln 256) the surface emissivity is, when you don't say which 5 grid cells you are looking at. Where are the 5 grid cells? This sort of information would be better represented as a spatial plot on top of a map, so that we can see the area of interest. Figure 6: a/b/c... missing on figures. Figure 7a/10a: timing offset? Ln 356: 'In Figure 9 the sampled percentiles are referenced to the max Tb on 21 December'. I don't understand what this means. Is it the wrong Figure you

are pointing to? Ln 594-595: I don't understand what you are trying to say here. Figure 24: are you able to plot RH? As this is a key variable in determining föhn winds.

---

## Author Comment (AC1) · 19 Nov 2020

We thank this referee for the very thorough and constructive review of the manuscript. Responding to these questions and suggestions will go a long way toward improving the final version.
* * *

---

## Author Comment (AC2) · 20 Nov 2020

We thank this referee for a thorough and constructive review of the manuscript. The points about length and organization are well taken. In the first version, presenting new material involving subtle effects and new methods, we erred on the side of presenting the six cases in a uniform and detailed manner, to ensure that the major points came across. This reviewer feedback is very helpful toward a more efficient and elegant organization of the paper.

---

## Referee Comment (RC3) · Anonymous Referee #3 · 30 Nov 2020

General comments:

This paper presents a very interesting piece of work. It is innovating and brings together existing technologies. The methods are clear and the analysis is thorough. The paper applies the method of Scott et al. 2019 to identify large-scale meteorological drivers of surface melt to classify melt events at three locations. The mechanisms are thermal blanketing, all wave radiative enhancement and föhn effect. The satellite PMW technique reveals brightness temperature that enable to detect surface melt based on the mean base temperature of the last season exceeded by 30K developed and refined previously.This is complemented by SEB budget from ERA 5. Finally the LWP and IWP are used to infer the presence and type of clouds. This paper classify events to three mechanisms with a future aim to predict future atmospheric stress on Antarctic ice shelves. Some of the sensor suffer from known bias and the distinction between the different mechanisms is not always straightforward, which leads to inconclusive re-

sults. None of the cases is clearly classified and some are mixed type of events, which prohibits a long term forecast of the type and frequency of surface melt.

Overall, the paper is good but I have experienced difficulties to grasp the importance of the biases as they are not explained but only a reference is given. Could you quickly explain the biases/known issues of the sensors... for the readers unaware of the literature? The paper lacks a bit of precision in the terms used, as well as consistency. Then, the structure of the results and discussion part could be improved for an easier and more straightforward read. Specifically, I think that a map of the locations with air masses direction superimposed could help. Then, consider grouping the cases by location or by driving mechanism, or by temporal period? In its current state, the results and discussion section is very lengthy and repetitive and it is difficult to have the results emerge. Finally, I think that you could split the results and discussion section in two separate sections. That would allow for clarity as you could describe the events individually and then discuss them in relation to the others and in light of the different mechanisms in the discussion section.

Specific comments:

P2 L40-43 : you talk about surface melt then end with tropospheric warming but there is no link established between the warming and the surface melting.

P2 L70: could you give a magnitude instead? (e.g. a magnitude x times smaller than...)

P3 L73: what do you mean with "the SEB does not close" and in general, this paragraph could be rephrased: explaining eq 1, then what a positive or negative ME entails, then mentioning SH and LH, then G (currently SH is mentioned, then G then SH again).

P3 L86: can you rephrase "at some point during the episode" ?

P4 L104-106: please rephrase

P4 L109-110: either leave this out or comment on the link with what you are studying

[Figure]

P4 L112-119: link your description of föhn winds to surface melt. (e.g. brings a large positive turbulent flux input to the surface, great enough to initiate surface melt )

P4 L124-129: this is great but is never exploited afterwards, mention this also in your conclusion

P5 L146: Add a general map of Antarctica with an inset indicating your location. Add meridians and parallels of reference. Also, delimit the areas of study (the pixels used for each of the regions for instance)

P5 L146-147: The Pine Island and Thwaites Glacier region presents the greatest concern for the West Antarctic Ice Sheet : please support with a reference

P7 L180: please rephrase, it is not clear whether ERA5 or the AWARE data is over/underestimating

P7 L183: what are "ERA5 cloud microphysical discrepancies" and how they can impact a ME time series? please rephrase

P7 L205: why do you change location

P7 L208: please expand the sentence on errors: what are these errors, what cause them and of what magnitude are they?

P8 L222: please support this with a reference

P11 onwards: I suggest you group together by area of study, per month or per driving mechanism or per area as this section is long and tedious to read. A second suggestion is to separate the results from the discussion, having the melt occurrence in the results and the determination of the mechanism in the discussion

P11 L244-249: a map or adding these locations to Fig1 could help here

P11 L249: what is the Bennartz and al. thin cloud range? could you give numbers or a short explanation?

P11 L251: what is "satellite melt"? change to " surface melt detected by the satellite"

P11 L252: begins in some of the region: can you be more precise

P11 L254: can you please explain your sampling method: how fare away are these cells from the others?

P11 L255 + 262: what is a "recovery"? what/whose recovery?

P11 L259: if worth mentioning, also worth giving a possible explanation why?

P11 L264-268: there is no real description of the figure, but a direct jump to explanations

P11 L272 can you please quickly explain what Silber et al defines as unrealistic?

P15 L306-310 not all the locations mentioned here are indicated on Figure1, it would be nice to have a visual illustration of this paragraph or add the locations on Figure 1

P19 L351: refer to case 3.2 explicitly (or merge the discussion of the two cases together)

P19 L354: you mention the "dry snow range", can you quickly explain what this means, and do the same for the "dry surface range" mentioned on P15 L315?

P19 L359-361: There are no plots of spatial variability of the melt components, what are you referring to? I do not understand this sentence: are you describing Figure 12 (no spatial component)? then consider adding the figure number explicitly. And what are the three regions considered here?

P19 L363: what does "At this location" refer to? Pine Island or Thwaites Glacier? or both?

P19 L377: do you have a reference for the difficulties for phase discrimination by MODIS?

P19 L447: The Amundsen-Bellingshausen Sea region could be added in Fig 1, same

for the Amundsen sea embayment P28 L459: why "new" SW flux?

P35 Fig 24 : why not plot both stations on the same graph? and maybe also ERA5 (Figure 21) for the near surface wind speed. that would allow for a direct comparison of the two datasets

P32 L521: I could not find a Figure of ERA5 2m surface temperature that you mention. Consider adding it and plotting AWS data and ERA5 on the same graph for near surface wind speed and temperature (see previous comment)

P34 L548- P36 L560: consider grouping the events by mechanism rather than event/date

P36 L566 to P37 L586: this looks to me more a discussion than a conclusion paragraph

P37 L594-595: I do not understand this sentence: "the number of melt events is a reasonal [...] number for this type of analysis": what are you trying to express?

Technical corrections:

P1 L31: no need to add the MISI acronym you do not use it subsequently

P3 L85: "of an episode"

P7 L185: please rephrase "except often"

P7 L202: Replace "so we examine" by "we therefore examine" or "So, we examine"

P7 L206: remove the "," between melt and event

P8 Fig 2: please rephrase the legend, it is a time-series comparison between two datasets

P8 L217: "if cloud microphysics are" or "the cloud microphysics scheme is"

P11 L261: consider changing "region of interest" by "period of interest/investigated" as region is spatial, not temporal

P11 L263: add a "." at the end of your sentence (Figure6a)

P11 L263: is Figure 4b meant to be Figure 6b?

From page 13 on, there are no a) b) c) in the figures any more

P19 L356: "Figure 9b": do you mean "Figure 11b"?

P19 L369: consider replacing "the result" by "this induces a ..."

P28 L466+467: "freezing point"

P28 L474: "may be overestimated"

P28 L480: remove one of the mentions to "Figure 10"

P30 L502: "increased by nearly 10K"

---

## Author Comment (AC3) · 30 Nov 2020

We thank this referee for a very thoughtful review of the manuscript. The points about improving the graphics and more detailed discussion about sensor biases are well taken, and this review will be very helpful toward improving the final paper.

---

## Author Comment (AC4) · 12 Jan 2021

Manuscript Revision Overview

The three referees collectively have a positive response to our manuscript, find its topic appropriate for the journal, and find that the overall analysis and conclusions are sound. The referees also give several helpful suggestions that when addressed will lead to a stronger final paper. The seven major revisions suggested by the reviewers can be summarized as follows:

1. Provide more background in relation to the existing literature.

2. Make the results section and discussion of the case studies more quantitative.

3. Add synoptic maps to support the meteorological discussions of the case studies.

[Figure]

4. Reorganize presentation of the case studies, both in the text and in the combinations of figures.

5. Investigate the influence of the Amundsen Sea Low on the cases.

6. Revise Figure 1 (map) to include more information.

7. Amplify the discussion of cloud phase and its influence on surface melting.

These major revisions are all entirely doable, and generation of the suggested synoptic maps is already underway.

In addition, the referees collectively give (by our count) 95 minor suggestions and technical/editorial corrections, all of which can also be easily addressed.

In summary, we look forward to revising the manuscript according to the above, and we thank the three referees for their thorough and very helpful reviews.

―――――――――――――――――――

---

## Editor Comment (EC1) · Ruth Mottram (Editor) · 8 Feb 2021

Thanks for your summary of your planned changes. Please upload a new version of the manuscript together with a detailed response to the reviewers so that I can make an editorial decision. Best wishes

---

## Author Comment (AC5) · 30 Apr 2021

**Energetics of Surface Melt in West Antarctica, by Madison L. Ghiz et al., tc-2020-311**

**Replies to Referee #1**

Summary

The manuscript uses multiple data sources (satellite, AWS, reanalysis) to explore the contributing meteorological factors influencing the SEB and surface melting over West Antarctica. Specifically, the authors examine the influence of optically thick cloud that induces a thermal blanketing effect, optically thin clouds that enhance all-wave radiation, and foehn winds. They explore these mechanisms by evaluating several case studies from various locations in West Antarctica.

Adherence to evaluation criteria

This paper specifically examines the effects of processes that have previously been considered in isolation or in a less direct manner, and quantifies their impact on melting and unraveling their effect on the surface energy balance (SEB). This represents new insight and advances our understanding of cloud processes and foehn events on the SEB of West Antarctica, and so demonstrates the originality required for publication in The Cryosphere and fits well within the aims and scope of the journal.

**We thank the referee for this endorsement regarding the manuscript's suitability for The Cryosphere.**

Overall, I am convinced of the scientific rigour of the methodology and believe that the manuscript contains enough detail to make the methods reproducible. I have some minor comments and clarifications on certain aspects of the methods, which are given below. The specific contributions of this work in relation to the existing literature could be outlined more explicitly, but generally there is adequate consideration of related work. The limitations of the method are discussed and addressed appropriately.

The results and conclusions are presented clearly, although the text can be overly descriptive and somewhat repetitive in places. I have provided some suggestions for re-structuring some of the case study sections to improve readability. Overall the language is satisfactory, but there are still some outstanding typos and errors that could be rectified with a thorough proof-read. I have highlighted some below in my comments. The figures used are appropriate but are again repetitive. I have offered some suggestions in my comments that may enhance the reader's understanding of the results.

**We are grateful that the referee finds that the work is sound overall, and this referee makes numerous suggestions that have significantly improved the manuscript.**

The authors reach substantial conclusions regarding the importance of the three mechanisms outlined for determining melt. These findings are likely applicable to other parts of Antarctica, which gives them broader significance. The results fit within the current literature regarding Antarctic SEB and the drivers of melt, and represent a development in our scientific understanding of the effect of cloud properties and atmospheric drivers on melting.
**This is the impact we had hoped for, that this manuscript might present useful new methods for evaluating and projecting future melt on vulnerable West Antarctic ice shelves.**

General points

- The manuscript is very descriptive, at times overly so. It may help the reader if you more explicitly draw the links between the general synoptic conditions that are described at the start of each case study and their impact on the mechanisms involved in the SEB/melt.

**The various suggestions from all three referees have improved the quantitative discussion throughout the manuscript. The revision includes maps showing synoptic conditions and their linkage with the local energetics driving surface melt.**

- Each of the case studies is comprehensively evaluated and a great deal of analysis has evidently been done. However, it does get a little repetitive reading each of the case studies in turn because they all follow the same structure. My suggestion would be to combine the discussion of e.g., Siple + PIG/Thwaites (2[nd] and 3[rd] cases), and possibly the Ross Ice Shelf (6[th] case) into a 'December 2011' case, which might help the reader and facilitate better discussion of the influence of synoptic meteorology on melting at these two/three locations. You can then discuss the ways the same event affects the SEB similarly/differently at the different sites.

**We have adopted this suggestion, and now the results are structured into four case studies instead of six, and the final case study is the "December 2011" large-scale case in which melt occurs from contrasting mechanisms at our three different sites.**

- The similarity of the figures is also a little repetitive. Whilst I understand that this is difficult to avoid because of the nature of your analysis, I wonder if there is a way of combining some elements of the figures to communicate the differences between case studies/mechanisms more concisely? For example, could you show a scatter plot of IWP/LWP vs melt or Rnet for the optically thin/thick cloud cases to see how the optical depth influences these? Or contrast the time series of melt/emissivity for different case studies in one figure with multiple sub-plots? Some spatial representation of these case studies, e.g., maps of ERA-5 modelled ME, synoptic meteorology, emissivity or specific fluxes might also help the reader understand these cases better.

**The manuscript reorganization following the previous recommendation should eliminate much of the repetition. However, we find that to discuss the total melt energy in each case, it doesn't help to remove any plots from the composite melt energy figures. All of individual energy components should be available visually for inspection. We do include maps and improved discussion of synoptic meteorology, and plots of spatial variability in the passive microwave brightness temperature from which the melt is detected (in Appendix B).**

- It would be helpful for the reader to more clearly indicate which mechanism you believe is responsible for driving the melt at the start of each case study section, possibly even in the subtitle heading.

**This recommendation is adopted, and each case study discussion now begins with an identification of the mechanism.**

- I would like to see more discussion of the impact of cloud phase on melting. Ice and liquid have quite different radiative impacts, so this might affect your results.

**There is now more discussion of the ice phase versus liquid water, including vertical distribution, particularly in the discussion of our first case (January 2015).**

- More concrete linkages should be drawn between the melt mechanisms and large-scale drivers, such as those identified by Scott et al. (2019).

**With our new figures that lead each case study by depicting the synoptic conditions, the revised manuscript should be much stronger in this respect.**

- If you only refer to something once or twice, you probably don't need to define an acronym for it (e.g. ARM, ARs, GIS, AMRC, DMSP). This helps the reader because they don't have to refer back to remind themselves what the letters stand for.

**The acronym usage has been cleaned up in the revision.**

- Do you see the effect of the Amundsen-Sea Low (ASL) in your results? This has been shown to be linked with the SAM (and ENSO) and can influence the advection of maritime air onto the continent (Hosking et al., 2013 – doi:/10.1002/joc.3558 ; Clem et al., 2017 doi: 10.1175/JCLI-D-16-0891.1)

**We have identified the ASL signature where it appears, and these references are now cited.**

- How do your results compare with high-resolution regional modeling? Are there implications for e.g. surface mass balance? E.g. Lenaerts et al. 2017 doi:10.1017/aog.2017.42; Deb et al., 2018 doi; 10.1029/2018GL077092.

**These are local individual cases, but they do show qualitative consistency with large-scale modeling analyses such as Deb et al. (2018). This is now mentioned in the conclusion and Deb et al. (2018) is now cited.**

Specific points

[L35-37] This would be a good place to reference Lhermitte et al. (2020) on structural destabilization of the West Antarctic ice shelves. Doi: /10.1073%2Fpnas.1912890117

**This reference is now cited as suggested.**

[Para starting L80] quite verbose, could be revised to be clearer

**This paragraph has been revised following several referees' suggestions.**

[L83-85] Does the fact that the satellite retrievals are instantaneous affect melt detection?

**The paragraph should now be clearer in this respect.**

[L105] define GIS on first usage

**Correction made.**

[L106-107] Suggest including a citation to support this point (frequency of cloud LWP 10-40 g m-2 in the Antarctic, plus radiative effect of this cloud)

**Three supporting citations are now given in this paragraph.**

[L111-116] Citation for definition of foehn – e.g. Elvidge & Renfrew (2016) for various mechanisms doi:/10.1175/BAMS-D-14-00194.1

**This citation has been made for the definition of foehn.**

[Section 2] How are foehn detected in the AWS data? Further detail is required here.

**This detail is now provided in this paragraph.**

[L186-194] Discrepancies of 50-100 W m-2 are quite considerable, and certainly could mean the difference between melt and no melt, especially in non-summer seasons. All reanalyses struggle with cloud properties, so you would expect some level of disagreement in terms of radiative fluxes, but I wonder if this could be improved upon. Did you look at other reanalysis products to find out whether any others perform better?

**Looking at other reanalysis than ERA5 is beyond the scope of this work, but our concluding section now mentions another model often used for this type of analysis (RACMO2). RACMO2 does better than ERA5 in a "first order" sense in that more cloud liquid water (and thus more radiation from cloud) is produced, but this is done in the form of tuning the microphysical scheme, and therefore it is still helpful to use satellite cloud property retrievals to gain insight into the exact mechanism in any given melt event case.**

[L206] comma should be after 'event' not 'melt'

**Correction made.**

[Fig 2] Is there missing data in panels d and e between December 10-11?

**There is a small segment of missing data, and this is now mentioned.**

[L263] missing full stop after parenthesis

**Correction made.**

[L276] misplaced comma – add after 'daily statistics'

**Correction made.**

[L 280-281] Explanation of Fig 5b is unclear. Suggest that you explain which 5 cells were selected and/or how they were chosen, and rephrasing to more clearly state that these are expressed as percentiles of the max Tb observed with satellite retrievals.

**This explanation has been clarified.**

[L292] Some comment on how you might expect the high IWP values used in the ERA5 radiative transfer scheme to influence the radiative fluxes would be informative here. e.g. competing SW/LW effects

**This paragraph has been expanded to provide the requested explanation.**

[293-295] What is the implication of the greater vertical extent of ice cloud vs liquid regions? How might this influence the surface energy balance? I am wondering about e.g. temperature and altitude effects, as well as how the ice/liquid water paths influence LW/SW scattering and emission of the clouds. Cloud vertical profile is shown to be important in determining radiative fluxes and melt in e.g. Gilbert et al. (2020) doi:10.1002/qj.3753 on the peninsula in summer.

**In addition to amplifying this paragraph, Gilbert et al. (2020) is now cited.**

[L322] missing 'the' before 'lowest sun' ?

**Correction made.**

[L323-325] Quantifying these statements would be helpful. What, specifically does 'unexceptional' mean? Perhaps give an average. Similarly for 'moderate to high LWP'.

**This paragraph has been made more quantitative.**

[para beginning L318] Presumably the source of the SH flux is warm air advection associated with the warm air intrusion during Dec 2011? Possibly worth re-stating this here.

**Restated as suggested.**

[327-330] Further explanation of the mechanisms causing delayed-onset melt and then sustained melt would be informative.

**Further explanation is now given, referring to the magnitude of the ME.**

[L331-332 (Fig 8 caption)] see comment above at line 280-281.

**This explanation has been clarified.**

[L344] Not sure 'low overcast' is a noun. Please revise (e.g. low cloud/ overcast conditions/ similar)

**Revision made as suggested.**

[L358-361] Can you speculate about the sources of inhomogeneity in the emissivities? E.g. snow properties, surface albedo, melt ponding, topography etc.?

**Additional discussion is given as suggested**.

[L381-383 (Fig 11 caption)] See comment above at line 280-281.

**This explanation has been clarified.**

[Section 3.3] – I would consider combining this section with 3.2 (and possibly 3.6) in order to better discuss the influence of synoptic conditions on the cloud properties, emissivity and SEB at these locations. As it is, the manuscript is quite repetitive as each case study follows the same format with very similar figures, so this may help the reader dray parallels and distinctions between the same event manifested at different locations.

**The revised manuscript now makes this combination.**

[L435-436 (Fig 14 caption)] See comment above at line 280-281.

**This explanation has been clarified.**

[para beginning 445] – this sounds like the signature of the ASL in the Ross Sea. Could be worth tying this in, to make more connection between large-scale synoptic conditions and local effects.

**The ASL is now included in this discussion.**

[L499-501] This signature (positive SH, negative LH) is frequently indicative of foehn conditions. Adding a statement + citation to this effect would strengthen your argument (e.g. Datta et al., 209; Elvidge et al., 2020, 10.1029/2020JD0032463; Kuipers Munneke et al., 2012, doi: 10.5194/tc-6-353-2012; 2018, doi:10.1029/2018GL077899).

**Excellent point, and it is now added along with these citations.**

[L514-516] Might the optically thin clouds suggested by your results be indicative of the 'cloud clearance' effect of foehn? i.e. clouds that might have been optically thicker in the absence of foehn (such as those you have shown at PIG/Thwaites/Siple Dome) due to the presence of warm, most air from the Southern Ocean are thinned by the warming/drying effect of foehn?

**This discussion now includes the possibility of foehn "cloud clearing."**

[para beginning L566] Might ground-based remote sensing from the AWARE project be useful for examining cloud microphysics? Could this be a way of improving the detection of these sorts of cloud-driven melt events?

**The potential utility of AWARE data is now mentioned in this paragraph.**

**Replies to Referee #2**

This research looks at how various meteorological conditions alter the surface energy balance (SEB) and produce melt at a number of locations in the West Antarctic region. Surface melting, even during summer, is still spatially limited in the Antarctic, compared to Greenland. However, recent studies, including this one, have highlighted the potential for intense melt events on the surface of ice shelves, which has important implications for ice shelf stability and dynamics. The authors present a number of case studies and explain the causes for the melt, and which SEB components initiated it. A range of processes and mechanisms are included, such as thick thermal-blanket clouds, thin cloud cover and föhn winds. The authors use a range of available data and evaluate it thoroughly. I believe this work is novel, relevant and an important contribution to the field. I also fits the scope of the journal very well. An interesting aspect was the inclusion of manual observations of clouds and visibility, which you don't regularly read in manuscripts now, but undoubtedly offer important insight into cloud characteristics as they are so poorly represented in reanalysis and models. Whilst the analysis is robust and thorough, the manuscript suffers from some presentation issues as well as being overly long and descriptive. Some of this is due to the number of case studies, which leads to repetition of both figure types and meteorological description. However, I think that with some re-structuring and inclusion of some maps, the manuscript is a good scientific publication which should be accepted. I have outlined the larger issues below and also included line-references for more specific edits. Whilst I give some areas to cut out and reduce the length, I do urge the authors to also go through and decide on any other areas which can be slimmed down.

**We thank the referee for endorsing this work's novelty and suitability for The Cryosphere, and also appreciate the mention of our using field camp observations, which are indeed underutilized. We have adopted the referee's suggestions for restructuring and clarification on maps.**

Minor suggestions: The length of the manuscript and the repetition in results and their presentation currently makes it feel a little harder to read than it needs to be. There are 6 case studies provide, which is quite a lot for a 'case study' style paper. However, all are valid and important. I therefore suggest restructuring or combining some case studies. Here I have some potential ideas, but I think the authors should decide what works best for them and the manuscript. 1) Combine the December 2011 case studies. Whilst different processes are at play in these case studies, the larger-scale processes and synoptic situation will be the same/similar. Combining them to discuss the elements which are the same will reduce repetition, whilst highlighting the differences between them which show how local-scale processes are important for melting. 2) Combine the case studies by location. E.g combine the PI and Thwaites case studies for December 2011, January 2012 and February 2013. Again, I know that different processes are at play, but it could be advantageous to plot these together to see how melt magnitude/length etc vary between the different meteorological conditions. 3) Combine by melt-mechanism: combine the 'thermal blanket' cases, the ones dominated by turbulent fluxes etc. This would highlight which situation happens more frequently, and which ones are less frequent (but could perhaps generate more melt). The conclusion summarizes the case studies very well, and groups them into the cases that have an individual mechanism, and ones that have interplaying mechanisms. Perhaps you could try to include this structure and succinct approach throughout the results. I really like the conclusions section: it is clear, rounds the paper off well and includes the limitations to the study.

**Following these suggestions and those of the other referees, we have grouped and defined our case studies by the driving synoptic condition, and each case study discussion leads off with a synoptic-scale map. We appreciate the referee's endorsement of the conclusion section.**

The introduction is quite long and currently reads a bit like thesis chapter or review paper. This hides your motivation a little, even though there is a need for your research. I would move the section on SEB modelling in to the data/methods section, and slim it down. Currently your objectives are at the end of the fourth page, but could be at the forefront if some lines/paragraphs were removed.

**As suggested, we have moved the SEB background section to the start of Section 2, and now the introduction is indeed more succinct.**

Be clearer with which AWS observations you use. As they are mentioned in the data section, but only in passing (e.g line 155/156). And they are mentioned in the figure captions (e.g Fig 20 'over the RIS regions containing Tom and

Sabrina'), but I couldn't be sure where exactly the data are used besides Figure 24. If you do use them elsewhere, you need to make it clearer which locations and which variables. I would also include them on your Figure 1 map.

**The Figure 1 map now includes the AWS locations. With the manuscript now restructured, it should be clearer when and where the Tom and Sabrina AWS data are used.**

Figure: Figure 1 should be improved to include more information. For instance, an insert of the whole of Antarctica is needed with indication of where this is zoomed in section is. Number 3 is missing. Could you add the AWS locations? Some schematics or overlays could aid the reader in understanding the large-scale flow in the region, or some arrows to highlight the main air flow directions in your case studies. Additional figures would also be preferential in highlighting main weather systems for the different case studies. Often in case study analysis (especially in atmosphere-SEB studies), there is a modelling aspect which provides some cross sectional of 2d plots with maps of airflow and such things, which aids understanding for the reader. I am not suggesting you do extra work, but only that you use figures to better represent more useful information. Perhaps some synoptic charts would help show information, such as the Dec 2011 case with the low-pressure system and föhn winds. Such maps could also indicate the locations of the grid cells used to look at the spatial patterns of emissivity. Many figures of the same style and type- a consequence of a multi-case analsysis- but could be panelled together to avoid repetition. Or move some to a supplement if some of them are not definitely useful. For instance, is the LWP and IWP plot needed for the case studies where cloud properties are less relevant?

**Figure 1 has been substantially improved with more information, and synoptic maps are now included as requested. Reorganizing the case studies around the driving synoptic condition now avoids repetition, though we do find it helpful (and perhaps necessary) to include all of the melt energy components in the figures, so the reader is entirely clear about what influences the total melt energy.**

Some of the decision making on ERA5 vs CERES cloud be moved to a supplement. You could state only what you finally use in the manuscript and move the evaluations and mix-match tests to supplement. Otherwise, it is quite hard to follow which method/dataset you are actually using. For example, after line 200/page 7, I assumed that you weren't using ERA5 anymore. However, you stated at the end of page 9 that you do indeed use ERA5.

**As suggested, we have moved this discussion and all three related figures to Appendix A.**

Is there a timing offset between ERA5 and CERES LWP? In some of the figures, the peaks appear to be always a day or two later.

**The timing offsets of a day or two probably just reflect uncertainties in the reanalysis' numerical modeling, compared with the timing of the cloud cover that the satellite imager (MODIS in the CERES data product) actually observes.**

Specific Suggestions:

Ln 105: GIS hasn't been defined previously

**Correction made.**

Ln 106: Do you have a reference or evidence that these clouds are common in the polar regions?

**References are now given.**

Ln 110: Föhn winds are only common in area with mountain ranges and perpendicular to air flow, which is a relatively small area of the Antarctic. I would say, common in some regions within the Antarctic. Ln 118,119: Föhn winds can't occur anywhere or on any ice shelf- there are many ice shelves in the East where föhn winds do not occur. These may be affected by katabatic winds, but evidence of föhn is lacking. Zhou et al. 2018 only looked at W. Antarctic ice shelves and föhn. Föhn are only common on Antarctic Peninsula are Ross Ice Shelf.

**The paragraph has been revised following these comments.**

Throughout: do you have any DOIs or URLs to point the reader to the sources of data?

**URLs are now given in the Data Availability statement.**

Ln 206: Add 2016 to the data here.

**Correction made.**

Ln 250-260: I have a hard time knowing how spatially uniform (ln 256) the surface emissivity is, when you don't say which 5 grid cells you're looking at. Where are the 5 grid cells? This sort of information would be better represented as a spatial plot on top of a map, so that we can see the area of interest.

**We now include an Appendix B that gives examples of the spatial variability in satellite-measured brightness temperature.**

Figure 6 a/b/c…missing on figures.

**The figures have all been corrected.**

Figure 7a/10a: timing offset?

**See above response on this topic.**

Ln 356: 'In Figure 9 the sampled percentiles are reference to the max Tb on 21 December'. I don't understand what this means. Is it the wrong Figure you are pointing to?

**This discussion has been clarified, per the first referee's suggestion.**

Ln 594-595: I don't understand what you're trying to say here.

**This sentence has been removed, as it was too speculative.**

Figure 24: are you able to plot RH? As this is a key variable in determining föhn winds.

**We now include RH from the AWS in this final figure, and a related discussion.**

**Replies to Referee #3**

General comments:

This paper presents a very interesting piece of work. It is innovating and brings together existing technologies. The methods are clear and the analysis is thorough. The paper applies the method of Scott et al. 2019 to identify large-scale meteorological drivers of surface melt to classify melt events at three locations. The mechanisms are thermal blanketing, all wave radiative enhancement and föhn effect. The satellite PMW technique reveals brightness temperature that enable to detect surface melt based on the mean base temperature of the last season exceeded by 30K developed and refined previously. This is complemented by SEB budget from ERA 5. Finally the LWP and IWP are used to infer the presence and type of clouds. This paper classify events to three mechanisms with a future aim to predict future stress on Antarctic ice shelves. Some of the sensor suffer from known bias and the distinction between the different mechanisms is not always straightforward, which leads to inconclusive results. None of the cases is clearly classified and some are mixed type events, which prohibits a long term forecast of the type and frequency of surface melt.

**We are grateful that the referee finds the work interesting. It is true that some of our melt event cases turn out to have mixed mechanisms causing them (clouds in conjunction with sensible heat and/or foehn winds), but this type of analysis can identify this complexity. Nevertheless, the referee is correct that some melt events cannot yet be fully diagnosed, and we emphasize this in our January 2012 case study.**

Overall, the paper is good but I have experienced difficulties to grasp the importance of the biases as they are not explained fully but only a reference is given. Could you quickly explain the biases/known issues of the sensors… for the readers unaware of the literature?

**A brief discussion of the cloud remote sensing biases is now given, along with the standard references.**

The paper lacks a bit of precision in the terms used, as well as consistency. Then, the structure of the results and discussion part could be improved for an easier and more straightforward read. Specifically, I think that a map of the locations with air masses direction superimposed could help. Then, consider grouping the cases by location or by driving mechanism, or by temporal period? In its current state, the results and discussion section is very lengthy and repetitive and it is difficult to have the results emerge. Finally, I think that you could split the results and discussion section into two separate sections. That would allow for clarity as you could describe the events individually and then discuss them in relation to the others and in light of the different mechanisms in the discussion section.

**As suggested, the cases are now grouped by synoptic driving mechanism, and each case study discussion leads off with a synoptic-scale map showing relevant air mass direction. With the reorganization, there is less of a need to split off a separated discussion section prior to the conclusions.**

Specific comments:

P2 L40-43 : you talk about surface melt then end with tropospheric warming but there is no link established between the warming and the surface melting.

**A linkage is now given with the Lhermitte et al. (2020) reference.**

P2 L70: could you give a magnitude instead? (e.g. a magnitude x time smaller than…)

**A magnitude is now given, along with a reference (van As et al., 2005).**

P3 L73: what do you mean with "the SEB does not close" and in general, this paragraph could be rephrased: explaining eq 1, then what a positive or negative ME entails, then mentioning SH and LH, then G (currently SH is mentioned, then G then SH again).

**This paragraph has been restructured and clarified as requested.**

P3 L86: can you rephrase "at some point during the episode"?

**The word "interval" is now used.**

P4 L104-106: please rephrase

**Not sure what is specifically unclear here, but the acronym Greenland Ice Sheet is now spelled out.**

P4 L109-110: either leave this out or comment on the link with what you are studying.

**A link is now made with one of our case studies.**

P4 L112-119: link your description of fohn winds to surface melt. (e.g brings a large positive turbulent flux input to the surface, great enough to initiate surface melt)

**The description has been revised as suggested**.

P4 L124-129: this is great but is never exploited afterwards, mention this also in your conclusion

**The conclusion is now amplified compared with the previous version.**

P5 L146: Add a general map of Antarctica with an insert indicating your location. Add meridians and parallels of reference. Also, delimit the areas of study (the pixels used for each of the regions for instance)

**All of these revisions have been made in the new Figure 1.**

P5 L146-147: The Pine Island and Thwaites Glacier region represents the greatest concern for the West Antarctic Ice Sheet : please support with a reference

**This is now supported by the Alley et al. (2015) reference.**

P7 L180: please rephrase, it is not clear whether ERA5 or AWARE data is over/underestimating

**This has been clarified.**

P7 L183: what are "ERA5 cloud microphysical discrepancies: and how can they impact a ME time series? please rephrase

**The sentence has been rephrased as requested.**

P7 L205: why do you change location

**This discussion is now in Appendix A, which helps clarify that we want to use the reanalysis and satellite methods independently from surface data. Hence a move from WAIS Divide to Siple Dome.**

P7 L208: Please expand the sentence on errors: what are these errors, what cause them and of what magnitude are they?

**This phrase has been removed, as there is not actually enough related discussion in Silber et al. (2019).**

P8 L222: please support this with a reference

**Now supported by Silber et al. (2019).**

P11 onwards: I suggest you group together by area of study, per month or per driving mechanism ore per area as this section is long and tedious to read. A second suggestion is to separate the results from the discussion, having the melt occurrence in the results and the determination of the mechanism in the discussion

**The manuscript and case studies are now reorganized along these lines.**

P11 L244-249: a map or adding these locations to Fig1 could help here

**The new Figure 1 includes these locations.**

P11 L249: what is the Bennartz et al. thin cloud range? Could you give numbers or a short explanation?

**The quantitative range is now given.**

P11 L251: what is "satellite melt"? change to "surface melt detected by the satellite"

**Change made as requested.**

P11 L252: begins in some of the region: can you be more precise

**An explanation is now given referencing the daily box plots.**

P11 L255 + 262: what is a "recovery"? what/whose recovery?

**Now rephrased to omit the word "recovery."**

P11L259: if worth mentioning, also worth giving a possible explanation why?

**The sentence has been omitted in the interest of conciseness.**

P11 L264-268: there is no real description of the figure, but a direct jump to the explanations

**The figure is now explained in the caption.**

P11 L272: can you please quickly explain what Silber et al defines as unrealistic?

**This has been rephrased and clarified.**

P15 L306-310 not all the locations mentioned here are indicated on Figure1, it would be nice to have a visual illustration of this paragraph or add the locations on Figure 1

**All of the locations now appear on Figure 1.**

P19 L351: refer to case 3.2 explicitly (ore merge the discussion of the two cases together)

**The cases are now merged under "December 2011."**

P19 L354: you mention the "dry snow range", can you quickly explain what this means, and to the same for the "dry surface range" mentioned on P15 L315?

**This has been clarified, and a reference given (Mätzler, 1987).**

P19 L359-361: There are no plots of spatial variability of the melt components, what are you referring to? I do not understand this sentence: are you describing Figure 12 (no spatial component)? Then consider adding the figure number explicitly. And what are the three regions considered here?

**This sentence has been deleted.**

P19 L363: what does "At this location" refer to? Pine Island or Thwaites Glacier? or both?

**This has been clarified (both glaciers).**

P19 L377: do you have a reference for the difficulties for phase discrimination by MODIS?

**The Platnick et al. (2017) reference is now cited.**

P19 L447: The Amundsen-Bellingshausen Sea region could be added to Fig 1, same for the Amundsen sea embayment

**These have been added to Figure 1.**

P28 L459: why "new" SW flux?

**Corrected to "net."**

P35 Fig 24 : why not plot both stations on the same graph? And maybe also ERA5 (Figure 21) for the near surface wind speed. that would allow for a direct comparison of the two datasets

**The figure has been revised as suggested.**

P32 L251: I could not find a Figure of ERA5 2m surface temperature that you mention. Consider adding it and plotting AWS data and ERA5 on the same graph for near surface wind speed and temperature (see previous comment)

**The ERA5 2m temperature appears in the final panel of all of the melt component plots. ERA5 has substantial spatial variability in this case study, so a direct comparison with the AWS is difficult.**

P34 L548- P36 L560: consider grouping the events by mechanism rather than event/date

**This is the basis for the manuscript's reorganization.**

P36 L566 to P37 L586: this looks to me more a discussion than a conclusion paragraph

**A stylistic preference; I am accustomed to discussing the major limitations of a study in the conclusions section. But there are other valid ways to introduce this topic.**

P37 L594-595: I do not understand this sentence: "the number of melt events is a reasonal […] number for this type of analysis:" what are you trying to express?

**This sentence has been reviewed as suggested by another referee.**

Technical corrections:

P1 L31: no need to add the MISI acronym you do not use it subsequently

**Correction made.**

P3 L85: "of an episode"

**The word "interval" is now used.**

P7 L185: please rephrase "except often"

**Correction made.**

P7 L202: Replace "so we examine" by "we therefore examine" or "So, we examine"

**Correction made.**

P7 L206: remove the "," between melt and event

**Correction made.**

P8 Fig 2: please rephrase the legend, it is a time-series comparison between two datasets

**Correction made.**

P8 L217: "if cloud microphysics are" or "the cloud microphysics scheme is"

**Correction made.**

P11 L261: consider changing "region of interest" by "period of interest/investigated" as region is spatial, not temporal

**Correction made.**

P11 L263: add a "." At the end of your sentence (Figure6a)

**Correction made.**

P11 L263: is Figure4b meant to be Figure 6b?

**Figure numbering is now consistent.**

From page 13 on , there are no a) b) c) in the figures any more

**The figures have been corrected.**

P19 L356: "Figure 9b": do you mean "Figure 11b"?

**Figure numbering is now consistent.**

P19 L369: consider replacing "the result" by "this induces a …"

**Change made as requested.**

P28 L466+467: "freezing point"

**Correction made.**

P28 L474: "may be overestimated"

**Correction made.**

P28 L480: remove one of the mentions to "Figure 10"

**Correction made.**

P30 L502: "increased by nearly 10K"

**Correction made.**